# Shift Before You Learn: Enabling Low-Rank Representations in Reinforcement Learning

**Bastien Dubail**
KTH, Stockholm, Sweden
`bastdub@kth.se`

**Stefan Stojanovic**
KTH, Stockholm, Sweden
`stesto@kth.se`

**Alexandre Proutiere**
KTH, Digital Futures, Stockholm, Sweden
`alepro@kth.se`

## Abstract

Low-rank structure is a common implicit assumption in many modern reinforcement learning (RL) algorithms. For instance, reward-free and goal-conditioned RL methods often presume that the successor measure admits a low-rank representation. In this work, we challenge this assumption by first remarking that the successor measure itself is not approximately low-rank. Instead, we demonstrate that a low-rank structure naturally emerges in the shifted successor measure, which captures the system dynamics after bypassing a few initial transitions. We provide finite-sample performance guarantees for the entry-wise estimation of a low-rank approximation of the shifted successor measure from sampled entries. Our analysis reveals that both the approximation and estimation errors are primarily governed by a newly introduced quantitity: the spectral recoverability of the corresponding matrix. To bound this parameter, we derive a new class of functional inequalities for Markov chains that we call Type II Poincaré inequalities and from which we can quantify the amount of shift needed for effective low-rank approximation and estimation. This analysis shows in particular that the required shift depends on decay of the high-order singular values of the shifted successor measure and is hence typically small in practice. Additionally, we establish a connection between the necessary shift and the local mixing properties of the underlying dynamical system, which provides a natural way of selecting the shift. Finally, we validate our theoretical findings with experiments, and demonstrate that shifting the successor measure indeed leads to improved performance in goal-conditioned RL.

## 1 Introduction

In reinforcement learning (RL), the complexity of environment dynamics requires structural assumptions to achieve statistical efficiency. A widely adopted approach assumes that key quantities admit low-dimensional feature representations, effectively imposing low-rank structure on matrices underlying various RL components such as the Q-function (57; 55; 60), transition kernel (2; 33; 59), graph Laplacian (46; 47; 64), and successor representation (15; 58; 5). Some works even aim to learn universal low-dimensional representations transferable across tasks, as in Forward-Backward models (62; 63) and goal-conditioned RL (3; 21). Despite their empirical success and emerging theoretical analyses, fundamental questions remain:

*Why should low-rank structure arise in MDPs, and under what conditions does it yield accurate, learnable representations?*

39th Conference on Neural Information Processing Systems (NeurIPS 2025).

To address these questions, we examine how the long-term dynamics of an MDP naturally give rise to global structure that can be captured effectively through low-rank approximations. In particular, we demonstrate that a simple temporal shift of the successor measure can substantially improve its alignment with low-rank structure. This shift reweights transitions to emphasize long-term behavior, filtering out short-term noise and amplifying the structural signal present in the dynamics. Crucially, its effectiveness hinges on the mixing properties of the underlying Markov chain, which determine how rapidly the process forgets its initial conditions and reveals coherent global patterns. Our main contributions are:

(a) We introduce the notion of spectral recoverability (Definition 3) to quantify the approximation error incurred by low-rank representations. We show that standard successor measures lack spectral recoverability (Proposition 1), motivating the use of shifted

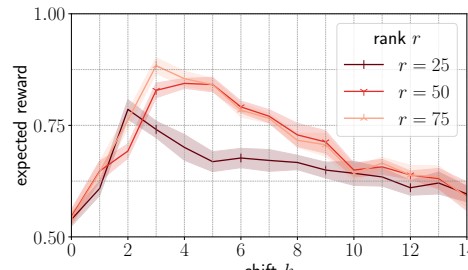

Figure 1: The discrete Medium PointMaze environment (see Section 6). Performance of goal-conditioned RL based on the rank-$r$ approximation of the $k$-shifted successor measure. Peak performance occurs at a non-zero shift, suggesting that shifting the successor measure can improve policy learning under low-rank constraints.

successor measures which discard initial transitions and emphasize long-term dynamics. We prove that sufficiently large shifts guarantee spectral recoverability (Section 5).

(b) We provide finite-sample performance guarantees for the entry-wise estimation of a low-rank approximation of the shifted successor measure from sampled entries (Thm. 1). Our analysis reveals that the estimation error is also governed by the spectral recoverability of the shifted successor measure.

(c) To characterize when spectral recoverability holds, we introduce a novel class of functional inequalities for Markov chains, which we call Type II Poincaré inequalities (Thm. 2). These inequalities allow us to quantify the amount of shift required for effective low-rank approximation and estimation. Moreover, we relate the required shift to the local mixing properties of the underlying dynamical system. These properties measure the extent to which the state space admits a decomposition into subsets within which the local dynamics mix rapidly.

(d) Finally, we validate our theoretical insights through experiments on learning the shifted universal successor measure in goal-conditioned RL. This representation enables the simultaneous learning of optimal policies for reaching a variety of goals. A representative result is shown in Figure 1.

## 2   Related Work

**Low-rank approximations in RL.**   Low-rank models are ubiquitous in reinforcement learning. These models rely on low-rank approximations of certain matrices: most notably the Laplacian (46; 47; 42; 64; 35; 25) and the successor representation (15; 58; 36; 43; 41; 62; 63), the latter often considered a better candidate for low-rank modeling (63). While these models are empirically effective and supported by intuitive heuristics based on spectral properties (see e.g. (38)), they often lack rigorous theoretical justification. Our work aims to address this gap by establishing a connection between low-rank structures and the mixing behavior of the underlying dynamics.

**Sample complexity bounds.**   Numerous studies have established performance guarantees for estimating low-rank structures in reinforcement learning (RL). Several approaches draw inspiration from matrix completion techniques and have been applied, for example, to the estimation of the Q-function (57; 55; 65; 60). Our work is closer to the low-rank/linear Markov Decision Process (MDP) framework explored in (66; 2; 68; 69; 59), where the transition kernel is modeled as a bilinear factorization of the form $P(s, a, s') = \psi(s, a)^\top \phi(s')$. A special case arises when the factors are constrained to be non-negative, yielding models such as (soft) state aggregation and block MDPs (19; 56; 68; 28). To the best of our knowledge, we are the first to analyze the sample complexity of estimating successor measures. Importantly, since successor representations are typically full-rank, imposing a strict low-rank assumption would be inappropriate. Alternative notions of rank have been proposed in the function approximation setting (30; 61; 18; 32); however these depend not only on

the dynamics but also on the choice of the function class. In contrast, our analysis does not rely on function approximation or any structural assumptions, and allows intrinsic structure to emerge naturally from the mixing properties of the underlying dynamics.

**Mixing phenomena.** To bridge matrix estimation and dynamical behavior, we introduce spectral recoverability, a parameter that quantifies both the SVD truncation error and the difficulty of recovering matrix entries from partial observations. Our approach is inspired by (11), who established minimax bounds for matrix completion under a bounded nuclear norm. In contrast, we focus on entrywise estimation, which requires consideration of singular vectors. Spectral recoverability thus blends classical notions of coherence and nuclear norm, enabling entrywise error analysis via the leave-one-out technique of (1). On the other hand, it connects to classical mixing measures in Markov chain theory and can thus be bounded by revisiting classical tools such as functional inequalities (17) and spectral analysis (22). However, unlike traditional approaches that focus on global mixing times, our focus is on statistical estimation for which local and thereby weaker notions of mixing may suffice. This geometric intuition shares conceptual similarities with the works of (45; 29) on decomposable Markov chains and of (39) on spectral partitioning of graphs via eigenvectors of the adjacency matrix, which thus also connect with the block Markov chains mentioned previously.

## 3    Preliminaries

### 3.1    MDPs and shifted successor measures

Consider a Markov Decision Process (MDP) with finite state space $\mathcal{S}$ and action space $\mathcal{A} := \bigcup_{s \in \mathcal{S}} \mathcal{A}_s$, where $\mathcal{A}_s$ denotes the set of actions available in state $s$. Define the set of state-action pairs as $\mathcal{X} := \bigcup_{s \in \mathcal{S}} \{s\} \times \mathcal{A}_s$, and let $n$ denote its cardinality. We write $x = (s, a)$ to denote a generic element of $\mathcal{X}$. The dynamics of the MDP are governed by a transition matrix $P \in \mathbb{R}^{\mathcal{X} \times \mathcal{S}}$, where $P(s, a, s')$ represents the probability of transitioning to state $s'$ when taking action $a$ in state $s$. A policy is defined as a stochastic matrix $\pi \in \mathbb{R}^{\mathcal{S} \times \mathcal{A}}$, where $\pi(s, a)$ denotes the probability of selecting action $a$ in state $s$. The policy $\pi$ induces a Markov chain over $\mathcal{X}$ with transition matrix $P_\pi$, defined as: $P_\pi((s, a), (s', a')) = P(s, a, s')\pi(s', a')$. The MDP is completed by specifying a reward function $R : \mathcal{X} \to \mathbb{R}$. When the state-action pair $(s, a)$ is visited at time step $t \geq 0$, a reward of $R(s, a)$ is received. Given a discount factor $\gamma \in (0, 1)$ the performance of a policy $\pi$ is characterized by its Q-function: $Q^{(R,\pi)}(s, a) = \mathbb{E}\left[ \sum_{t \geq 0} \gamma^t R(s_t^\pi, a_t^\pi) \mid (s_0^\pi, a_0^\pi) = (s, a)\right]$, where $(s_t^\pi, a_t^\pi)$ is the state-action pair visited under $\pi$ at time $t$, or through its value function $V^{(R,\pi)}(s) := \sum_{a \in A_s} \pi(s, a) Q^{(R,\pi)}(s, a)$.

The Q-function can be expressed as a matrix-vector product. To make this explicit, define the successor measure as $M_\pi := (I - \gamma P_\pi)^{-1} \in \mathbb{R}^{\mathcal{X} \times \mathcal{X}}$. Then $Q^{(R,\pi)}(s, a) = \sum_{t \geq 0} \gamma^t P_\pi^t R(s, a) = M_\pi R(s, a)$, where we use matrix product notation: $M_\pi R(s, a) := \sum_{(s', a') \in \mathcal{X}} M_\pi((s, a), (s', a')) R(s', a')$. This formulation separates the dynamics from the rewards, showing that evaluating a policy for any reward function reduces to computing $M_\pi$ (15; 63). The problem of estimating the successor measure is referred to as *reward-free policy evaluation*. For this problem, we would like to obtain guarantees w.r.t. the $\| \cdot \|_{\infty,\infty}$ norm defined as $\|A\|_{\infty,\infty} := \sup_{f \in \mathbb{R}^\mathcal{X} : \|f\|_\infty = 1} \|Af\|_\infty$. Indeed, suppose that we have an estimate $\widehat{M_\pi}$ of $M_\pi$, and hence an estimate $\widehat{Q}^{(R,\pi)} = \widehat{M_\pi} R$ of the Q-function. This in turn allows us to improve the policy by acting greedily with respect to $\widehat{Q}^{(R,\pi)}$. However, for this procedure to be reliable, we require entry-wise control over the error in $\widehat{Q}^{(R,\pi)}$, which can be guaranteed by bounding the error in $\widehat{M_\pi}$ in the $\| \cdot \|_{\infty,\infty}$ norm: $\left\| \widehat{Q}^{(R,\pi)} - Q^{(R,\pi)} \right\|_\infty = \left\| \widehat{M_\pi} R - M_\pi R \right\|_\infty \leq \left\| \widehat{M_\pi} - M_\pi \right\|_{\infty,\infty} \|R\|_\infty$. As we show later in the paper, obtaining accurate estimates of $M_\pi$ can be statistically challenging. The objective of this paper is to explain why shifting the successor measure may address this challenge.

**Definition 1** (*k*-shifted successor measure). Let $k \geq 0$. The $k$-shifted successor measure is defined as $M_{\pi,k} := P_\pi^k (I - \gamma P_\pi)^{-1}$.

The $k$-shifted successor measure $M_{\pi,k}$ captures the dynamics of policy $\pi$ starting from time step $k$ onward. It allows us to quantify the cumulative discounted reward collected under $\pi$ after the first $k$ steps. For any reward function $R$, it satisfies: $M_{\pi,k} R(s, a) = \sum_{t \geq 0} \gamma^t P_\pi^{t+k} R(s, a)$.

## 3.2 Measure-induced norms and SVD

To analyze the accuracy of estimators of the (shifted) successor measure w.r.t. to the $\|\cdot\|_{\infty,\infty}$ norm and make the link with mixing phenomena, we will use measure-induced norms and SVD (refer to Appendix A.1 for a detailed description). Consider a probability measure $\nu$ on $\mathcal{X}$ whose support is $\mathcal{X}^1$. For $f, g \in \mathbb{R}^\mathcal{X}$, define the $\nu$-scalar product as $\langle f, g \rangle_\nu := \sum_{x \in \mathcal{X}} \nu(x) f(x) g(x)$, so that $(\mathbb{R}^\mathcal{X}, \langle \cdot, \cdot \rangle_\nu)$ is a Hilbert space. We define for all $f \in \mathbb{R}^\mathcal{X}$, $M \in \mathbb{R}^{\mathcal{X} \times \mathcal{X}}$ the $\nu$-induced norms as: for any $p, q \in [1, \infty]$,

$$\|f\|_{\ell^p(\nu)} := \begin{cases} \left( \sum_{x \in \mathcal{X}} \nu(x) |f(x)|^p \right)^{1/p} & \text{if } p < \infty \\ \max_{x \in \mathcal{X}} |f(x)| & \text{if } p = \infty \end{cases} , \qquad \|M\|_{\ell^p(\nu), \ell^q(\nu)} := \sup_{f \in \mathbb{R}^\mathcal{X}:\, f \neq 0} \frac{\|Mf\|_{\ell^q(\nu)}}{\|f\|_{\ell^p(\nu)}}.$$

For simplicity, we keep the measure implicit and use the notation $\|f\|_p = \|f\|_{\ell^p(\nu)}$ and $\|M\|_{p,q} = \|M\|_{\ell^p(\nu), \ell^q(\nu)}$. Note that $\|\cdot\|_\infty$ does not depend on $\nu$. We will be mostly interested in the spectral norm $\|M\|_{2,2}$ and the two-infinity norm $\|M\|_{2,\infty}$, as we always have $\|\cdot\|_{\infty,\infty} \leq \|\cdot\|_{2,\infty}$. Using $\nu$, we can define the notions of adjoint of a vector $f$ and of a matrix $M$: $f^\dagger(x) = \nu(x) f(x)$ and $M^\dagger(x, y) = \frac{\nu(x) M(y, x)}{\nu(y)}$. This allows us to revise the notion of singular value decomposition by replacing the usual transpose operator with the adjoint.

**Definition 2** ($\nu$-SVD). The $\nu$-SVD of the matrix $M \in \mathbb{R}^{n \times n}$ takes the form $M = U \Sigma V^\dagger$ where $\Sigma = \mathrm{Diag}((\sigma_i)_{i=1}^n)$ is a diagonal matrix made of non-negative values that we always assume to be in non-increasing order: $\sigma_1 \geq \sigma_2 \geq \ldots$, while $U, V \in \mathbb{R}^{n \times n}$ are unitary in the sense $U^\dagger U = U U^\dagger = I$ and $V^\dagger V = V V^\dagger = I$. The $\nu$-SVD can be expressed as $M = \sum_{i=1}^n \sigma_i \psi_i \phi_i^\dagger$, where the left and right singular vectors $(\psi_i)_i, (\phi_i)_i$ form orthonormal bases ($\psi_i^\dagger \psi_i = 1$ and $\psi_i^\dagger \psi_j = 0$ for $i \neq j$). The entries of $U, V$ are then $U(x, i) = \sqrt{\nu(i)} \psi_i(x)$, $V(x, i) = \sqrt{\nu(i)} \phi_i(x)$.

Given $r \geq 0$, we write $[M]_r = U_r \Sigma_r V_r^\dagger$ for the $\nu$-SVD truncated to rank $r$ and $[M]_{>r} = M - [M]_r$. We finally note that the usual SVD corresponds to the case where $\nu$ is uniform, up to a normalizing factor $n$. In what follows, to simplify, the $\nu$-SVD is referred to as the SVD.

## 3.3 Spectral recoverability

Our goal is to estimate the (shifted) successor measure with entry-wise guarantees by approximating the corresponding matrix via an estimate of its truncated SVD. Truncated SVD is a well-established technique for matrix approximation when considering the Frobenius or nuclear norm. By the Eckart–Young–Mirsky theorem, for a matrix $M \in \mathbb{R}^{n \times n}$, its rank-$r$ truncated SVD $[M]_r$ provides the optimal rank-$r$ approximation with respect to the Frobenius norm, with error $\|[M]_{>r}\|_F^2 = \sum_{i=r+1}^n \sigma_i^2$ entirely determined by the spectral tail. When estimating the matrix from samples of its entries, the entry-wise error often depends on the coherence of the top $r$ singular vectors. Coherence measures how concentrated or spread out the singular vectors are with respect to the standard basis. High coherence implies that a few entries dominate, making estimation from partial observations harder, while low coherence suggests that all entries are comparably informative. For detailed discussions, see, e.g., (8; 53; 48). In our setting, we adopt a similar notion of coherence. For the top $r$ left singular vectors $(\psi_i)_{i=1}^r$ of $M$, we define the coherence as: $c((\psi_i)_{i=1}^r) := \frac{1}{r} \|U_r\|_{2,\infty}^2 = \max_{x \in [n]} \frac{1}{r} \sum_{i=1}^r \psi_i(x)^2$.

When we seek guarantees in entry-wise norms such as $\|\cdot\|_{2,\infty}$ or $\|\cdot\|_{\infty,\infty}$, it is not clear whether the truncated SVD $[M]_r$ still yields a meaningful approximation of $M$. It is also not obvious what quantity governs the estimation error when attempting to recover $[M]_r$ from sampled entries. To address these questions, we introduce the concept of spectral (ir)recoverability, which serves as a suitable quantity for controlling the approximation and estimation errors when the $\|\cdot\|_{2,\infty}$ or $\|\cdot\|_{\infty,\infty}$ norms are considered.

**Definition 3** (Spectral (ir)recoverability). Let $M \in \mathbb{R}^{n \times n}$ and let $M = \sum_{i=1}^n \sigma_i \psi_i \phi_i^\dagger$ be its SVD. The spectral irrecoverability of $M$ is $\xi(M) := \max_{x \in [n]} \sum_{i=1}^n \sigma_i \psi_i(x)^2$. The spectral recoverability is $\xi(M)^{-1}$.

The spectral irrecoverability of a matrix $M$ can be interpreted as a nuclear norm weighted by the left singular vectors of $M$, and it quantifies both the low-rank structure and coherence of the matrix. As

---

[1]We discuss extensions where this is not the case in Appendix C.

stated in the following lemma, proved in Appendix A, the low-rank approximation error of $M$ in the $\| \cdot \|_{2,\infty}$ or $\| \cdot \|_{\infty,\infty}$ norm is controlled by $\xi(M)$.

**Lemma 1.** *Let $M \in \mathbb{R}^{n \times n}$. We have: for any $1 \le r < n$, $\|M - [M]_r\|_{2,\infty} \le \sqrt{\sigma_{r+1}\xi(M)}$.*

This lemma serves as an analogue, under the $\| \cdot \|_{2,\infty}$ norm, of the "key lemma" from (11) (specifically, Lemma 3.5), which underpins a universal thresholding SVD procedure in the Frobenius norm setting.

In our context, the lemma implies that $\|M - [M]_r\|_{2,\infty} \le \varepsilon$ for the largest rank $r$ such that $\sigma_r \ge \epsilon^2/\xi(M)$. This provides a principled criterion for selecting the rank $r$ in a truncated SVD when targeting an accuracy level $\epsilon$ in the $\| \cdot \|_{2,\infty}$ norm. Additionally, for the problem of estimating the matrix from sample entries with $\| \cdot \|_{2,\infty}$ guarantees, we derive a sample complexity lower bound scaling as $\xi(M)$, see Appendix B.

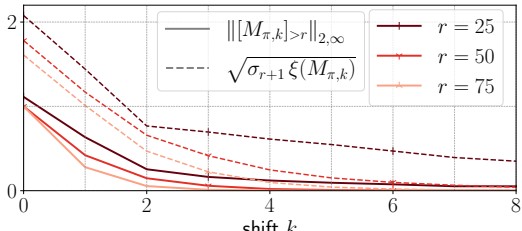

Figure 2: Approximation error as a function of the shift parameter $k$ and rank $r$. The theoretical upper bound serves as a first-order proxy for the entry-wise error. We use the standard $\| \cdot \|_{2\to\infty}$ norm, which matches (up to a $\sqrt{n}$ factor) the variant from Section 3.2 under the uniform measure $\nu$. See Section 6 for experimental details.

We conclude with a few remarks. $\xi(M)$ and $\|M\|_{2,\infty}$ are closely related as $\|M\|_{2,\infty}^2 = \max_x \sum_i \sigma_i^2 \psi_i(x)^2 \le \sigma_1 \xi(M)$. When $M$ has rank $r$, the spectral irrecoverability satisfies: $\xi(M) \le \sigma_1 \|U_r\|_{2,\infty}^2 = r\sigma_1 c((\psi_i)_{i=1}^r)$ which connects $\xi(M)$ to classical notions of coherence. Finally, as shown in Fig. 2, the low-rank approximation error of the shifted successor measure improves when the shift $k$ increases (see Section 5 for theoretical justifications).

## 4 Estimation of the Shifted Successor Measure

### 4.1 Main result

We assume access to a dataset of transitions $(s, a, s')$ collected offline. Let $Z_{s,a}$ denote the number of independent transitions observed from the state-action pair $(s, a)$. Our analysis provides estimation error bounds conditional on these counts, so we may treat the $Z_{s,a}$ as deterministic. Using the data, we form the empirical estimator $\widehat{P}(s, a, s') = Y_{s,a,s'}/Z_{s,a}$, and given a policy $\pi$ we can then also form $\widehat{P}_\pi((s,a),(s',a')) = \widehat{P}(s,a,s')\pi(s',a')$ as the empirical estimator of $P_\pi$. We can then build a simple estimator of the $k$-shifted successor measure $M_{\pi,k} = P_\pi^k(I - \gamma P_\pi)^{-1}$ by taking $\widehat{M}_{\pi,k} = \widehat{P}_\pi^k(I - \gamma \widehat{P}_\pi)^{-1}$.

Our final estimator of $M_{\pi,k}$ is obtained by computing the truncated $\nu$-SVD $[\widehat{M}_{\pi,k}]_r$ of $\widehat{M}_{\pi,k}$. We derive guarantees for this estimator under any probability measure $\nu$ of the following form. Let $\mu$ be a probability measure on $\mathcal{S}$; we define $\nu$ such that $\nu(s,a) = \mu(s)\pi(s,a)$ for all $(s,a)$. In the following theorem, $\sigma_i$ denotes the $i$-th singular value of $\widehat{M}_{\pi,k}$ in the $\nu$-SVD, and $\nu_{\pi,\text{inv}}$ denotes the invariant measure of the Markov chain $P_\pi$. We also define for $\delta \in (0, 1)$:

$$\Gamma_\delta := \max\left(k, (1-\gamma)^{-1}\right)^2 \sqrt{\max_{(s,a),(s',a')\in\mathcal{X}} \frac{\nu(s,a)}{Z_{s,a}\nu(s',a')} \log(rn/\delta)}, \tag{1}$$

$$\mathcal{E}_{\text{estim}} := \frac{\sigma_1 \max\left(\|M_{\pi,k}\|_{2,\infty}, \left\|M_{\pi,k}^\dagger\right\|_{2,\infty}\right)}{\sigma_r(\sigma_r - \sigma_{r+1})} \left\|\frac{d\nu}{d\nu_{\pi,\text{inv}}}\right\|_\infty \left\|\frac{d\nu_{\pi,\text{inv}}}{d\nu}\right\|_\infty \Gamma_\delta, \tag{2}$$

$$\mathcal{E}_{\text{approx}} := \sqrt{\sigma_{r+1}\xi(M_{\pi,k})}. \tag{3}$$

**Theorem 1.** *There is a universal constant $C > 0$ such that for any $k \ge 0$, any probability measure $\nu$ on $\mathcal{X}$, any $1 \le r < n$, and all $\delta \in (0, 1)$, we have, if $\Gamma_\delta \le 1$, with probability at least $1 - \delta$,*

$$\left\|[\widehat{M}_{\pi,k}]_r - M_{\pi,k}\right\|_{2,\infty} \le C\mathcal{E}_{\text{estim}} + \mathcal{E}_{\text{approx}}. \tag{4}$$

In the proof presented in Appendix C, we show that $C\mathcal{E}_{\text{estim}}$ and $\mathcal{E}_{\text{approx}}$ are upper bounds on the estimation and approximation errors, respectively: $\|[\widehat{M}_{\pi,k}]_r - [M_{\pi,k}]_r\|_{2,\infty} \leq C\mathcal{E}_{\text{estim}}$ and $\|[M_{\pi,k}]_r - M_{\pi,k}\|_{2,\infty} \leq \mathcal{E}_{\text{approx}}$.

## 4.2 Discussion

We discuss the terms involved in the estimation error upper bound below.

(a) The term $A := \frac{\sigma_1 \max(\|M_{\pi,k}\|_{2,\infty}, \|M_{\pi,k}^{\dagger}\|_{2,\infty})}{\sigma_r(\sigma_r - \sigma_{r+1})}$ comes from the so-called leave-one-out analysis, a step in the proof that aims at going from error bounds in spectral norm to error bounds in $\|\cdot\|_{2,\infty}$. The numerator can be controlled via the spectral recoverability of $M_{\pi,k}$ since $\|M_{\pi,k}\|_{2,\infty} \leq \sigma_1 \xi(M_{\pi,k})$. For $A$ to be controlled, we hence need to control the spectral recoverability of $M_{\pi,k}$, to have $r$ such that $\sigma_1/\sigma_r$ is bounded and the gap $\sigma_r - \sigma_{r+1}$ is significant. In Appendix C, we discuss how to control $\sigma_r - \sigma_{r+1}$ in case of bounded spectral irrecoverability.

(b) The term $B := d(\nu, \nu_{\pi,\text{inv}}) := \left\|\frac{d\nu}{d\nu_{\pi,\text{inv}}}\right\|_{\infty} \left\|\frac{d\nu_{\pi,\text{inv}}}{d\nu}\right\|_{\infty}$ involves the Radon-Nikodym derivative of $\nu$ w.r.t. $\nu_{\pi,\text{inv}}$ and $\nu_{\pi,\text{inv}}$ w.r.t. $\nu$. It captures the discrepancy between $\nu$, used to compute the SVD, and the invariant measure $\nu_{\pi,\text{inv}}$ of the Markov chain under policy $\pi$. The choice of $\nu$ is under the control of the practitioner. In practice, it may correspond to the empirical distribution of the dataset or be chosen arbitrarily, for example, as the uniform distribution, in which case the SVD reduces to the standard SVD. On the other hand, the invariant distribution $\nu_{\pi,\text{inv}}$ is more naturally aligned with the dynamics and yields the tightest possible bound. Setting $\nu = \nu_{\pi,\text{inv}}$ eliminates the multiplicative factor $B$, resulting in the best-case guarantee. However, estimating $\nu_{\pi,\text{inv}}$ exactly may not necessarily be feasible. Theorem 1 accommodates potential mismatch between $\nu$ and $\nu_{\pi,\text{inv}}$, showing that it is sufficient for $\nu$ to approximate the invariant measure up to a constant factor.

(c) The term $C := \max\left(k, (1-\gamma)^{-1}\right)^2$ comes from extending the concentration results in spectral norm of $\widehat{P}$ to the shifted successor measure $\widehat{M}_{\pi,k}$. The form of this term critically relies on a comparison of $\nu$ with the invariant measure, allowing us to exploit contraction properties and avoid exponential dependence in $k$ or $(1-\gamma)^{-1}$.

(d) The term $D := \max_{(s,a),(s',a') \in \mathcal{X}} \frac{\nu(s,a)}{Z_{s,a}\nu(s',a')} \log(rn/\delta)$ can eventually be traced back to the concentration in spectral norm of the empirical estimator $\widehat{P}$, and is the only term that depends on the number of observations: if we want $\xi$ small this factor shows how large each $Z_{s,a}$ should. Because of the ratio $\frac{\nu(s,a)}{\nu(s',a')}$, the result applies primarily to the case where $\nu$ exhibits some kind of homogeneity.

**Corollary 1.** *Assume that $\xi(M_{\pi,k})$, $\sigma_1/\sigma_r$, $\max_{(s,a),(s',a')} \frac{\nu(s,a)}{\nu(s',a')}$ and $d(\nu, \nu_{\pi,\text{inv}})$ are $\mathcal{O}(1)$, and that $\sigma_r - \sigma_{r+1} = \Omega(1)$. Then a sufficient condition for $\left\|[\widehat{M}_{\pi,k}]_r - [M_{\pi,k}]_r\right\|_{2,\infty} = \mathcal{O}(\varepsilon)$ with probability at least $1 - \delta$ is that the number of observations per state-action pair satisfies $\min_{(s,a)} Z_{s,a} = \Theta\left(\frac{\log(nr/\delta)}{\varepsilon^2}\right)$.*

From the above result, we deduce that under the structural assumptions made on $M_{\pi,k}$, the sample complexity to obtain an estimation error scaling as $\varepsilon$ in the $\|\cdot\|_{2,\infty}$ norm scales as $n/\varepsilon^2$ up to the logarithmic term. Without structure, this sample complexity would necessarily scale as $n^2/\varepsilon^2$. We provide a more detailed discussion about these assumptions, including the role of the measure $\nu$, the rank, etc. in Appendix C.

## 5   When Low-rank Structure Emerges: Local Mixing Phenomena

There is no reason to expect the transition kernel $P$ (or $P_\pi$) to exhibit a low-rank structure, and in view of the following proposition (proved in Appendix G), the same observation holds for the successor measure.

**Proposition 1.** *Let $P_\pi \in \mathbb{R}^{n \times n}$. For all $\gamma \in (0,1), k \geq 0, i \in [n]$, $\frac{\sigma_i(P_\pi^k)}{1+\gamma} \leq \sigma_i(M_{\pi,k}) \leq \frac{\sigma_i(P_\pi^k)}{1-\gamma}$. Consequently $\|M_\pi\|_{2,\infty} \geq \frac{\sqrt{n}}{1+\gamma}$ and $\|M_{\pi,k}\|_{2,\infty} \geq \frac{\|P_\pi^k\|_F}{1+\gamma}$.*

However, the situation changes when we consider powers of the transition matrix: for some $k > 1$, the matrix $P_\pi^k$ may become approximately low-rank. Specifically, if the Markov chain is ergodic, then $P_\pi^k$ approaches a rank-1 matrix as $k$ nears the mixing time. This observation suggests that the $k$-shifted successor measure $M_{\pi,k}$ may also exhibit low-rank structure for high values of $k$. However, the mixing time can be prohibitively long, and applying such a large shift would be impractical. This raises a natural question: can a low-rank structure emerge at smaller values of $k$, before the chain has fully mixed? We address this question by developing theoretical tools to determine from which value of $k$ the $\|\cdot\|_{2,\infty}$ norm and the spectral irrecoverability of $M_{\pi,k}$ become bounded. We relate this threshold to a concept we refer to as *local mixing* of the underlying Markov chain.

For notational convenience, throughout this section, we write $P$ (resp. $M_k$) in place of $P_\pi$ (resp. $M_{\pi,k}$). We also define $\nu_{\min} := \min_{x \in \mathcal{X}} \nu(x) > 0$. We observe that $\|M_k\|_{2,\infty} \leq \|M\|_{\infty,\infty} \|P^k\|_{2,\infty} = (1-\gamma)^{-1} \|P^k\|_{2,\infty}$, and hence in what follows we restrict our attention to upper bounding $\|P^k\|_{2,\infty}$. We discuss how to perform a similar analysis for $\|M_k^\dagger\|_{2,\infty}$ and $\xi(M_k)$ in Appendix G.

## 5.1 Local mixing estimates via Poincaré inequalities

To estimate the smallest value of $k$ for which $\|P^k\|_{2,\infty}$ becomes bounded, we develop and leverage functional inequalities inspired by those used to analyze the mixing times of Markov chains (see, e.g., (49; 54)). Appendix G provides a detailed introduction to these techniques, as well as the proofs of all the results of this section. We introduce the Dirichlet form $\mathcal{E}_{PP^\dagger}(f,g) = \langle (I - PP^\dagger)f, g \rangle_\nu$ for all $f, g \in \mathbb{R}^n$. The next theorem shows that deriving functional inequalities on the Dirichlet form allows us to control $\|P^k\|_{2,\infty}$.

**Theorem 2.** *Suppose there exist $\lambda, C \geq 0$ such that $P$ satisfies the type II[2] Poincaré inequality*

$$\forall f \in \mathbb{R}^n : \quad \lambda \|f\|_2^2 \leq \mathcal{E}_{PP^\dagger}(f,f) + C\lambda \|f\|_1^2. \tag{5}$$

*Then for all $k \geq 0$: $\left\|P^k\right\|_{2,\infty}^2 \leq \left(\nu_{\min}^{-1} - C\right)(1-\lambda)^k + C$.*

When $\nu$ is the invariant measure of $P$, the Courant-Fischer theorem (Theorem 3.1.2 in (27)) yields (5) with $\lambda = 1 - \sigma_2(P)^2$ and $C = 1$, which in turn leads to a bound on the mixing time that depends on the singular gap of $P$. However, as we show below, type II inequalities can also be derived using higher-order singular values of $P$. This leads to significantly faster exponential decay rates for $\|P^k\|_{2,\infty}^2$, albeit at the cost of a larger limiting constant $C$. Interestingly, our analysis reveals a connection between this limiting constant and the coherence of the singular vectors.

**Theorem 3.** *Suppose the underlying measure $\nu$ is the invariant measure of $P$. Let $P = U\Sigma V^\dagger$ be the SVD of $P$, and $\sigma_1 \geq \ldots \geq \sigma_n \geq 0$ be the corresponding to singular values.*
*(a) For all $r \in [n]$, for all function $f \in \mathbb{R}^n$, we have*

$$\frac{1 - \sigma_{r+1}^2}{2} \|f\|_2^2 \leq \mathcal{E}_{PP^\dagger}(f,f) + (1 - \sigma_{r+1}^2) \|U_r\|_{2,\infty}^2 \|f\|_1^2. \tag{6}$$

*(b) For all $r \in [n]$, the result of Theorem 2 holds with $\lambda = (1 - \sigma_{r+1}^2)/2$ and $C = 2\|U_r\|_{2,\infty}^2$.*

As a consequence, if the coherence $r^{-1}\|U_r\|_{2,\infty}^2$ of the $r$ first left singular vectors of $P$ is known to be bounded by $C/2$ (independent of $n$), then we can suggest to apply a shift $k \approx \log(Cr\nu_{\min})/\log((1 + \sigma_{r+1}^2)/2)$ to ensure that $\|P_k\|_{2,\infty} = \mathcal{O}(1)$ and that $M_k$ can be estimated efficiently by using a low-rank approximation. Such a shift $k$ is typically much smaller than the mixing time (when $\sigma_2$ is close to 1 while $\sigma_{r+1}$ remains bounded away from it). Note however that the singular values and the coherence of the singular vectors of $P$ may be unknown in practice. In such cases, we propose an alternative method to study the decay rate of $\|P^k\|_{2,\infty}$.

## 5.2 Decomposable Markov chains

Another strategy to analyze the decay rate of $\|P^k\|_{2,\infty}$ is to study *local mixing* behavior of the Markov chain via type II Poincaré inequalities, and combine these inequalities to derive a *global* type II Poincaré inequality. We formalize this idea as follows.

---

[2]This terminology is inspired by (54, Chapter 2), where the author distinguishes two variants of Nash's argument, the second giving no direct bounds on mixing times (see Theorem 2.3.4).

**Definition 4** (Induced Markov chain). Given a Markov chain on $[n]$ with transition matrix $P$ and a subset $S \subseteq [n]$, the induced chain on $S$ is the Markov chain on $S$ with transition matrix $P_S$ given by

$$\forall x, y \in S : P_S(x,y) := \begin{cases} P(x,y) & \text{if } y \neq x, \\ P(x,x) + \sum_{z \notin S} P(x,z) & \text{if } y = x. \end{cases}$$

The induced measure $\nu_S$ is the measure on $S$ given by $\nu_S(x) := \nu(x)/\nu(S)$ for all $x \in S$. We also denote by $\mathcal{E}_{\nu_S, (PP^\dagger)_S} := \mathcal{E}_{(PP^\dagger)_S}$ the Dirichlet form constructed with the scalar product $\langle \cdot, \cdot \rangle_{\nu_S}$.

**Proposition 2.** *Let $P$ be Markov chain on $[n]$ with invariant measure $\nu$ and $S \subset [n]$. Suppose the induced chains $(PP^\dagger)_S, (PP^\dagger)_{S^c}$ both satisfy a type II Poincaré inequality with respect to the induced measure: for $B \in \{S, S^c\}$,*

$$\forall f \in \mathbb{R}^B, \quad \lambda_B \|f\|^2_{\ell^2(\nu_B)} \leq \mathcal{E}_{\nu_B, (PP^\dagger)_B}(f,f) + \lambda_B C_B \|f\|^2_{\ell^1(\nu_B)},$$

*Then $P$ satisfies: $\forall f \in \mathbb{R}^n$, $\lambda \|f\|^2_{\ell^2(\nu)} \leq \mathcal{E}_{\nu, PP^\dagger}(f,f) + \lambda C \|f\|^2_{\ell^1(\nu)}$ with $\lambda = \min(\lambda_S, \lambda_{S^c})$ and $C = \max\left(\frac{C_S}{\nu(S)}, \frac{C_{S^c}}{\nu(S^c)}\right)$.*

This result shows how to combine local type II Poincaré inequalities. It can be applied inductively to consider more complex partitions of the state space, i.e., with more than two subsets. When comparing to Theorem 3, we note $\max_i \nu(S_i)^{-1}$ plays here a role analogous to the coherence. Proposition 2 is very general, and we illustrate its application through the following simple example.

**The 4-room environment.** Consider a Markov chain whose transition graph $G$ can be partitioned into 4 rooms or connected subgraphs $(G_i)_{i \in [4]}$, as shown in Fig. 3. $G$ is obtained by adding an edge between each pair $(G_i, G_{i+1})$. Consider the simple random walk on $G$ (at each step moves to a neighbor in $G$ uniformly at random). It is an irreducible reversible Markov chain with transition matrix $P$ and stationary distribution $\nu$. The chain induced by $P^2$ on $G_i$ is also reversible with spectral gap $\lambda_i$. The latter allows us to upper bound the (local) mixing time of the chain on $G_i$ as $\lambda_i^{-1} \log(\nu_{\min})^{-1}$. Proposition 2 yields an explicit bound on $\|P^k\|_{2,\infty}$ which in turn, thanks to reversibility, leads to a lower bound of the spectral recoverability of $P^k$. In summary, we can state the following result, proved in Appendix G.

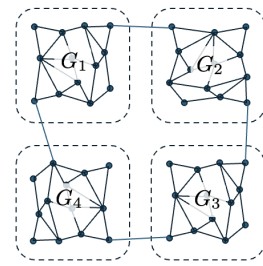

Figure 3: The four-room environment.

**Theorem 4.** *For all $k \geq 0$, we have:*
$$\|P^k\|^2_{2,\infty} \leq (\min_{i \in [4]} \nu(G_i))^{-1} + (1 - \min_{i \in [4]} \lambda_i)^k \nu_{\min}^{-1}.$$
*Furthermore, suppose that $\min_{i \in [4]} \nu(G_i) \geq c$ for some constant $c > 0$. Then for all $\varepsilon \in (0,1)$, for all $k \geq 2 \max_{i \in [4]} \lambda_i^{-1} \log(\nu_{\min}^{-1} \varepsilon^{-1} \sqrt{2/c})$, $\left\| P^k - [P^k]_4 \right\|_{2,\infty} \leq \varepsilon$.*

The above theorem illustrates how we can decompose a Markov chain into sub-chains so as to understand the shift needed to estimate the matrix efficiently using a low-rank matrix. Assume for example that the graphs $G_i$ are bounded-degree expanders (26). Then we have $\lambda_i^{-1} = O(1)$ and $\nu$ is uniform up to a $\Theta(1)$ factor. The required shift, $\log(n)$, is much smaller than the mixing time of the chain on $G$, scaling as $n \log(n)$. We give further details and examples in Appendix G.

## 6 Numerical Experiments[3]

We now turn to empirical validation of our theoretical findings. In the previous sections, we analyzed how shifting affects the estimation of successor measures and the emergence of low-rank structure. Here, we test the hypothesis that these structural changes translate into tangible differences in learned behavior. Since accurate successor measures yield uniformly accurate Q-value estimates, we expect the impact of shifting to be reflected not only in the estimated Q-values, but more importantly, in the resulting policies. One domain where the practical relevance of successor measures can be directly examined is goal-conditioned reinforcement learning (GCRL) (3; 9; 21; 63), where the objective is to learn policies $\pi_g(a|s)$ that reach arbitrary goal states $g \in \mathcal{S}$. This setting provides a natural testbed for our analysis, as the quality of estimated successor measures directly determines the accuracy of goal-conditioned value estimates and, consequently, the learned policies.

---

[3]Code available at `https://github.com/stestoKTH/shift-SM`

Following (63) consider the goal-specific reward function $R_g(s,a) = P(s' = g|s,a)$. Recall that $M_{\pi,k} = P_\pi^k(I - \gamma P_\pi)^{-1}$, and thus $P(I - \gamma P_\pi)^{-1} = \sum_b \pi(b|\cdot)P(I - \gamma P_\pi)^{-1} = \sum_b M_{\pi,k=1}(\cdot,\cdot,\cdot,b)$. As shown in Proposition 1 of (21), the corresponding state-action value function can be written as the marginalized successor measure: $Q^{(R_g,\pi_g)}(s,a) = \sum_b M_{\pi_g,k=1}(s,a,g,b)$. This implies that the optimal policy is obtained by acting greedily with respect to $\sum_b M_{\pi_g,k=1}(s,a,g,b)$. Our experiments follow the setup of (20; 21), where the critic learns $Q^{(R_g,\pi_{\mathcal{D}})}$ for a goal-marginalized policy $\pi_{\mathcal{D}}(a|s) = \int_{\mathcal{S}} \pi_g(a|s)d\rho_{\mathcal{D}}(g)$, with $\rho_{\mathcal{D}}(g)$ denoting the empirical distribution of goals in the dataset $\mathcal{D}$. This setup reflects a common GCRL scenario, where the agent reuses past experience collected under different goals to improve sample efficiency.

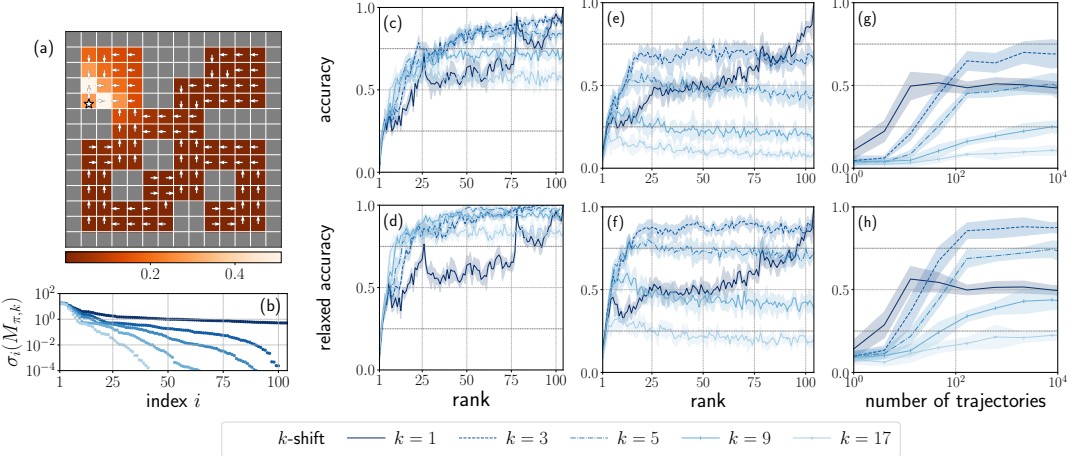

Figure 4: (a) Discrete *Medium Pointmaze* environment. Each state $s$ is colored by $\max_a \sum_{b \in \mathcal{A}} M_{\pi_{\mathcal{D}},k=1}(s,a,g,b)$, with $\gamma = 0.95$, goal $g$ marked by a star, and actions follow a uniform policy $\pi_{\mathcal{D}}$. Arrows indicate the greedy policy $\pi(s|g) = \text{argmax}_a \sum_b M_{\pi_{\mathcal{D}},k=1}(s,a,g,b)$. (b) Singular values of shifted successor measures. (c–d) Accuracy (probability of reaching a random goal) and relaxed accuracy (reaching its 2-neighborhood) as a function of rank and shift for true successor measures. (e–f) Same as (c–d), but for successor measures learned via TD. (g–h) Accuracy vs. number of trajectories of length $H = 100$. Results are averaged over 100 random goals and 5 seeds.

We explore how low-rank approximation (via truncated SVD) and temporal shifting of successor measures affect the performance of goal-conditioned policies. We perform experiments in the *Medium PointMaze* environment with 104 discrete states and 4 actions (see Figure 4 (a)). Additional numerical experiments are provided in Appendix H. In Figure 4 (b), we observe that shifting successor measures sharpens the spectrum, accelerating singular value decay. To quantify goal-reaching performance, we report:

(upper row) accuracy, the probability of reaching the exact goal from a random initial state, and
(lower row) relaxed accuracy, the probability of reaching any state within two steps of the goal.

The relaxed accuracy reflects that, in many scenarios, reaching a nearby state is practically sufficient. For all evaluations, the policy acts greedily with respect to the corresponding successor measure matrix. Figure 4 (c) shows that even when using an oracle successor measure, introducing a temporal shift improves performance, especially when combined with low-rank approximation. This benefit is particularly notable when success is defined more flexibly, as shown in Figure 4 (d). These results suggest that shifting enhances the expressiveness of successor measures while compensating for rank constraints.

To estimate successor measures from data, we apply Temporal Difference (TD)-learning with TD-errors $\mathbb{1}[s_{t+k+1} = g, a_{t+k+1} = b] + \gamma \widehat{M}_{\pi_{\mathcal{D}},k}(s_{t+k+1},a_{t+k+1},g,b) - \widehat{M}_{\pi_{\mathcal{D}},k}(s_t,a_t,g,b)$, where $(g,b) \in \mathcal{S} \times \mathcal{A}$ and $(s_t,a_t,s_{t+k+1},a_{t+k+1})$ are sampled from $\mathcal{D}$. As shown in Figure 4 (e-f), larger shifts degrade performance when successor measures are learned via TD. This aligns with the intuition that estimating long-horizon dynamics is harder and introduces more error, particularly in low-data regimes. Finally, we assess how data efficiency depends on the shift parameter by fixing the rank to $r = 40$ and varying the number of samples in Figure 4 (g-h). We find that a moderate shift ($k = 3$)

consistently yields the best performance, suggesting a trade-off: while shifting improves expressivity, its estimation must remain tractable. This is also illustrated in Fig. 1 in §1.

**The choice of rank and shift parameters.** As shown in our results, the performance of policies derived from low-rank approximations improves substantially even for small rank values, consistent with prior findings (67; 57). In practice, we recommend selecting a rank much smaller than the state-space dimension. Note, however, that the optimal rank often depends on the chosen shift value, and the two parameters should thus be tuned jointly. Prior work - for instance, HIQL (51) - already treats the number of steps to a subgoal as an environment-specific hyperparameter.

## 7 Limitations and Future Work

Our work leaves open many questions, especially on the algorithmic implications of our theoretical findings. We highlight below the main limitations of this paper.

**Downstream optimization of policies.** Our main result provides guarantees for estimating shifted successor measures under a fixed policy, effectively performing reward-free policy evaluation. However the effective benefit for downstream policy optimization, once a reward is given, remains unclear and is not addressed in this paper. In line with prior studies (63; 37), our numerical experiments in Section 6 show that considering policies that are greedy w.r.t. Q-functions estimated under uniform or exploratory policies can perform well in practice. This motivates our focus on the evaluation problem, leaving the theoretical and practical understanding of such greedy policies, why and when they work, as an open direction for future research.

**Dependence on a generative model.** Theorem 1 makes the strong assumption of access to an i.i.d. dataset of transitions. This assumption effectively sidesteps the challenges of exploration and the use of sampled trajectories, which we leave as an important direction for future work.

**Extension to continuous settings.** Our work is restricted to tabular MDPs for simplicity. However, most ideas extend naturally to continuous spaces by replacing matrices with linear operators and measures (6; 63). Extending Theorem 1 under suitable smoothness assumptions, following (57; 59; 65), is a promising direction.

**Limitations of the experimental results.** Shifting removes local information, and for tasks such as goal reaching, where rewards are sparse and given only at the end, this has little impact on achieving optimal performance. However, in more general settings, we expect that combining shifted successor measures with estimates of local transitions could further improve performance. Our numerical results illustrate the theoretical findings and consider low-rank approximations using SVD. It remains unclear whether alternative low-rank approximation methods would exhibit different behavior. In Appendix H.5, we discuss potential extensions to non-tabular settings. An open question is how much of this phenomenon carries over to function approximation settings and how it can be leveraged effectively there.

## 8 Conclusion

In this work, we considered the problem of estimating shifted successor measures. Our main result established an upper bound on the sample complexity for a simple estimator based on SVD truncation. Unlike previous work, we make no structural assumption on the matrix, showing that structure would generally emerge naturally from local mixing phenomena. This led us to introduce shifted successor measures, to better distinguish between small-range transitions, which remain inherently high-rank, and long-range transitions where mixing phenomena take place and give rise to an approximately low-rank structure. This was empirically confirmed. Our experiments show that shifted successor measures are better approximated by their low-rank SVDs than the non-shifted counterpart, and that the use of shifts can bring performance improvements in (goal-conditioned) RL. These two main contributions open up many possibilities. From a theoretical perspective, we believe that our approach could be used to assess the sample complexity of estimating universal representations like the Forward-Backward model of (62). On the more practical side, the idea of shifting surely requires a more complete empirical analysis to better understand its impact across diverse RL settings.

## Acknowledgments

This research was supported by the Wallenberg AI, Autonomous Systems and Software Program (WASP) funded by the Knut and Alice Wallenberg Foundation, the Swedish Research Council (VR), and Digital Futures.

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

# Contents

# A  Measure-induced Norms and SVDs, Shifted Successor Measures and Spectral Recoverability

In this appendix we establish the formalism that is used throughout the paper. We start with general notation.

**Notation**   Given integers $m \leq n$, we write $[m, n] =: \{m, \ldots, n\}$ and in particular $[n] =: \{1, \ldots, n\}$. We write $n \wedge m := \min(n, w)$. Vectors are seen as column vectors, measures are identified with row vectors. We write $\mathbb{1}$ for the all-one vector, $\mathbb{1}_i$ the the indicator vector at $i$. Thus we write $\mathbb{1}_i \mathbb{1}_j^\top$ for the matrix with only one non-zero entry at coordinate $(i, j)$, equal to 1. We use the notation $\preceq$ for positive semi-definite inequalities, abbreviated as p.s.d.. We use the usual Landau notations $\mathcal{O}(\cdot), \Theta(\cdot)$, etc. for asymptotic analysis.

## A.1  Measure-induced norms and SVDs

**Norms with respect to a measure**   Given a measure $\mu$ on $[n]$, let $\langle f, g \rangle_\mu := \sum_{i \in [n]} \mu(i) f(i) g(i)$. Note that up to a factor $n$, the usual inner product is recovered by taking the uniform measure for $\mu$. Given $p \in [1, \infty]$, we define the $\ell^p$ norm

$$\|f\|_{\ell^p(\mu)} := \begin{cases} \left( \sum_{x \in \mathcal{X}} \mu(x) |f(x)|^p \right)^{1/p} & \text{if } p < \infty \\ \max_{x \in \mathcal{X}} |f(x)| & \text{if } p = \infty \end{cases} .$$

For simplicity we may keep the measure implicit and write only $\|f\|_p = \|f\|_{\ell^p(\mu)}$. We employ the term "norm" although $\|\cdot\|_p$ define norms only if $\mu$ has full support. Generally speaking, a lot of notions considered in the sequel may not be properly defined if $\mu$ does not have full support. Rather than always requiring the measure to have full support, we take the convention that the results become trivial when an object is ill-defined because of the norm. In particular, we define $\mu_{\min} := \min_i \mu(i)$ and consider that $\mu_{\min}^{-1} = +\infty$ if $\mu$ does not have full support.

**Adjoint operator and SVD**   When considering rectangular $A \in \mathbb{R}^{n \times m}$ we need to define two underlying measures $\mu, \nu$ on $[m]$ and $[n]$. If $n = m$, we always take $\mu = \nu$. The adjoint operator $A^\dagger \in \mathbb{R}^{m \times n}$ is the unique operator that satisfies $\langle Af, g \rangle_\nu = \langle f, A^\dagger g \rangle_\mu$ for all $f \in \mathbb{R}^m, g \in \mathbb{R}^n$. It is explicitely given by

$$A^\dagger(i, j) := \frac{\nu(i) A(i, j)}{\mu(j)}. \tag{7}$$

If $n = m$ and $\mu = \nu$ is uniform, $A^\dagger = A^\top$ is nothing but the transpose of $A$. Thus every notion that could normally be defined with a transpose will be here considered with the adjoint instead.

This applies in particular to the singular value decomposition (SVD). The left singular vectors $(\psi_i)_{i=1}^{n \wedge m}$, resp. right singular vectors $(\phi_i)_{i=1}^{n \wedge m}$ of $A \in \mathbb{R}^{n \times m}$ are defined as the eigenvectors of the self-adjoint matrix $AA^\dagger$, resp. $A^\dagger A$, corresponding to singular values $\sigma_1 \geq \ldots \geq \sigma_{n \wedge m} \geq 0$ which we always assume to be in non-increasing order. In matrix form, the SVD writes $A = U \Sigma V^\dagger$ where $\Sigma = \mathrm{Diag}((\sigma_i)_{i=1}^{n \wedge m})$ while $U \in \mathbb{R}^{n \times n}, V \in \mathbb{R}^{m \times m}$ are unitary in the sense $U^\dagger U = U U^\dagger = I$ and $V^\dagger V = V V^\dagger = I$. This implies that $U(x, i) := \sqrt{\mu(i)} \psi_i(x), V(x, i) := \sqrt{\mu(i)} \phi_i(x)$ for all $i, x \in [n]$, so the $i$-th column of $U, V$ does not exactly contain the entries of $i$-th singular vector. Given $r \in [n \wedge m]$, we write $[A]_r = U_r \Sigma_r V_r^\dagger$ for the SVD truncated to rank $r$ and $[A]_{>r} = A - [A]_r = U_{>r} \Sigma_{>r} U_{>r}^\dagger$.

**Norm of a row vector**   If $f \in \mathbb{R}^n$ is seen as a column vector, we define the row vector $f^\dagger$ by $f^\dagger(i) := f(i) \mu(i)$. This allows to have $\langle f, g \rangle_\mu = f^\dagger g$. Conversely for a row vector $\rho$ we define $\rho^\dagger$ as a column vector by $\rho^\dagger(i) := \rho(i) / \mu(i)$. We then define $\ell^p(\mu)$ norms of row vectors by the fact that $\|f^\dagger\|_{\ell^p(\mu)} = \|f\|_{\ell^{p^\dagger}(\mu)}$ where $p^\dagger$ is the Hölder conjugate of $p$, defined by $\frac{1}{p} + \frac{1}{p^\dagger} = 1$. In particular note that for indicator vectors,

$$\|\mathbb{1}_i\|_{\ell^2(\mu)} = \sqrt{\mu(i)}, \qquad \|\mathbb{1}_i^\top\|_{\ell^2(\mu)} = \frac{1}{\sqrt{\mu(i)}}. \tag{8}$$

**Matrix norms**  Given a matrix $A \in \mathbb{R}^{n \times m}$ and $p, q \in [1, \infty]$, we define the operator norm

$$\|A\|_{\ell^p(\mu), \ell^q(\nu)} := \sup_{\substack{f \in \mathbb{R}^n \\ f \neq 0}} \frac{\|Af\|_{\ell^q(\nu)}}{\|f\|_{\ell^p(\mu)}}.$$

Since the $\ell^\infty(\mu)$ norm does not depend on a measure, we will write more simply $\ell^\infty$. As for vectors, we may also write more simply $\|A\|_{p,q}$ when the underlying measures are clear. The $\|\cdot\|_{2,2}$ norm will also be called spectral norm. Our definition of row vector norms made to ensure the following property: if $\rho$ is a row vector then we can also upper bound $\|\rho A\|_{p,q} \leq \|\rho\|_q \|A\|_{p,q}$. For later use, we also recall the standard fact that

$$\|A\|_{p,q} := \|A^\dagger\|_{q^\dagger, p^\dagger}. \tag{9}$$

which is a consequence of Hölder's inequality.

In the sequel we will be specifically interested in the following norms, that can be distinguished in two categories:

1. unitarily invariant norms, including the spectral, nuclear and Frobenius norm, which are respectively the $\ell^\infty, \ell^1, \ell^2$ norms of singular values:

$$\|A\|_{2,2} = \sigma_1, \quad \|A\|_* = \sum_{i=1}^n \sigma_i, \quad \|A\|_F = \operatorname{tr}(A^\dagger A)^{1/2} = \left(\sum_{i=1}^n \sigma_i^2\right)^{1/2}. \tag{10}$$

2. "entrywise" norms:

$$\|A\|_{\infty,\infty} = \max_{i \in [n]} \sum_{j \in [m]} |A(i,j)|, \quad \|A\|_{2,\infty} = \max_{i \in [n]} \left(\sum_{j \in [m]} \frac{|A(i,j)|^2}{\mu(j)}\right)^{1/2}. \tag{11}$$

Unlike unitarily invariant norms, these depend on singular vectors: for the two-to-infinity we can make the dependence explicit in the left singular vectors: it is easily checked that

$$\|A\|_{2,\infty}^2 = \max_{x \in [n]} \sum_{i=1}^n \sigma_k^2 \psi_i(x)^2 \tag{12}$$

By duality (9), $\|A\|_{1,2}^2 = \|A^\dagger\|_{2,\infty}^2 = \max_{j \in [j]} \sum_{k=1}^{n \wedge m} \sigma_k^2 \phi_k(j)^2$. Note the inequalities

$$\|\cdot\|_? \leq \|\cdot\|_{2,\infty} \leq \nu_{\min}^{-1/2} \|\cdot\|_? \tag{13}$$

for all $\|\cdot\|_? \in \{\|\cdot\|_{2,2}, \|\cdot\|_F, \|\cdot\|_{\infty,\infty}\}$, as well as the submultiplicative inequalities

$$\|AB\|_{2,\infty} \leq \|A\|_{\infty,\infty} \|B\|_{2,\infty}, \quad \|AB\|_{2,\infty} \leq \|A\|_{2,\infty} \|B\|_{2,2}.$$

**Stochastic matrices and invariant measures**  We will often use an arbitrary measure, but in the context of finite Markov chains invariant measures are the most natural choices. On top of giving a probabilistic meaning and making a link with mixing as argued in Section 5, we will mostly interested in invariant measures to obtain contraction properties. Given a stochastic matrix $P \in \mathbb{R}^{n \times m}$, it is readily seen from (11) that $\|P\|_{\infty,\infty} = 1$. On the other hand

$$\|P^\dagger\|_{\infty,\infty} = \|P\|_{1,1} = \max_{j \in [m]} \frac{\sum_{i \in [n]} \nu(i) P(i,j)}{\mu(j)}.$$

hence $\|P^\dagger\|_{\infty,\infty} \leq 1$ if and only if $\nu P \leq \mu$ pointwise. If $n = m$, this forces $\mu$ to be an *invariant measure*. The Riesz-Thorin interpolation theorem then implies that $\|P\|_{p,p} \leq 1$ for all $p \in [1, \infty]$. In particular this implies that the spectral norm $\|P\|_{2,2} = \sigma_1 = 1$ (corresponding to the all-one eigenvector and singular vector) and all singular values are bounded by $1$.

## A.2  Spectral recoverability: Proof of Lemma 1

*Proof of Lemma 1.* From (12) and the definition of the spectral irrecoverability we immediately see that

$$\|M - [M]_r\|_{2,\infty}^2 = \sum_{i \geq r+1} \sigma_i^2 \psi_i(x)^2 \leq \sigma_{r+1} \xi(M).$$

$\square$

# B   Sample Complexity Lower bounds for $\|\cdot\|_{2,\infty}$ Guarantees

In this appendix, we provide a minimax lower bound on the estimation error of (non-shifted) successor measures under a generative model, i.e., when observing independent transitions of the Markov chain.

**Definition 5.** Let $\mathcal{P}$ a subset of stochastic matrices of size $n \times m$. Given $P \in \mathcal{P}$ and a vector $Z \in \mathbb{N}^n$ with non-negative integer entries, consider a family $(x_t, y_t)_{t=1}^T \in ([n] \times [m])^T$ obtained by sampling $Z_i$ transitions under $P(i, \cdot)$ for each $i \in [n]$, independently of all other transitions. We write $\mathbb{P}_P$ for the law of $(x_t, y_t)_{t=1}^T$. Given a map $f : \mathcal{P} \to \mathbb{R}^d$, a norm $\|\cdot\|$ on $\mathbb{R}^d$, $\varepsilon > 0$ and $\delta \in [0, 1]$, an estimator $\widehat{M}$ of $f(P)$ is said to be $(\varepsilon, \delta)$-PAC with for $\mathcal{P}$ and the norm $\|\cdot\|$ if for all stochastic matrix $P \in \mathcal{P}$, $\mathbb{P}_P \left[ \left\| \widehat{M} - f(P) \right\| > \varepsilon \right] \le \delta$.

The following proposition shows the sample complexity of estimating of the successor measure is essentially the same as that of estimating the transition matrix itself.

**Proposition 3.** *Let $P \in \mathbb{R}^{n \times n}$ be a stochastic matrix and $\gamma, \varepsilon \in [0, 1)$. Suppose $\widehat{M}$ is a $(\varepsilon, \delta)$-PAC estimator of $M = (I - \gamma P)^{-1}$ for the norm $\|\cdot\|_{\infty,\infty}$. Then $\widehat{P} := \frac{1}{\gamma}(I - \widehat{M}^{-1})$ is a $(4\varepsilon/\gamma, \delta)$-PAC estimator of $P$ for the $\|\cdot\|_{\infty,\infty}$ norm.*

*Proof.* Suppose $\left\| \widehat{M} - M \right\|_{\infty,\infty} \le \varepsilon$. First we show $\widehat{M}$ almost satisfies the Bellman equation: using that $M = I + \gamma P M$

$$\left\| \widehat{M} - (I + \gamma P \widehat{M}) \right\|_{\infty,\infty} = \left\| (I - \gamma P)(\widehat{M} - M) \right\|_{\infty,\infty}$$
$$\le \left\| (I - \gamma P) \right\|_{\infty,\infty} \left\| (\widehat{M} - M) \right\|_{\infty,\infty}$$
$$\le (1 + \gamma \|P\|_{\infty,\infty})\varepsilon$$
$$\le 2\varepsilon.$$

Then using $\gamma \widehat{P} = I - \widehat{M}^{-1}$

$$\left\| \gamma(\widehat{P} - P) \right\|_{\infty,\infty} = \left\| I - \widehat{M}^{-1} - \gamma P \right\|_{\infty,\infty}$$
$$= \left\| (\widehat{M} - I - \gamma P \widehat{M})\widehat{M}^{-1} \right\|_{\infty,\infty}$$
$$\le \left\| \widehat{M} - I - \gamma P \widehat{M} \right\|_{\infty,\infty} \left\| \widehat{M}^{-1} \right\|_{\infty,\infty}$$
$$\le 2\varepsilon \left\| I - \gamma \widehat{P} \right\|_{\infty,\infty}$$
$$\le 4\varepsilon.$$

$\square$

By the previous proposition, we are led to derive a lower bound on the sample complexity for estimating the transition matrix.

**Theorem 5.** *For all integer $n$ large enough, for all $\kappa \in [1, n]$, there exists a family $\mathcal{P}_\kappa$ of Markov chains on $[n]$ which satisfies:*

*(i) every $P \in \mathcal{P}_\kappa$ is reversible with uniform invariant measure,*

*(ii) for all $P \in \mathcal{P}_\kappa$, we have $\xi(P) \le \kappa$,*

*(iii) there exists a universal constant $C > 0$ such that for all $\varepsilon > 0$, if $\left( \sum_{x \in [n]} Z_x \right) \le C\varepsilon^{-2} \max(n, \kappa^2)$, then there exists no $(\varepsilon, \delta)$-PAC estimator for $\mathcal{P}_\kappa$ and the $\|\cdot\|_{\infty,\infty}$ norm.*

In (68, Theorem 2), the authors consider the problem of estimating a rank $r$ transition matrix from a trajectory and prove a minimax lower bound on the sample-complexity of order $rn/\varepsilon^2$. Our lower bound attempts to mimic this result by replacing the rank $r$ with the spectral irrecoverability, but only proves a lower bound of order $\max(n, \kappa^2)\varepsilon^{-2}$. Our class of examples is based on block Markov chains which allows to express the spectral irrecoverability as that of a smaller chain (Lemma 3. Intuitively, the sample-complexity of $\kappa^2$ is that of learning the smaller chain, while $n$ is the complexity required to learn the partition into blocks. To get a lower bound of order $\kappa n \varepsilon^{-2}$, we believe it is necessary to consider a soft partitioning of states, a.k.a state aggregation or mixed membership model as in (68). However we do not know how to extend the result of Lemma 3 to that case.

## B.1 Block Markov chains

Our class of examples consist of Block Markov chains similar to those considered in (56).

**Definition 6.** Consider a Markov chain on $[n]$ with transition matrix $P$. It is a block Markov chain with $k$ blocks if there exists a stochastic matrix $Q$ on $[k]$, a partition of $[n]$ into $k$ subsets $V_1, \ldots, V_k$ and a stochastic matrix $p \in \mathbb{R}^{k \times n}$ such that

$$\forall x, y \in [n]: \quad P(x,y) = Q(V(x), V(y))p(V(y), y) \tag{14}$$

where we write $V(x)$ for the subset of the partition containing $x$. Furthermore we require that $p(i,x) > 0$ implies $x \in V_i$, which implies that (14) writes matricially as $P = Qp$. We call $Q$ the inter-block matrix and $p$ the emission matrix.

**Lemma 2.** *Let $P$ be a block Markov chain with inter-block matrix $Q$ and emission matrix $p$. Then for all invariant measure $\mu$ of $Q$, $\mu p$ is invariant for $P$. Secondly, when these measures are taken as underlying the notions of adjoint, we have $pp^\dagger = I$ and*

$$P = p^\dagger Q p.$$

*Proof.* Suppose $\mu$ is invariant. Then we check that for all $y \in [n]$,

$$\begin{aligned}
\sum_{x \in [n]} \mu p(x) P(x,y) &= \sum_{i \in [k], x \in [n]} \mu(i) p(i,x) Q(V(x), V(y)) p(V(y), y) \\
&= \sum_{i \in [k], x \in [n]} \mu(i) p(i,x) Q(i, V(y)) p(V(y), y) \\
&= \sum_{i \in [k]} \mu(i) Q(i, V(y)) p(V(y), y) \\
&= \mu(V(y)) p(V(y), y) = \mu p(y).
\end{aligned}$$

The second and last line have used that $p(i,x) > 0$ implies $x \in V_i$. This is also crucial for the statement: computing the adjoint of $p$ we have

$$p^\dagger(x, i) = \frac{\mu(i) p(i,x)}{\mu p(x)} = \mathbb{1}_{x \in V_i}.$$

Thus $pp^\dagger = I$ and $P(x,y) = \sum_{i,j \in [k]} \mathbb{1}_{x \in V_i} Q(i,j) p(j,y) = p^\dagger Q p(x,y)$ for all $x, y \in [n]$. $\qquad \square$

Our interest for block Markov chains comes from the following.

**Lemma 3.** *Under the assumptions made in the previous lemma, $\xi(P) = \xi(Q)$. In particular $\xi(P) \leq \mu_{\min}^{-1}$.*

*Proof.* From the lemma, we can thus compute $P^\dagger = p^\dagger Q^\dagger p$ and

$$PP^\dagger = p^\dagger Q Q^\dagger p, \qquad P^\dagger P = p^\dagger Q^\dagger Q p.$$

In particular this shows that if $\phi$, resp. $\psi$ is a right, resp. left singular vector of $Q$ associated with singular value $\sigma$ then $p^\dagger \phi$, resp. $p^\dagger \psi$ is a right, resp. left singular vector of $P$ associated with singular value $\sigma$. Note also that $P$ is a rank $k$ matrix so all non-zero singular values are obtained this way. Thus from Definition 3

$$\xi(P) = \max_{x \in [n]} \sum_{i=1}^{k} \sigma_i (p^\dagger \psi_i(x))^2 = \max_{j \in [k]} \sum_{i=1}^{k} \sigma_i (\psi_i(j))^2 = \xi(Q).$$

$\square$

## B.2 Proof of Theorem 5

Our class of examples for the minimax lower bound are made of block Markov chain as described in the previous section. We consider in fact two different classes: for the first one we fix the block partition and the emission probabilities, and make vary the inter-block matrix, while for the second we fix the block partition and the inter-block matrix, and make vary the possible emission probabilities. We build these using a similar process as for the lower bound for reward-free RL of (31).

**Lemma 4.** *Consider an integer $n \geq 0$ and $\mathcal{A}_n := \{b \in \{-1, 0, 1\}^n : \sum_i b_i = 0\}$. If $n$ is sufficiently large there exists a subset $\mathcal{B}_n \subset \mathcal{A}_n$ such that*

*(i) $|\mathcal{B}_n| \geq e^{n/40}$,*

*(ii) for all $b \neq b' \in \mathcal{B}_n$, $\|b - b'\|_1 \geq \frac{n-1}{2}$.*

*Proof.* If $n$ is odd, we simply set the last entry of all $b$ to 0 (hence the $n - 1$ in (ii)). Therefore we suppose now $n$ is even and write $n = 2m$. We construct the set $\mathcal{B}$ at random. Let $b_0 \in \mathcal{A}_n$ such that $b_0(i) = 1$ if $i \in [m]$ and $b_0(i) = -1$ if $i \in [m + 1, 2m]$. All the vectors of $\mathcal{A}_n$ can be obtained by permuting the entries of $b_0$. Consequently consider the set

$$\mathcal{B} = \{S_i b_0, i \in [N]\}$$

where $(\sigma_i)_{i=1}^N$ are independent permutations chosen uniformly at random and $S_i(k, l) = \mathbb{1}_{l = \sigma_i(k)}$ is the permutation matrix of $\sigma_i$. By union bound and symmetry for all $t \geq 0$

$$\mathbb{P}\left[\exists b \neq b' \in \mathcal{B}, \|b - b'\|_1 \geq t\right] \leq N(N - 1)\mathbb{P}\left[\|Sb_0 - b_0\|_1 \geq t\right]$$

where $S$ is the matrix of a uniform permutation. Observe

$$\|Sb_0 - b_0\|_1 = 2\sum_{i=1}^m \mathbb{1}_{\sigma(i) > m} + 2\sum_{i=m+1}^{2m} \mathbb{1}_{\sigma(i) \leq m}$$

$$= 2\sum_{i=1}^{2m} A_{i, \sigma(i)}$$

where $A_{i,j} = \mathbb{1}_{i \leq m, j > m} + \mathbb{1}_{i > m, j \leq m}$. Since this matrix has its entries in $[0, 1]$, we can apply Chatterjee's concentration inequality for uniform permutations (10, Prop. 1.1) to $X = \sum_{i=1}^n A_{i\sigma(i)}$ to obtain

$$\mathbb{P}\left[|X - 2m| \geq t\right] \leq 2\exp\left(\frac{-t^2}{8m + t}\right).$$

We deduce that with probability at least $1 - 2N(N - 1)e^{-m/9}$, all pairs $b \neq b' \in \mathcal{B}$ satisfy $\|b - b'\|_1 \geq m = n/2$. Thus by taking $N \leq e^{m/40}$ this remains larger than $1 - e^{-\Theta(m)}$ so for $m$ large enough the set $\mathcal{B}$ satisfies the requirements with positive probability. $\square$

We now construct our two classes as follows. Consider integers $k, m \geq 1$ large enough, $n := km$ and the partition $[n] = \bigcup_{i=1}^k [(i - 1)m + 1, im] =: \bigcup_{i=1}^k V_i$. For the first class, let $Q$ be any reversible Markov chain on $[k]$ with uniform invariant measure. Then given $\varepsilon \in (0, 1/3)$ and a family of vectors $B = (b_i)_{i=1}^k \in \mathcal{B}_m^k$ taken from the previously constructed set, define for all $i \in [k], y \in [n]$ the emission probabilities:

$$p_B(i, y) := \frac{1 + 3\varepsilon b_i(y \mod m)}{m}. \tag{15}$$

Having $\varepsilon < 1/3$ and $b_i \in \{-1, 0, 1\}^k$ makes the entries of $p_B$ non-negative, and the fact that $\sum_j b_i(j) = 0$ implies that $\sum_y p_B(i, y) = 1$, so $p_B$ defines a stochastic matrix. Then let $P_B$ be the block markov chain with block partition $\bigcup_{i=1}^k V_i$, inter-block matrix $Q$ and emission matrix $p_B$. We construct the first class $\mathcal{P}_k^{(1)} := (P_B)_{B \in \mathcal{B}_m^k}$ as the collection of such matrices for all emission probabilities.

For the second class, let $p_0$ be the uniform emission matrix, defined by $p_0(i, y) = 1/m\mathbb{1}_{y \in V_i}$. We then want to use the set $\mathcal{B}_k$ to construct a family of inter-block matrices $Q_B$, however we require the chains to be reversible. A simple way to produce reversible is by considering a random walk on a network: given a non-directed graph $G$ on $n$ vertices equipped with non-negative weights $c = (c(e))_e$ on its edges, setting $P(x, y) := \frac{c(x,y)}{\sum_z c(x,z)}$ defines a reversible Markov chain with invariant measure $\mu(x) \propto \sum_y c(x, y)$ proportional to the sum of weights. Thus we will use the set $\mathcal{B}_k$ to define weights. Given a family of vectors $B = (b_i)_{i=1}^k \in \mathcal{B}_k^k$ define

$$c_B(i, j) := 1 + 3\varepsilon b_i(j). \tag{16}$$

Since $\varepsilon < 1/3$, the weights are non-negative and we can define a stochastic matrix $Q_B$ with transition probabilities proportional to the $c_B$. It is automatically reversible, with invariant measure at $i$ being

proportional to $\sum_{j\in[k]} c_B(i,j) = k$, so the uniform measure is invariant. Let $P_B$ be the block markov chain with block partition $\bigcup_{i=1}^k V_i$, inter-block matrix $Q_B$ and emission matrix $p_0$. We construct the second class $\mathcal{P}_k^{(2)} := (P_B)_{B\in\mathcal{B}_k^k}$ as the collection of such matrices for all inter-block matrices. Finally let $\mathcal{P}_k := \mathcal{P}_k^{(1)} \cup \mathcal{P}_k^{(2)}$.

**Lemma 5.** *For some constant $C > 0$, for $k, m$ large enough we have*

- *if $\left(\sum_{x\in[n]} Z_x\right) \le C\varepsilon^{-2}n$, there exists no $(\varepsilon, 1/2)$-PAC estimator for the class $\mathcal{P}_k^{(1)}$ and the $\|\cdot\|_{\infty,\infty}$ norm.*

- *if $\left(\sum_{x\in[n]} Z_x\right) \le C\varepsilon^{-2}k^2$, there exists no $(\varepsilon, 1/2)$-PAC estimator for the class $\mathcal{P}_k^{(2)}$ and the $\|\cdot\|_{\infty,\infty}$ norm.*

*Proof of Theorem 5.* Given $\kappa \ge 1$, let $k := \lfloor \kappa \rfloor$ and define the family $\mathcal{P}_k$ as described above. Every chain of $P \in \mathcal{P}$ is a block Markov chain with inter-block matrix $Q$ which is reversible with uniform invariant measure, hence Lemma 2 shows that $P^\dagger = P$ is also reversible with uniform invariant measure. Then Lemma 3 shows $\xi(P) = \xi(Q) \le k \le \kappa$. This proves the class $\mathcal{P}$ satisfies the requirements (i) and (ii). Finally Lemma 5 shows (iii). $\square$

The proof of Lemma 5 is based on Fano's inequality, as stated in (31, Lemma D.10).

**Proposition 4** (Fano's inequality)**.** *Let $P_1, \ldots, P_M$ be $M$ probability measures on a space $\Omega$. For any estimator $\hat{j}$ on $\Omega$*

$$\frac{1}{M}\sum_{j=1}^l P_j\left[\hat{j} \ne j\right] \ge 1 - \frac{\inf_{P_0} \frac{1}{M}\sum_{j=1}^M \mathrm{KL}(P_j, P_0) + \log 2}{\log M}$$

*where the infimum is on all probability measures on $\Omega$.*

*Proof of Lemma 5.* We start with $\mathcal{P}_k^{(1)}$. Consider $P_B, P_{B'} \in \mathcal{P}_k^{(1)}$. If $P_B \ne P_{B'}$ there exists $x \in [n]$ such that $P_B(x, \cdot) \ne P_{B'}(x, \cdot)$. Then by construction

$$\begin{aligned}
\|P_B - P_{B'}\|_{\infty,\infty} &\ge \sum_{y\in[n]} |P_B(x,y) - P_{B'}(x,y)| \\
&= \frac{3\varepsilon}{n} \sum_{y\in[m]} \left|b_{V(x)}(y) - b'_{V(x)}(y)\right| \\
&= \frac{3\varepsilon}{n} \left\|b_{V(x)} - b'_{V(x)}\right\|_1 \ge \frac{3\varepsilon}{2}(1 - 1/n)
\end{aligned}$$

by the second condition of Lemma 4. For $n$ large enough this is strictly larger than $\varepsilon$. Thus an $(\varepsilon, \delta)$-PAC estimator of $\mathcal{P}_k$ for the $\|\cdot\|_\infty$ norm yields an $(\varepsilon, \delta)$-PAC estimator of $B$. Given a stochastic matrix $P$ on $[n]$, let us write $\mathbb{P}_P$ for the law of the process generated when sampling independent $Z_i$ transitions at every state $i$. Thus by Fano's inequality (Proposition 4)

$$\frac{1}{|\mathcal{B}_m^k|}\sum_{B\in\mathcal{B}_m^k} \mathbb{P}_{P_B}\left[\hat{B} \ne B\right] \ge 1 - \frac{\frac{1}{|\mathcal{B}_m^k|}\sum_{B\in\mathcal{B}_m^k} \mathrm{KL}(\mathbb{P}_{P_B}, \mathbb{P}_{P_0}) + \log 2}{\log|\mathcal{B}_m^k|}$$

for any stochastic matrix $P_0$. The process generated by $P$ is a product of independent multinomial $\mathrm{Multinom}(Z_i, P(i, \cdot))$, thus we can compute

$$\mathrm{KL}(\mathbb{P}_P, \mathbb{P}_Q) = \sum_{x,y\in[n]} Z_x P(x,y) \log\left(\frac{P(x,y)}{Q(x,y)}\right).$$

We take for $P_0$ the block Markov chain with partition $(V_i)_i$, inter-block matrix $Q$ and uniform emission probability $p_0(i, y) = 1/m \mathbb{1}_{y \in V_i}$. Then exploiting the block structure we have

$$
\begin{aligned}
\mathrm{KL}(\mathbb{P}_{P_B}, \mathbb{P}_{P_0}) &= \sum_{x,y \in [n]} Z_x P_B(x, y) \log\left(\frac{P_B(x, y)}{P_0(x, y)}\right) \\
&= \sum_{x,y \in [n]} Z_x Q_B(V(x), V(y)) p_B((V(y), y)) \log\left(\frac{p_B(V(y), y)}{p_0(V(y), y)}\right) \\
&= \sum_{x \in [n]} \sum_{i \in [k], y \in V_i} Z_x Q_B(V(x), i) p_B(i, y) \log\left(\frac{p_B(i, y)}{p_0(i, y)}\right).
\end{aligned}
$$

Now for every $i \in [k]$, by (15)

$$
\begin{aligned}
\sum_{y \in V_i} p_B(i, y) \log\left(\frac{p_B(i, y)}{p_0(i, y)}\right) &= \sum_{j \in [m]} \frac{1 + 3\varepsilon b_i(j)}{m} \log\left(1 + 3\varepsilon b_i\right) \\
&\leq \sum_{j \in [m]} \left(3\varepsilon b_i(j) + \frac{9\varepsilon^2 b_i(j)^2}{m}\right) \\
&\leq 9\varepsilon^2.
\end{aligned}
$$

The second line uses the inequality $\log(1 + u) \leq u$, the last line is the consequence of having $\sum_j b_i(j) = 0$ and $b_i(j)^2 \in \{0, 1\}$. Summing over $i$ we get

$$
\mathrm{KL}(\mathbb{P}_{P_B}, \mathbb{P}_{P_0}) \leq 9\varepsilon^2 \sum_{\in [n]} Z_x
$$

so all in all we deduce

$$
\frac{1}{|\mathcal{B}_m^k|} \sum_{B \in \mathcal{B}_m^k} \mathbb{P}_{P_B}\left[\hat{B} \neq B\right] \geq 1 - \frac{9\varepsilon^2 \left(\sum_{x \in [n]} Z_x\right) + \log 2}{\log |\mathcal{B}_k|} \geq 1/2
$$

if $\left(\sum_{x \in [n]} Z_x\right) \leq \frac{\log(|\mathcal{B}_k|) - 2\log 2}{18\varepsilon^2}$. By Lemma 4 $\log |\mathcal{B}_m^k| \geq km/40 = n/40$ therefore if $\left(\sum_{x \in [n]} Z_x\right) \geq \frac{n/40 - 2\log 2}{18\varepsilon^2}$ there exists no $(\varepsilon, 1/2)$-PAC estimator of $\mathcal{P}_k^{(1)}$. This proves the first statement.

The second statement is proved similarly: as above an $(\varepsilon, \delta)$-PAC estimator of $\mathcal{P}_k^{(2)}$ necessarily yields an $(\varepsilon, \delta)$-PAC estimator of $B \in \mathcal{B}_k^k$. Applying Fano's inequality with $P_0$ the matrix with all entries equal to $1/n$, we are now led to compute

$$
\begin{aligned}
\mathrm{KL}(\mathbb{P}_{P_B}, \mathbb{P}_{P_0}) &= \sum_{x,y \in [n]} Z_x Q_B(V(x), V(y)) \frac{1}{m} \log\left(\frac{nQ_B(V(x), V(y))}{m}\right) \\
&= \sum_{i,j \in [k]} \sum_{x \in V_i} Z_x Q_B(i, j) \log\left(kQ_B(i, j)\right) \\
&= \sum_{i,j \in [k]} \sum_{x \in V_i} Z_x \frac{1 + 3\varepsilon b_i(j)}{k} \log\left(1 + 3\varepsilon b_i(j)\right)
\end{aligned}
$$

after which the proof follows the same arguments. $\qquad\square$

# C   Discussion and proof of Theorem 1

In this appendix we discuss limitations and possible extensions of our main result, Theorem 1, state a few key intermediate results used in the proof, and proceed to the proof.

## C.1   Discussion

**Control of the singular gap**    We suggested that Theorem 1 requires in practice a large gap $\Delta_r :=$ $\sigma_r - \sigma_{r+1}$. From the obvious upper bound $\Delta_r \leq \sigma_r$, the best we can hope for is having $\Delta_r \geq c\sigma_r$ with a constant $c < 1$. The following lemma shows that with bounded spectral irrecoverability we can always achieve $\Delta_r \geq c\sigma_r^2$ up to a small look-ahead in the singular values.

**Lemma 6.** *Let $A \in \mathbb{R}^{n \times m}$. Then for all $r$ there exists $r' \geq r$ such that*

$$\sigma_{r'+1} \leq \left(1 - \frac{(1-1/e)\sigma_r}{3\xi(A)}\right)\sigma_{r'} \quad \text{and} \quad \sigma_{r'} \geq e^{-2}\sigma_r.$$

*Proof.* First we bound $\sum_{i=1}^{n} \sigma_i \leq \xi(A)$. Consider now $r \in [n]$ and an arbitrary integer $l \geq 1$. If $\sigma_{r+i+1} \geq (1-1/l)\sigma_{r+i}$ for all $i \in [0, l-1]$ then

$$\sum_{i=1}^{r+l} \sigma_i \geq (r-1)\sigma_r + \sigma_r \sum_{i=0}^{l}(1-1/l)^i \geq \sigma_r \left(r - 1 + l\left[1 - (1-1/l)^{l+1}\right]\right)$$

$$\geq \sigma_r(r - 1 + l(1 - e^{-1}))$$

where we used the bound $(1-1/l)^{l+1} \leq e^{-1-1/l} \leq e^{-1}$. We thus get a contradiction if the right hand side is larger than $\xi(A)$, which occurs if $l \geq \frac{1}{1-1/e}\left(\frac{\xi(A)}{\sigma_r} - r + 1\right)$. Thus by taking $l = \left\lceil \frac{2}{1-1/e}\frac{\xi(A)}{\sigma_r} \right\rceil$ there must exist $i \leq l - 1$ such that $\sigma_{r+i+1} < (1 - 1/l)\sigma_{r+i}$. From its expression we can bound $l \leq \frac{3\xi(A)}{(1-1/e)\sigma_r}$, while by taking the smallest possible $i$ we also have

$$\sigma_{r+i} \geq (1 - 1/l)^l \sigma_r \geq e^{-2}\sigma_r,$$

noting that $l \geq 2$ and using the inequality $\log(1 - u) \geq -u/2$ for $u \leq 1/2$.    $\square$

**Extension to non-recurrent Markov chains**    Our result requires $\nu$ to have full support and to have its density w.r.t. an invariant measure $\nu_{\pi,\mathrm{inv}}$ bounded from above and below. This apparently rules out absorbing chains, and more generally chains with transient states where invariant measures do not have full support. We argue however that our result could also be applied in that case by decomposing the chain adequately. We can decompose the state space $\mathcal{X} = \bigcup_{i=1}\mathcal{R}_i \cup \bigcup_{j=1}\mathcal{T}_j$ into irreducible recurrent classes $\mathcal{R}_i$ and irreducible transient classes $\mathcal{T}_j$ (for all $x, y \in \mathcal{T}_j$ there exists a path of positive probability entirely contained in $\mathcal{T}_j$, but the chain eventually leaves $\mathcal{T}_j$) (see (7)). Our result could then be applied immediately on each $\mathcal{R}_i$ by taking the corresponding invariant measure, but it could also be applied to $\mathcal{T}_j$ as well. Indeed, an inspection of the proof reveals the only reason we require the invariant measure is to have contraction properties for $P_\pi$ in $\ell^p(\nu_{\pi,\mathrm{inv}})$, which holds if $P_\pi^\dagger$ is substochastic and $\nu$ is excessive, in the sense $\nu P_\pi \geq \nu$ pointwise. On a recurrent class an excessive measure coincides with an invariant measure but on a transient class $\mathcal{T}_j$ an excessive measure is a quasi-stationary measure (14), which describes the asymptotic behaviour of the chain conditioned to never leave $\mathcal{T}_j$. It can be obtained concretely as follows: restricting $P_\pi$ to a transient class $\mathcal{T}_j$ gives a substochastic matrix with non-negative entries, so we can still apply the Perron-Frobenius theorem. The first left eigenvector is the quasi-stationary measure we are looking for.

**Dependence in $\nu$ and $\nu_{\pi,\mathrm{inv}}$**    We have already explained after Theorem 1 why we consider two measures: $\nu$ is known by the practicioner and used to compute the SVD, while the invariant measure $\nu_{\pi,\mathrm{inv}}$ is more adapted to the analysis. Our proof makes use of a very rough comparison of the norms to relate the two and there is potentially room for improvement.

**Dependence in the policy $\pi$**    The very core of our proof relies on a concentration inequality for $\widehat{P} \in \mathbb{R}^{\mathcal{X} \times \mathcal{S}}$ (Theorem 8), which is independent of a policy. This is the key argument to obtain an off-policy result, which could also be used to make the result of Theorem 1 hold simultaneously over a set of policies (assuming for instance bounded density w.r.t. a reference policy). We are limited however by the invariance properties required for the measures, so we have preferred to state the result for a fixed policy only.

**Dependence in** $(1-\gamma)^{-1}$    Supposing that $k \leq (1-\gamma)^{-1}$, our estimation error (2) has a dependence in $(1-\gamma)^{-2}$ – which means that the sample complexity of our algorithm for estimating the (shifted) successor measure with $\varepsilon$-accuracy scales as $(1-\gamma)^{-4}$. This is probably sub-optimal, as learning an $\varepsilon$-optimal policies (in reward-specific RL) should have a sample complexity in $(1-\gamma)^{-3}$ (4). Further note that if one attempted to apply our result for the family of policies considered in a policy improvement scheme, we would typically require an additional factor $(1-\gamma)^{-1}$ in the sample complexity. From these observations, we conjecture that the sample complexity of estimating the (shifted) successor measure should scale as $(1-\gamma)^{-2}$.

**Dependence on the uniformity of the measure**    Our result also features a ratio $\max_{(s,a),(s',a')\in\mathcal{X}} \frac{\nu(s,a)}{Z_{s,a}\nu(s',a')}$ over all pairs $(s,a),(s',a')$. This forbids a highly heterogeneous measure but we believe this could be an artifact of the proof. For the most part of our argument, in particular for the concentration in spectral norm (Theorem 7), we are led to consider a ratio only over neighbouring pairs, i.e. such that $P(s,a,s') > 0$, which can be much smaller. The consideration of a ratio over all pairs come from a rough comparison between the $2-\infty$ and spectral norms in the leave-one-out analysis.

**Bound for the non-shifted successor measure**    Finally we note that Theorem 1 can be used to derive a bound on the estimation error of $M_\pi$ in $\|\cdot\|_{\infty,\infty}$ norm:

$$\left\| [\widehat{M}_{\pi,k}]_r - M_\pi \right\|_{\infty,\infty} \leq C\mathcal{E}_{\text{estim}} + \mathcal{E}_{\text{approx}} + 2k\gamma. \tag{17}$$

This is based on (40, Lemma 24.6). Let $X$ denote the chain with transition matrix $P_\pi$ and let $T \sim \mathcal{G}(1-\gamma)$ be a geometric variable independent of $X$ taking values in $\{0,1,\ldots\}$. Note that $M_\pi = (1-\gamma)\mathbb{E}\left[P_\pi^T\right]$ and $M_{\pi,k} = (1-\gamma)\mathbb{E}\left[P_\pi^{T+k}\right]$. Then writing $\|\mu - \nu\|_{\text{TV}}$ for the total variation distance between two measures $\mu,\nu$, it is simple to notice that

$$\begin{aligned}
\|(1-\gamma)M_{\pi,k} - (1-\gamma)M_\pi\|_{\infty,\infty} &= 2\max_{x\in\mathcal{X}} \|\mathbb{P}_x\left[X_T = \cdot\right] - \mathbb{P}_x\left[X_{T+k} = \cdot\right]\|_{TV} \\
&\leq 2\left\|\mathbb{P}\left[T = \cdot\right] - \mathbb{P}\left[T+k = \cdot\right]\right\|_{TV} \\
&\leq 2k\gamma(1-\gamma)
\end{aligned}$$

by (40, Lemma 24.6).

## C.2    Main steps of the proof of Theorem 1

The strategy to prove Theorem 1 consists in the following steps: we first prove concentration bounds for the simple estimator $\widehat{M}_{\pi,k}$ in spectral norm and strengthen them to $2-\infty$ norm. We have summed up the main steps in the diagram of Figure 5.

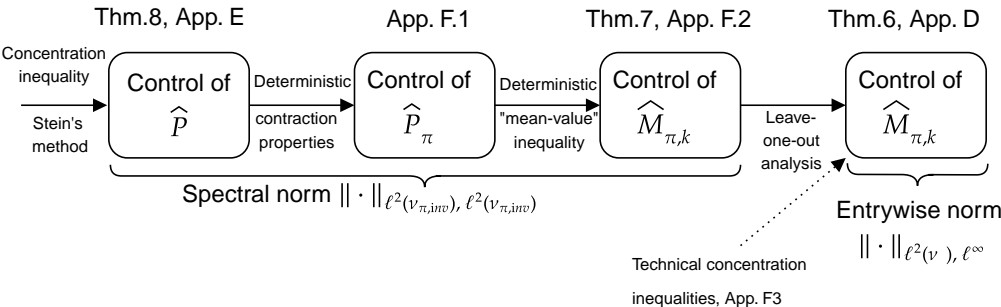

Figure 5: Main steps of the proof of Theorem 1.

We focus on the latter part of the proof first, i.e., obtaining bounds in entrywise norms from bounds in spectral norms. We state a general result for the estimation of a matrix, based on the so-called leave-one-out analyis (1). The proof is given in Appendix D.

**Theorem 6** (Leave-one-out analysis)**.** *Let $\nu$ a probability measure on $[n]$ with full support, $M, \widehat{M} \in \mathbb{R}^{n\times n}$ be positive semi-definite self-adjoint matrices w.r.t. $\nu$ and $E := \widehat{M} - M$. Write $M = U\Lambda U^\dagger$,*

$\widehat{M} = \widehat{U}\widehat{\Lambda}\widehat{V}$ *for the eigendecompositions of $M$ and $\widehat{M}$ respectively. Let $r \in [n]$, $H_r = \widehat{U}_r^\dagger U_r$ and $\Delta_r := \lambda_r(M) - \lambda_{r+1}(M)$, with the convention that $\lambda_{n+1}(M) := 0$. Suppose there exist $A, \varepsilon > 0$ such that $\|EU_r\|_{\ell^2(\nu),\ell^2(\nu)} \leq A\varepsilon$, $\|EM\|_{\ell^2(\nu),\ell^\infty} \leq \varepsilon \|M\|_{\ell^2(\nu),\ell^\infty}$ and*

$$\left\|E(\widehat{U}_r H_r - U_r)\right\|_{\ell^2(\nu),\ell^\infty} \leq \varepsilon \left( \left\|\widehat{U}_r H_r - U_r\right\|_{\ell^2(\nu),\ell^\infty} + \frac{A\varepsilon \|U_r\|_{\ell^2(\nu),\ell^\infty}}{\Delta_r} \right).$$

*Then there exists a universal constant $C > 0$ such that if $\varepsilon \leq \Delta_r/4A$*

$$\left\|\widehat{U}_r H_r - U_r\right\|_{\ell^2(\nu),\ell^\infty} \leq \frac{CA \|M\|_{\ell^2(\nu),\ell^\infty} \varepsilon}{|\lambda_r| \Delta_r} \tag{18}$$

*and*

$$\left\|[\widehat{M}]_r - [M]_r\right\|_{\ell^2(\nu),\ell^\infty} \leq \frac{CA |\lambda_1| \|M\|_{\ell^2(\nu),\ell^\infty} \varepsilon}{|\lambda_r| \Delta_r}. \tag{19}$$

The previous result requires several controls on the error matrix $E$ in spectral norm. The bound required on $\|E\|_{\ell^2(\nu),\ell^2(\nu)}$ will be the consequence of the following, which is an analogue of Theorem 1 for spectral norm. Note that the underlying measure is here required to be invariant (we explain how go back to an arbitrary measure $\nu$ in the proof of Theorem 1).

**Theorem 7** (Concentration in spectral norm). *Let $P \in \mathbb{R}^{\mathcal{X} \times \mathcal{S}}$ be the transition matrix of a finite MDP. Let $\mu$ be a probability measure on $\mathcal{S}$, $\pi$ a policy and $\nu(s, a) := \mu(s)\pi(s, a)$, which defines a probability measure on the set $\mathcal{X}$ of state-action pairs. Let $k \geq 0$, $\gamma \in (0, 1)$ and write*

$$C_{k,\gamma} := \frac{8\max(k, (1-\gamma)^{-1})}{1-\gamma}.$$

*For all policy $\pi$, for all $t \geq 0$ if $\nu$ is invariant for $P_\pi$ then*

$$\mathbb{P}\left[ \left\|\widehat{M}_{\pi,k} - M_{\pi,k}\right\|_{\ell^2(\nu),\ell^2(\nu)} \geq t \right] \leq 4n \exp\left( \frac{-t^2 \min_{(s,a)\sim(s',a')} \frac{Z_{s,a}\nu(s',a')}{\nu(s,a)+\nu(s',a')}}{8C_{k,\gamma}(t + 2C_{k,\gamma})} \right).$$

*Recall that $n$ denotes the cardinality of $\mathcal{X}$. The minimum is here over pairs $(s, a), (s', a') \in \mathcal{X}$ such that $P(s, a, s') > 0$.*

The proof of Theorem 7 can be split in three main steps: we will first prove a concentration inequality for $\widehat{P}$ using Stein's method of exchangeable pairs (see Theorem 8 in Appendix E). We will then extend this concentration result to $\widehat{P}_\pi$ and $\widehat{M}_{\pi,k}$ using deterministic arguments in Appendix F, where we will also establish a set of technical concentration inequalities, gathered in the following lemma.

Given $l \in [n]$, let

$$\widehat{P}^{(l)} := \widehat{P} + \mathbb{1}_l P(l, \cdot) - \mathbb{1}_l \widehat{P}(l, \cdot), \tag{20}$$

where $\mathbb{1}_l$ denotes the column vector with coordinates all equal to $0$ except for the $l$-th, equal to 1, and $P(l, \cdot)$ is the $l$-th row of $P$. $\widehat{P}^{(l)}$ is the matrix obtained by replacing the estimation of the $l$-th row by the true value of the matrix $P$, so that $\widehat{P}^{(l)} = \mathbb{E}\left[ \widehat{P} \mid (Y_{s,a,s'})_{(s,a) \neq l, s'} \right]$. We also write $\widehat{M}_{\pi,k}^{(l)} := \widehat{P}_\pi^{(l)}(I - \gamma \widehat{P}_\pi^{(l)})^{-1}$.

**Lemma 7** (Leave-one-out concentration). *Let $P \in \mathbb{R}^{\mathcal{X} \times \mathcal{S}}$ be the transition matrix of a finite MDP, $A \in \mathbb{R}^{\mathcal{S} \times p}$, $\pi$ be a policy and $\nu, \rho$ be probability measures on $\mathcal{X}, [p]$ respectively. Suppose $\nu$ is invariant for $P_\pi$. For some universal constants $C_1, C_2$ the following holds. Let $k \geq 0$, $\gamma \in [0, 1)$ and*

$$C_{k,\gamma} := \frac{C_1 \max(k, (1-\gamma)^{-1})}{1-\gamma}.$$

*For all $l \in [n]$, and $t \geq 0$ we have*

$$\mathbb{P}\left[\left\|\widehat{M}_{\pi,k}(l,\cdot)A - \widehat{M}_{\pi,k}^{(l)}(l,\cdot)A\right\|_{\ell^2(\rho)} \geq t \;\middle|\; (Y_{s,a,s'})_{(s,a)\neq l,s'}\right]$$

$$\leq (k+2)(p+1)\exp\left(\frac{-t^2 Z_l}{C_{k,\gamma}^2 \|A\|_{\ell^2(\rho),\ell^\infty}^2}\right), \tag{21}$$

$$\mathbb{P}\left[\left\|\widehat{M}_{\pi,k}A - M_{\pi,k}A\right\|_{\ell^2(\rho),\ell^\infty} \geq t\right]$$

$$\leq n(k+2)(p+1)\exp\left(\frac{-t^2 Z_{\min}}{C_{k,\gamma}^2 \|A\|_{\ell^2(\rho),\ell^\infty}^2}\right), \tag{22}$$

$$\mathbb{P}\left[\left\|\widehat{M}_{\pi,k}A - \widehat{M}_{\pi,k}^{(l)}A\right\|_{\ell^2(\rho),\ell^2(\nu)} \geq t\right]$$

$$\leq (k+2)(p+1)\exp\left(\frac{-t^2 Z_l}{C_{k,\gamma}^2 \nu(l)\|A\|_{\ell^2(\rho),\ell^\infty}^2}\right) + C_2 kpn\exp\left(\frac{-\min_{i\sim j}\frac{Z_i\nu(j)}{\nu(i)+\nu(j)}}{C_{k,\gamma}^2}\right), \tag{23}$$

$$\mathbb{P}\left[\left\|\widehat{M}_{\pi,k}^{(l)} - M_{\pi,k}\right\|_{\ell^2(\nu),\ell^2(\nu)} \geq t\right] \leq 4n\exp\left(\frac{-t^2 \min_{i\sim j}\frac{Z_i\nu(j)}{\nu(i)+\nu(j)}}{8C_{k,\gamma}\left(t + 2C_{k,\gamma}\|P^\dagger\|_{\infty,\infty}\right)}\right). \tag{24}$$

### C.3 Proof of Theorem 1

We now apply the results of the previous subsection to the estimator of the shifted successor measure. Fix a policy $\pi$, $\gamma \in (0,1)$, $k \geq 0$ and $M_{\pi,k} = P_\pi^k(I - \gamma P_\pi)^{-1}$. Recall the estimator $\widehat{M}_{\pi,k} := \widehat{P}_\pi^k(I - \gamma\widehat{P}_\pi)^{-1}$. Since the arguments require self-adjoint matrices let

$$M := \begin{pmatrix} 0 & M_{\pi,k} \\ M_{\pi,k}^\dagger & 0 \end{pmatrix}, \qquad \widehat{M} := \begin{pmatrix} 0 & \widehat{M}_{\pi,k} \\ \widehat{M}_{\pi,k}^\dagger & 0 \end{pmatrix} \tag{25}$$

and write $E := \widehat{M} - M$. Let $(\sigma_i)_{i=1}^n$, $(\widehat{\sigma}_i)_{i=1}^n$ be the singular values of $M_{\pi,k}$ and $\widehat{M}_{\pi,k}$ arranged in non-increasing order, and $M = U\Sigma U^\dagger$, $\widehat{M} = \widehat{U}\widehat{\Sigma}\widehat{U}^\dagger$ be the eigendecompositions of $M, \widehat{M}$, corresponding to eigenvalues $(\lambda_i)_{i=1}^{2n}$, $(\widehat{\lambda}_i)_{i=1}^{2n}$ arranged in non-increasing order of absolute values. These are related as $\lambda_{2i-1} = \sigma_i$, $\lambda_{2i} = -\sigma_i$ for all $i \in [n]$. We need thus to truncate the eigendecomposition of $M$ and $\widehat{M}$ to rank $2r$, however for notational simplicity, we write $r$ in subscripts instead of $2r$ except for $|\lambda_{2r}|$, but this is $\sigma_r$ by what precedes.

*Proof of Theorem 1.* In this proof we write $a \lesssim b$ if there exists a universal constant $C > 0$ such that $a \leq Cb$. Set

$$\varepsilon := \frac{\max(k, (1-\gamma)^{-1})}{1-\gamma}\sqrt{\max_{\substack{(s,a) \\ (s',a')\in\mathcal{X}}} \frac{\nu_{\pi,\text{inv}}(s,a)}{Z_{s',a'}\nu_{\pi,\text{inv}}(s',a')}\log(krn/\delta)} \tag{26}$$

Our goal is to apply Theorem 6 with $\widehat{M}$. We thus need to control $\|E\|_{\ell^2(\nu),\ell^2(\nu)}$, $\|EM\|_{\ell^2(\nu),\ell^\infty}$ and $\|E(\widehat{U}_r H_r - U_r)\|_{\ell^2(\nu),\ell^\infty}$, which we will bound using Theorem 7 and Lemma 7. Note however these only provide bounds in spectral norm with respect to $\nu_{\pi,\text{inv}}$, while we need a control with respect to $\nu$. We address this issue with a rough comparison of norms: for all $f \in \mathbb{R}^n$, we have

$$\|f\|_{\ell^2(\nu)}^2 \leq \left\|\frac{d\nu}{d\nu_{\pi,\text{inv}}}\right\|_\infty \|f\|_{\ell^2(\nu_{\pi,\text{inv}})}^2, \qquad \|f\|_{\ell^2(\nu_{\pi,\text{inv}})}^2 \leq \left\|\frac{d\nu_{\pi,\text{inv}}}{d\nu}\right\|_\infty \|f\|_{\ell^2(\nu)}^2$$

which in turn implies the comparisons of matrix norms

$$\|B\|_{\ell^2(\nu),\ell^2(\nu)} \leq \sqrt{\left\|\frac{d\nu}{d\nu_{\pi,\text{inv}}}\right\|_\infty \left\|\frac{d\nu_{\pi,\text{inv}}}{d\nu}\right\|_\infty} \|B\|_{\ell^2(\nu_{\pi,\text{inv}}),\ell^2(\nu_{\pi,\text{inv}})} \tag{27}$$

and

$$\|B\|_{\ell^2(\nu),\ell^\infty} \le \sqrt{\left\|\frac{d\nu_{\pi,\mathrm{inv}}}{d\nu}\right\|_\infty} \|B\|_{\ell^2(\nu_{\pi,\mathrm{inv}}),\ell^\infty}, \tag{28}$$

$$\|B\|_{\ell^2(\nu_{\pi,\mathrm{inv}}),\ell^\infty} \le \sqrt{\left\|\frac{d\nu}{d\nu_{\pi,\mathrm{inv}}}\right\|_\infty} \|B\|_{\ell^2(\nu),\ell^\infty}, \tag{29}$$

for all matrix $B$. Write $A := \sqrt{\left\|\frac{d\nu}{d\nu_{\pi,\mathrm{inv}}}\right\|_\infty \left\|\frac{d\nu_{\pi,\mathrm{inv}}}{d\nu}\right\|_\infty}$. Thus up to a factor $A$, we can use the concentration inequalities in spectral norms with respect to $\nu_{\pi,\mathrm{inv}}$. Note the maximum over all pairs $(s,a)$ in the definition of $\varepsilon$ (26) upper bounds the maximum over neighbouring pairs in Theorem 7. Thus if the values $Z_{s,a}$ are suffciently large to make $\varepsilon \le 1$ the theorem and (27) show that

$$\|E\|_{\ell^2(\nu_{\pi,\mathrm{inv}}),\ell^2(\nu_{\pi,\mathrm{inv}})} \lesssim \varepsilon, \qquad \|E\|_{\ell^2(\nu),\ell^2(\nu)} \lesssim A\varepsilon \tag{30}$$

with probability at least $1 - \delta$.

Similarly Equation 22 of Lemma 7 shows that with probability at least $1 - \delta$ we have

$$\|EM\|_{\ell^2(\nu),\ell^\infty} \lesssim \|M\|_{\ell^2(\nu),\ell^\infty}\,\varepsilon.$$

Finally we claim that with probability at least $1 - \delta$

$$\left\|E(\widehat{U}_r H_r - U_r)\right\|_{\ell^2(\nu),\ell^\infty} \lesssim \varepsilon\left(\left\|\widehat{U}_r H_r - U_r\right\|_{\ell^2(\nu),\ell^\infty} + \frac{A\varepsilon}{\Delta_r}\|U_r\|_{\ell^2(\nu),\ell^\infty}\right). \tag{31}$$

From Theorem 6 and a union bound this will be sufficient to get that

$$\left\|[\widehat{M}_{\pi,k}]_r - [M_{\pi,k}]_r\right\|_{\ell^2(\nu),\infty} \lesssim \frac{A\sigma_1 \|M\|_{\ell^2(\nu),\ell^\infty}\,\varepsilon}{\sigma_r(\sigma_r - \sigma_{r+1})}.$$

with probability $1 - \delta$. Theorem 1 follows from observing that $\|M\|_{\ell^2(\nu),\ell^\infty} = \max\left(\|M_{\pi,k}\|_{\ell^2(\nu),\ell^\infty}, \left\|M^\dagger_{\pi,k}\right\|_{\ell^2(\nu),\ell^\infty}\right)$, that $\nu_{\pi,\mathrm{inv}}$ in the definition of $\varepsilon$ (26) can be replaced by $\nu$ at the cost of an additional factor $A$, and using Lemma 1 for the approximation error.

*Proof of the claim (31).* We now prove the claim. Given $l \ge 1$, recall the definition of $\widehat{P}^{(l)}$ in (20) and let $\widehat{M}^{(l)}, \widehat{U}^{(l)}_r$, etc. be the matrices obtained as their general counterparts $\widehat{M}, \widehat{U}_r$, etc. but using $\widehat{P}^{(l)}$ instead of $\widehat{P}$. First we use triangle inequality to bound

$$\left\|E(\widehat{U}_r H_r - U_r)\right\|_{\ell^2(\nu),\ell^\infty} = \max_{l\in[n]}\left\|E(l,\cdot)(\widehat{U}_r H_r - U_r)\right\|_{\ell^2(\nu)}$$

$$\le \max_{l\in[n]}\left\|E(l,\cdot)(\widehat{U}_r H_r - \widehat{U}^{(l)}_r H^{(l)}_r)\right\|_{\ell^2(\nu)}$$

$$+ \max_{l\in[n]}\left\|E(l,\cdot)(\widehat{U}^{(l)}_r H^{(l)}_r - U_r)\right\|_{\ell^2(\nu)}.$$

The first term is bounded as

$$\left\|E(l,\cdot)(\widehat{U}_r H_r - \widehat{U}^{(l)}_r H^{(l)}_r)\right\|_{\ell^2(\nu)} \le \left\|\mathbb{1}^\top_l E\right\|_{\ell^2(\nu_{\pi,\mathrm{inv}})} \left\|\widehat{U}_r H_r - \widehat{U}^{(l)}_r H^{(l)}_r\right\|_{\ell^2(\nu),\ell^2(\nu_{\pi,\mathrm{inv}})}$$

$$\le \nu_{\pi,\mathrm{inv}}(l)^{-1/2}\|E\|_{\ell^2(\nu_{\pi,\mathrm{inv}}),\ell^2(\nu_{\pi,\mathrm{inv}})}$$

$$\times \left\|\widehat{U}_r H_r - \widehat{U}^{(l)}_r H^{(l)}_r\right\|_{\ell^2(\nu),\ell^2(\nu_{\pi,\mathrm{inv}})}$$

$$\lesssim \nu_{\pi,\mathrm{inv}}(l)^{-1/2}\varepsilon\left\|\widehat{U}_r H_r - \widehat{U}^{(l)}_r H^{(l)}_r\right\|_{\ell^2(\nu),\ell^2(\nu_{\pi,\mathrm{inv}})} \tag{32}$$

with probability at least $1 - \delta$, where in the second inequality we used (8).

On the other hand since $(\widehat{U}^{(l)}_r H^{(l)}_r - U_r)$ is independent of $Y_{l,\cdot}$. Equation 21 of Lemma 7 proves that conditional on $(Y_{s,a,s'})_{(s,a)\ne l,s'}$, with probability at least $1 - \delta$

$$\left\|E(l,\cdot)(\widehat{U}^{(l)}_r H^{(l)}_r - U_r)\right\|_{\ell^2(\nu)} \le \varepsilon\left\|\widehat{U}^{(l)}_r H^{(l)}_r - U_r\right\|_{\ell^2(\nu),\ell^\infty}. \tag{33}$$

The latter norm is bounded as

$$\left\|\widehat{U}_r^{(l)}H_r^{(l)} - U_r\right\|_{\ell^2(\nu),\ell^\infty} \leq \left\|\widehat{U}_r^{(l)}H_r^{(l)} - \widehat{U}_r H_r\right\|_{\ell^2(\nu),\ell^\infty} + \left\|\widehat{U}_r H_r - U_r\right\|_{\ell^2(\nu),\ell^\infty}$$

$$\leq \nu_{\pi,\mathrm{inv\,min}}^{-1/2}\left\|\widehat{U}_r^{(l)}H_r^{(l)} - \widehat{U}_r H_r\right\|_{\ell^2(\nu),\ell^2(\nu_{\pi,\mathrm{inv}})}$$

$$+ \left\|\widehat{U}_r H_r - U_r\right\|_{\ell^2(\nu),\ell^\infty}. \tag{34}$$

We are thus left with bounding

$$\left\|\widehat{U}_r^{(l)}H_r^{(l)} - \widehat{U}_r H_r\right\|_{\ell^2(\nu),\ell^2(\nu_{\pi,\mathrm{inv}})} = \left\|\widehat{U}_r^{(l)}\widehat{U}_r^{(l)\dagger}U_r - \widehat{U}_r\widehat{U}_r^\dagger U_r\right\|_{\ell^2(\nu),\ell^2(\nu_{\pi,\mathrm{inv}})}$$

$$\leq \left\|\widehat{U}_r^{(l)}\widehat{U}_r^{(l)\dagger} - \widehat{U}_r\widehat{U}_r^\dagger\right\|_{\ell^2(\nu),\ell^2(\nu_{\pi,\mathrm{inv}})}\left\|U_r\right\|_{\ell^2(\nu),\ell^2(\nu)}$$

$$= \left\|\widehat{U}_r^{(l)}\widehat{U}_r^{(l)\dagger} - \widehat{U}_r\widehat{U}_r^\dagger\right\|_{\ell^2(\nu),\ell^2(\nu_{\pi,\mathrm{inv}})}$$

$$\leq \left\|\frac{d\nu_{\pi,\mathrm{inv}}}{d\nu}\right\|_\infty^{1/2}\left\|\widehat{U}_r^{(l)}\widehat{U}_r^{(l)\dagger} - \widehat{U}_r\widehat{U}_r^\dagger\right\|_{\ell^2(\nu),\ell^2(\nu)}.$$

From the Davis-Kahan inequality (Prop. 7)

$$\left\|\widehat{U}_r^{(l)}\widehat{U}_r^{(l)\dagger} - \widehat{U}_r\widehat{U}_r^\dagger\right\|_{\ell^2(\nu),\ell^2(\nu)} \leq \frac{2\left\|(\widehat{M} - \widehat{M}^{(l)})\widehat{U}_r^{(l)}\right\|_{\ell^2(\nu),\ell^2(\nu)}}{\widehat{\sigma}_r^{(l)} - \widehat{\sigma}_{r+1}^{(l)}}$$

$$\leq \frac{2\left\|\frac{d\nu}{d\nu_{\pi,\mathrm{inv}}}\right\|_\infty^{1/2}\left\|(\widehat{M} - \widehat{M}^{(l)})\widehat{U}_r^{(l)}\right\|_{\ell^2(\nu),\ell^2(\nu_{\pi,\mathrm{inv}})}}{\widehat{\sigma}_r^{(l)} - \widehat{\sigma}_{r+1}^{(l)}}.$$

By Weyl's inequality (38) for all $i \in [n]$ we have $\left|\widehat{\sigma}_i^{(l)} - \sigma_i\right| \leq \left\|\widehat{M}^{(l)} - M\right\|_{\ell^2(\nu),\ell^2(\nu)}$, which is below $A\varepsilon$ up to a constant factor with probability at least $1 - \delta$ by (24) of Lemma 7. Hence $(\widehat{\sigma}_r^{(l)} - \widehat{\sigma}_{r+1}^{(l)})^{-1} \leq 2\Delta_r^{-1}$ if $A\varepsilon \leq \Delta_r/2$, and so we can bound

$$\left\|\widehat{U}_r^{(l)}H_r^{(l)} - \widehat{U}_r H_r\right\|_{\ell^2(\nu),\ell^2(\nu_{\pi,\mathrm{inv}})} \lesssim \frac{A\left\|(\widehat{M} - \widehat{M}^{(l)})\widehat{U}_r^{(l)}\right\|_{\ell^2(\nu),\ell^2(\nu_{\pi,\mathrm{inv}})}}{\Delta_r}.$$

Now comes the point where we use that the maximum in the definition of $\varepsilon$ involves all pairs $x, x' \in \mathcal{X}$: it has the consequence that

$$\max_{x,x'\in\mathcal{X}}\left(\frac{\nu_{\pi,\mathrm{inv}}(x) + \nu_{\pi,\mathrm{inv}}(x')}{Z_x\nu_{\pi,\mathrm{inv}}(x')}\right)\frac{Z_l\nu_{\pi,\mathrm{inv,min}}}{\nu_{\pi,\mathrm{inv}}(l)} \geq 1.$$

Therefore these term compensate each other when taking $t = \varepsilon\nu_{\pi,\mathrm{inv,min}}^{1/2}$ in Equation (23) of Lemma 7 which thus implies

$$\left\|(\widehat{M} - \widehat{M}^{(l)})\widehat{U}_r^{(l)}\right\|_{\ell^2(\nu),\ell^2(\nu_{\pi,\mathrm{inv}})} \lesssim \varepsilon\nu_{\pi,\mathrm{inv,min}}^{1/2}\left\|\widehat{U}_r^{(l)}\right\|_{\ell^2(\nu),\ell^\infty}$$

with probability at least $1 - \delta$. Then using Lemma 8 we bound

$$\left\|\widehat{U}_r^{(l)}\right\|_{\ell^2(\nu),\ell^\infty} \leq 2\left\|\widehat{U}_r^{(l)}H_r^{(l)}\right\|_{\ell^2(\nu),\ell^\infty}$$

$$\leq 2\left\|\widehat{U}_r^{(l)}H_r^{(l)} - U_r\right\|_{\ell^2(\nu),\ell^\infty} + 2\left\|U_r\right\|_{\ell^2(\nu),\ell^\infty}.$$

Combining the previous these inequalities we get

$$\nu_{\pi,\mathrm{inv,min}}^{-1/2}\left\|\widehat{U}_r^{(l)}H_r^{(l)} - \widehat{U}_r H_r\right\|_{\ell^2(\nu),\ell^2(\nu_{\pi,\mathrm{inv}})} \lesssim \frac{A\varepsilon}{\Delta_r}\left(\left\|\widehat{U}_r^{(l)}H_r^{(l)} - U_r\right\|_{\ell^2(\nu),\ell^\infty} + \left\|U_r\right\|_{\ell^2(\nu),\ell^\infty}\right) \tag{35}$$

and plugging this in (34) yields

$$\left\|\widehat{U}_r^{(l)}H_r^{(l)} - U_r\right\|_{\ell^2(\nu),\ell^\infty} \lesssim \frac{A\varepsilon}{\Delta_r}\left(\left\|\widehat{U}_r^{(l)}H_r^{(l)} - U_r\right\|_{\ell^2(\nu),\ell^\infty} + \left\|U_r\right\|_{\ell^2(\nu),\ell^\infty}\right) + \left\|\widehat{U}_r H_r - U_r\right\|_{\ell^2(\nu),\ell^\infty}. \tag{36}$$

so if $A\varepsilon/\Delta_r$ is sufficently small regrouping the two identical terms yields

$$\left\|\widehat{U}_r^{(l)} H_r^{(l)} - U_r\right\|_{\ell^2(\nu),\ell^\infty} \lesssim \left\|\widehat{U}_r H_r - U_r\right\|_{\ell^2(\nu),\ell^\infty} + \frac{A\varepsilon}{\Delta_r} \left\|U_r\right\|_{\ell^2(\nu),\ell^\infty}. \tag{37}$$

Plugging this back in (35) we get that

$$\nu_{\pi,\mathrm{inv\,min}}^{-1/2} \left\|\widehat{U}_r^{(l)} H_r^{(l)} - \widehat{U}_r H_r\right\|_{\ell^2(\nu),\ell^2(\nu_{\pi,\mathrm{inv}})} \lesssim \frac{A\varepsilon}{\Delta_r} \left\|\widehat{U}_r H_r - U_r\right\|_{\ell^2(\nu),\ell^\infty} + \left(\frac{A\varepsilon}{\Delta_r} + \frac{A^2\varepsilon^2}{\Delta_r^2}\right) \left\|U_r\right\|_{\ell^2(\nu),\ell^\infty}$$

$$\lesssim \frac{A\varepsilon}{\Delta_r} \left(\left\|\widehat{U}_r H_r - U_r\right\|_{\ell^2(\nu),\ell^\infty} + \left\|U_r\right\|_{\ell^2(\nu),\ell^\infty}\right).$$

All in all combining (32) and (33) $\left\|E(\widehat{U}_r H_r - U_r)\right\|_{\ell^2(\nu),\ell^\infty}$ is upper bounded by $\varepsilon$ times the latter equation $+ \varepsilon \times$ (37), so we obtain

$$\left\|E(\widehat{U}_r H_r - U_r)\right\|_{\ell^2(\nu),\ell^\infty} \lesssim \frac{A\varepsilon^2}{\Delta_r} \left(\left\|\widehat{U}_r H_r - U_r\right\|_{\ell^2(\nu),\ell^\infty} + \left\|U_r\right\|_{\ell^2(\nu),\ell^\infty}\right)$$

$$+ \varepsilon\left(\left\|\widehat{U}_r H_r - U_r\right\|_{\ell^2(\nu),\ell^\infty} + \frac{A\varepsilon}{\Delta_r} \left\|U_r\right\|_{\ell^2(\nu),\ell^\infty}\right)$$

$$\lesssim \varepsilon\left(\left\|\widehat{U}_r H_r - U_r\right\|_{\ell^2(\nu),\ell^\infty} + \frac{A\varepsilon}{\Delta_r} \left\|U_r\right\|_{\ell^2(\nu),\ell^\infty}\right).$$

using that $A\varepsilon \le \Delta_r/2$, which proves the claim.

$\square$

# D   Entry-wise guarantees: leave-one-out analysis

In this appendix we prove Theorem 6. The argument is based on the *leave-one-out* analysis introduced by Abbe & al (1), but our proofs are more aligned with the monograph (13). Overall, Theorem 6 is obtained after a few modifications in the proof of (13, Theorem 4.4). In all this section, we consider $\nu$ to be a probability measure on $[n]$ which for simplicity will be omitted from notation. The norms $\|\cdot\|$ with no subscript at all refer to the spectral norm $\|\cdot\|_{2,2}$.

## D.1   Entry-wise guarantees for SVD estimation: proof of Theorem 6

The theorem will be the consequence of the two following propositions. We use the same setup and notations as for Theorem 6. We recall that $H_r = \widehat{U}_r^\dagger U_r$.

**Proposition 5.** *Provided $\|EU_r\| \leq \Delta_r/2$, we have*

$$\left\|\widehat{U}_r H_r - U_r\right\|_{2,\infty} \leq \frac{\|EM\|_{2,\infty}}{|\lambda_r|^2}\left(1 + \frac{4\|EU_r\|}{|\lambda_r|}\right) + \frac{4\|M\|_{2,\infty}\|EU_r\|}{|\lambda_r|}\left(\frac{1}{|\lambda_r|} + \frac{1}{\Delta_r}\right)$$
$$+ \frac{2\left\|E(\widehat{U}_r H_r - U_r)\right\|_{2,\infty}}{|\lambda_r|}.$$

The following proposition shows how the control of the eigenspace via $\widehat{U}_r H_r - U_r$ implies a control on the matrix $[\widehat{M}]_r$ itself for the two-to-infinity norm.

**Proposition 6.** *Provided $\|EU_r\| \leq \Delta_r/8$,*

$$\left\|[\widehat{M}]_r - [M]_r\right\|_{2,\infty} \leq \frac{5}{2}|\lambda_1|\left\|\widehat{U}_r H_r - U_r\right\|_{2,\infty} + 4\|M\|_{2,\infty}\|EU_r\|\left(2\Delta_r^{-1} + |\lambda_r|^{-1}\right).$$

*Proof of Theorem 6.* Use Proposition 5 and the assumptions to bound $\|EU_r\| \leq \|E\| \leq A\varepsilon$ and

$$\left\|\widehat{U}_r H_r - U_r\right\|_{2,\infty} \leq \frac{\varepsilon\|M\|_{2,\infty}}{|\lambda_r|^2}\left(1 + \frac{4A\varepsilon}{|\lambda_r|}\right) + \frac{4A\|M\|_{2,\infty}\varepsilon}{|\lambda_r|}\left(\frac{1}{|\lambda_r|} + \frac{1}{\Delta_r}\right)$$
$$+ \frac{2\varepsilon}{|\lambda_r|}\left(\left\|\widehat{U}_r H_r - U_r\right\|_{2,\infty} + \frac{A\varepsilon\|U_r\|_{2,\infty}}{\Delta_r}\right).$$

If $\varepsilon \leq \Delta_r/4$ then $2\varepsilon \leq |\lambda_r|/2$ so we can rearrange terms to obtain

$$\left\|\widehat{U}_r H_r - U_r\right\|_{2,\infty} \leq \frac{2\varepsilon\|M\|_{2,\infty}}{|\lambda_r|^2}\left(1 + \frac{4A\varepsilon}{|\lambda_r|}\right) + \frac{8A\|M\|_{2,\infty}\varepsilon}{|\lambda_r|}\left(\frac{1}{|\lambda_r|} + \frac{1}{\Delta_r}\right)$$
$$+ \frac{4A\varepsilon^2\|U_r\|_{2,\infty}}{\Delta_r|\lambda_r|}.$$

Then use the crude bound $\|U_r\|_{2,\infty} \leq \lambda_r^{-1}\|U_r\Lambda_r\|_{2,\infty} = |\lambda_r|^{-1}\|M\|_{2,\infty}$. Keeping only the dominant term in the right-hand side and plugging this bound in Proposition 6 yields the results. □

## D.2   Technical lemmas

We now prove Propositions 5 and 6. We start by gathering three basic results that will be used in the proofs. The first one is Weyl's inequality, which states that for all matrices $\widehat{M}, M \in \mathbb{R}^{n\times n}$, for all $i \in [n]$,

$$\left|\lambda_i(M) - \lambda_i(\widehat{M})\right| \leq \left\|M - \widehat{M}\right\|. \tag{38}$$

Then we recall the classical Davis-Kahan inequalities. We refer to Corollary 2.8 of (13) for a proof.

**Proposition 7** (Davis-Kahan inequality). *For all $r \in [n]$, if $\left\|\widehat{M} - M\right\| \leq \Delta_r/2$ then*

$$\left\|\widehat{U}_{>r}^\dagger U_r\right\| = \left\|\widehat{U}_r\widehat{U}_r^\dagger - U_rU_r^\dagger\right\| \leq \frac{2\left\|(\widehat{M} - M)U_r\right\|}{\Delta_r}.$$

Finally we will need one more lemma. Given a matrix $A$ with singular value decomposition $A = U\Sigma V^\dagger$, define $\mathrm{sgn}(A) := UV^\dagger$.

**Lemma 8** ((13, Lemma 4.15))**.** *For all $r \geq 1$,*

$$\|H_r - \mathrm{sgn}(H_r)\| \leq \frac{2\|E\|^2}{\Delta_r^2} \tag{39}$$

*Furthermore if $\|E\| \leq \Delta_r/2$, then*

$$\|H_r^{-1}\| \leq 2. \tag{40}$$

We can now move to the proof of Propositions 5 and 6. The following lemma is taken from Lemma 4.16 of (13) and is an intermediate step towards Proposition 5. The only difference is that we do not assume $M$ to be of rank $r$, but this has no consequence on the proof.

**Lemma 9.** *(13)[Lemma 4.16] Provided $\|E\| \leq \Delta_r/2$,*

$$\left\|\widehat{U}_r H_r - \widehat{M} U_r \Lambda_r^{-1}\right\|_{2,\infty} \leq \frac{2\left\|\widehat{M}(\widehat{U}_r H_r - U_r)\right\|_{2,\infty}}{|\lambda_r|} + \frac{4\left\|\widehat{M} U_r\right\|_{2,\infty}\|E U_r\|}{|\lambda_r|^2}$$

*and*

$$\left\|\widehat{U}_r H_r - U_r\right\|_{2,\infty} \leq \frac{2\left\|\widehat{M}(\widehat{U}_r H_r - U_r)\right\|_{2,\infty}}{|\lambda_r|} + \frac{4\left\|\widehat{M} U_r\right\|_{2,\infty}\|E U_r\|}{|\lambda_r|^2} + \frac{\|E U_r\|_{2,\infty}}{|\lambda_r|}. \tag{41}$$

### D.3 Proof of Propositions 5 and 6

*Proof of Proposition 5.* The proposition is a simple continuation of Lemma 9. The first term of (41) is bounded using triangle inequality

$$\left\|\widehat{M}(\widehat{U}_r H_r - U_r)\right\|_{2,\infty} \leq \left\|M(\widehat{U}_r H_r - U_r)\right\|_{2,\infty} + \left\|E(\widehat{U}_r H_r - U_r)\right\|_{2,\infty}$$
$$\leq \|M\|_{2,\infty}\left\|(\widehat{U}_r H_r - U_r)\right\| + \left\|E(\widehat{U}_r H_r - U_r)\right\|_{2,\infty}.$$

Since $U_r^\dagger U_r = I$, one can notice that

$$\left\|\widehat{U}_r H_r - U_r\right\| = \left\|\widehat{U}_r \widehat{U}_r^\dagger U_r - U_r\right\|$$
$$= \left\|(\widehat{U}_r \widehat{U}_r^\dagger - U_r U_r^\dagger) U_r\right\|$$
$$\leq \left\|\widehat{U}_r \widehat{U}_r^\dagger - U_r U_r^\dagger\right\|.$$

Using the Davis-Kahan inequality (Prop. 7) we can thus bound

$$\left\|\widehat{M}(\widehat{U}_r H_r - U_r)\right\|_{2,\infty} \leq \frac{2\|M\|_{2,\infty}\|E U_r\|}{\Delta_r} + \left\|E(\widehat{U}_r H_r - U_r)\right\|_{2,\infty}. \tag{42}$$

Similarly the second term of (41) is bounded as

$$\left\|\widehat{M} U_r\right\|_{2,\infty} \leq \|M U_r\|_{2,\infty} + \|E U_r\|_{2,\infty}$$
$$\leq \|M\|_{2,\infty} + \|E U_r\|_{2,\infty}. \tag{43}$$

Finally we bound $\|E U_r\|_{2,\infty} \leq |\lambda_r|^{-1}\|E U_r \Lambda_r\|_{2,\infty} \leq |\lambda_r|^{-1}\|EM\|_{2,\infty}$, so combining (41) with (42) and (43) yields the result. $\qquad\square$

*Proof of Proposition 6.* Using the SVD decompositions of $\widehat{M}$ and $M$,

$$[\widehat{M}]_r - [M]_r = \widehat{U}_r \widehat{\Lambda}_r \widehat{U}_r^\dagger - U_r \Lambda_r U_r^\dagger$$
$$= \widehat{U}_r\left(\widehat{\Lambda}_r - H_r \Lambda_r H_r^\dagger\right)\widehat{U}_r^\dagger + \widehat{U}_r H_r \Lambda_r H_r^\dagger \widehat{U}_r^\dagger - U_r \Lambda_r U_r^\dagger. \tag{44}$$

**Bounding $\widehat{U}_r\left(\widehat{\Lambda}_r - H_r\Lambda_r H_r^\dagger\right)\widehat{U}_r^\dagger$:** Using $MU_r = U_r\Lambda_r$ and $\widehat{U}_r^\dagger\widehat{M} = \widehat{\Lambda}_r\widehat{U}_r^\dagger$

$$H_r\Lambda_r H_r^\dagger = \widehat{U}_r^\dagger U_r\Lambda_r H_r^\dagger = \widehat{U}_r^\dagger M U_r H_r^\dagger = \widehat{U}_r^\dagger(\widehat{M} + E)U_r H_r^\dagger$$
$$= \widehat{\Lambda}_r H_r H_r^\dagger + \widehat{U}_r^\dagger E U_r H_r^\dagger$$

and thus $\widehat{U}_r\left(\widehat{\Lambda}_r - H_r\Lambda_r H_r^\dagger\right)\widehat{U}_r^\dagger = \widehat{U}_r\widehat{\Lambda}_r(I - H_r H_r^\dagger)\widehat{U}_r^\dagger + \widehat{U}_r\widehat{U}_r^\dagger E U_r H_r^\dagger\widehat{U}_r^\dagger$. Then note

$$\left\|\widehat{U}_r\widehat{\Lambda}_r\right\|_{2,\infty} = \left\|[\widehat{M}]_r\widehat{U}_r\right\|_{2,\infty}$$
$$\leq \left\|[\widehat{M}]_r - [M]_r\right\|_{2,\infty} + \|M\|_{2,\infty}$$

and in particular

$$\left\|\widehat{U}_r\right\|_{2,\infty} = \left\|\widehat{U}_r\widehat{\Lambda}_r\widehat{\Lambda}_r^{-1}\right\|_{2,\infty} \leq \widehat{\lambda}_r^{-1}\left(\left\|[\widehat{M}]_r - [M]_r\right\|_{2,\infty} + \|M\|_{2,\infty}\right).$$

Thus we can bound

$$\left\|\widehat{U}_r\left(\widehat{\Lambda}_r - H_r\Lambda_r H_r^\dagger\right)\widehat{U}_r^\dagger\right\|_{2,\infty} \leq \left\|\widehat{U}_r\widehat{\Lambda}_r\right\|_{2,\infty}\left\|I - H_r H_r^\dagger\right\| + \left\|\widehat{U}_r\right\|_{2,\infty}\left\|\widehat{U}_r^\dagger E U_r H_r^\dagger\widehat{U}_r^\dagger\right\|$$
$$\leq \left(\left\|[\widehat{M}]_r - [M]_r\right\|_{2,\infty} + \|M\|_{2,\infty}\right)\left(\left\|I - H_r H_r^\dagger\right\| + \widehat{\lambda}_r^{-1}\|EU_r\|\right)$$

By the Davis-Kahan inequality (Prop. 7)

$$\left\|I - H_r H_r^\dagger\right\| = \left\|\widehat{U}_r^\dagger U_{>r}\right\|^2 \leq \frac{4\|EU_r\|^2}{\Delta_r^2}.$$

Then if $\|E\| \leq \Delta_r/2$, Weyl's inequality implies $\widehat{\lambda}_r \geq \lambda_r - \|E\| \geq \lambda_r/2$ and we can bound

$$\left\|\widehat{U}_r\left(\widehat{\Lambda}_r - H_r\Lambda_r H_r^\dagger\right)\widehat{U}_r^\dagger\right\|_{2,\infty} \leq 2\left(\left\|[\widehat{M}]_r - [M]_r\right\|_{2,\infty} + \|M\|_{2,\infty}\right)\|EU_r\|\left(\Delta_r^{-1} + \lambda_r^{-1}\right).$$

If furthermore $\|EU_r\| \leq \Delta_r/8$ we can make the factor in front of $\left\|[\widehat{M}]_r - [M]_r\right\|_{2,\infty}$ smaller than $1/2$ so by rearranging terms from (44) we get

$$\left\|[\widehat{M}]_r - [M]_r\right\|_{2,\infty} \leq 4\|M\|_{2,\infty}\|EU_r\|\left(\Delta_r^{-1} + \widehat{\lambda}_r^{-1}\right) + 2\left\|\widehat{U}_r H_r\Lambda_r H_r^\dagger\widehat{U}_r^\dagger - U_r\Lambda_r U_r^\dagger\right\|_{2,\infty}. \quad (45)$$

**Bounding $\widehat{U}_r H_r\Lambda_r H_r^\dagger\widehat{U}_r^\dagger - U_r\Lambda_r U_r^\dagger$:** One checks easily that

$$\widehat{U}_r H_r\Lambda_r H_r^\dagger\widehat{U}_r^\dagger - U_r\Lambda_r U_r^\dagger = (\widehat{U}_r H_r - U_r)\Lambda_r U_r^\dagger + U_r\Lambda_r(\widehat{U}_r H_r - U_r)^\dagger + (\widehat{U}_r H_r - U_r)\Lambda_r(\widehat{U}_r H_r - U_r)^\dagger.$$

The first of these terms can be bounded as

$$\left\|(\widehat{U}_r H_r - U_r)\Lambda_r U_r^\dagger\right\|_{2,\infty} \leq \left\|\widehat{U}_r H_r - U_r\right\|_{2,\infty}\|\Lambda_r\|\|U_r^\dagger\| \leq |\lambda_1|\left\|\widehat{U}_r H_r - U_r\right\|_{2,\infty},$$

the second as

$$\left\|U_r\Lambda_r(\widehat{U}_r H_r - U_r)^\dagger\right\|_{2,\infty} \leq \|U_r\Lambda_r\|_{2,\infty}\left\|\widehat{U}_r H_r - U_r\right\|$$
$$\leq \frac{2\|M\|_{2,\infty}\|EU_r\|}{\Delta_r}$$

using the same bound as for (42). Finally the third term combines the two bounds:

$$\left\|(\widehat{U}_r H_r - U_r)\Lambda_r(\widehat{U}_r H_r - U_r)^\dagger\right\|_{2,\infty} \leq \left\|\widehat{U}_r H_r - U_r\right\|_{2,\infty}\|\Lambda_r\|\left\|\widehat{U}_r H_r - U_r\right\|$$
$$\leq \frac{2|\lambda_1|\|EU_r\|\left\|\widehat{U}_r H_r - U_r\right\|_{2,\infty}}{\Delta_r}$$
$$\leq \frac{1}{4}|\lambda_1|\left\|\widehat{U}_r H_r - U_r\right\|_{2,\infty}$$

if $\|EU_r\| \leq \Delta_r/8$. All in all, this gives

$$\left\|\widehat{U}_r H_r\Lambda_r H_r^\dagger\widehat{U}_r^\dagger - U_r\Lambda_r U_r^\dagger\right\|_{2,\infty} \leq \frac{5}{4}|\lambda_1|\left\|\widehat{U}_r H_r - U_r\right\|_{2,\infty} + \frac{2\|M\|_{2,\infty}\|EU_r\|}{\Delta_r}. \quad (46)$$

Combining the two bounds (45) and (46) together yields the result. $\qquad\square$

# E Concentration in spectral norm for stochastic matrices

The goal of this appendix and the next is to prove Theorem 7 and Lemma 7. Instead of using off-the-shelf inequalities like Bernstein's inequality, we establish the concentration inequalities that we need using Stein's method of exchangeable pairs and more precisely the arguments of Chatterjee (10). We thus establish a general concentration inequality for the empirical estimator of a stochastic matrix in Theorem 8, which to the best of our knowledge is new. This appendix also gives a brief account of the method of exchangeable pairs for concentration and its extension to matrix inequalities developed in (52; 44).

## E.1 Main concentration inequality

**Theorem 8.** *Let $P \in \mathbb{R}^{n \times m}$ be a stochastic matrix and $\mu, \nu$ two probability measures on $[m], [n]$ respectively. Suppose that for each $i \in [n]$ we have drawn $Z_i$ independent samples from $P(i, \cdot)$ and for all $j \in [m]$, let $Y_{ij}$ count the number of samples with value $j$. Let $\widehat{P}(i, j) = Y_{ij}/Z_i$ be the empirical estimator of $P$. For all $t \geq 0$*

$$\mathbb{P}\left[\left\|\widehat{P} - P\right\|_{\ell^2(\mu), \ell^2(\nu)} \geq t\right] \leq (n + 3m) \exp\left(\frac{-t^2 \min_{i \sim j} \frac{Z_i \mu(j)}{\nu(i) + \mu(j)}}{8(t + 2\|P^\dagger\|_{\infty, \infty})}\right).$$

*where the minimum is over all pairs $(i, j) \in [n] \times [m]$ such that $P(i, j) > 0$, and the adjoint $P^\dagger$ is w.r.t. $\mu$ and $\nu$.*

## E.2 The method of exchangeable pairs

The method of exchangeable pairs consists eventually in establishing a differential inequality on the moment generating function (m.g.f.) that can be integrated to be combined with Chernoff's bound. In the matrix case, the argument can be extended to Hermitian matrices (and to more general matrices thanks to a classical dilation trick) using the matrix m.g.f.: letting $\bar{\mathrm{tr}} := n^{-1} \mathrm{tr}$ denote the normalized trace, the matrix m.g.f. of a Hermitian random matrix $Z \in \mathbb{C}^{n \times n}$ is

$$M_Z(\theta) := \mathbb{E}\,\bar{\mathrm{tr}}\left[e^{\theta Z}\right], \quad \theta \in \mathbb{R}. \tag{47}$$

We use a lower-case to denote the log of the m.g.f.:

$$m_Z(\theta) := \log M_Z(\theta).$$

We have the following m.g.f. bounds:

**Proposition 8.** *(52)[Prop. B.2] Let $Z \in \mathbb{C}^{n \times n}$ be a Hermitian random matrix. Let $\lambda_{\max}(Z), \lambda_{\min}(Z)$ denote repectively the maximal and minimal eigenvalue of $Z$. For all $t \in \mathbb{R}$,*

$$\mathbb{P}\left[\lambda_{\max}(Z) \geq t\right] \leq n \inf_{\theta > 0} \exp\left(-\theta t + m_Z(\theta)\right) \tag{48}$$

$$\mathbb{P}\left[\lambda_{\min}(Z) \leq t\right] \leq n \inf_{\theta < 0} \exp\left(-\theta t + m_Z(\theta)\right). \tag{49}$$

Suppose now $Z = \phi(X)$ where $X$ is a random variable taking values in a Banach space and $\phi$ is a map with Hermitian matrix values. We may write simply $\mathbb{E}[\phi]$ for $\mathbb{E}[\phi(X)]$. An exchangeable pair is simply a pair of random variables $(X, \tilde{X})$ such that $(X, \tilde{X}) \overset{(d)}{=} (\tilde{X}, X)$ in distribution. The technique requires next to find a map $K = K(X, \tilde{X})$ such that

1. $K(X, \tilde{X}) = -K(X, \tilde{X})$,
2. $\mathbb{E}\left[K(X, \tilde{X}) \mid X\right] = \phi(X) - \mathbb{E}[\phi]$.

where we write $\mathbb{E}[\phi] = \mathbb{E}[\phi(X)]$ to simplify notation. Combining the two properties we obtain that for any function $h$ with matrix values

$$\mathbb{E}[h(X)(\phi(X) - \mathbb{E}[\phi])] = \frac{1}{2}\mathbb{E}\left[(h(X) - h(\tilde{X}))K(X, \tilde{X})\right].$$

Applied to $h(X) = e^{\theta(\phi(X) - \mathbb{E}[\phi])}$ this implies that

$$\mathbb{E}\,\bar{\mathrm{tr}}\left[e^{\theta(\phi(X) - \mathbb{E}[\phi])}(\phi(X) - \mathbb{E}[\phi])\right] = \mathbb{E}\,\bar{\mathrm{tr}}\left[(e^{\theta(\phi(X) - \mathbb{E}[\phi])} - e^{\theta(\phi(\tilde{X}) - \mathbb{E}[\phi])})K(X, \tilde{X})\right].$$

One can then notice that $\mathbb{E}\,\bar{\mathrm{tr}}\left[\phi(X)e^{\theta\phi(X)}\right] = M'_{\phi(X)}(\theta)$. On the other hand, the right hand side can be further bounded using mean value inequality, which is straightforward in the scalar case while in the matrix case one arrives at the following.

**Lemma 10.** *(52)[Lemma B.4] For all $\theta \in \mathbb{R}$ we have*

$$\left|M'_{\phi(X) - \mathbb{E}[\phi]}(\theta)\right| \le \frac{1}{2}\,|\theta|\inf_{s>0}\mathbb{E}\,\bar{\mathrm{tr}}\left[(sV_\phi(X) + s^{-1}V_K(X))e^{\theta X}\right] \tag{50}$$

*where*

$$V_\phi(X) := \frac{1}{2}\mathbb{E}\left[(\phi(X) - \phi(\tilde{X}))^2 \mid X\right], \quad V_K(X) := \frac{1}{2}\mathbb{E}\left[K(X, \tilde{X})^2 \mid X\right].$$

The goal is then to obtain positive semi-definite (p.s.d) inequalities on $V_\phi$, typically of the form $V_\phi(X) \le \gamma I + \beta\phi(X)$. Here we write $A \ge B$ if $A - B$ is p.s.d.. This would result in a differential inequality on $M_{\phi(X) - \mathbb{E}[\phi]}(\theta)$ that can be integrated to obtain a bound on the log m.g.f.

$$m_Z(\theta) \le \frac{\gamma\theta^2}{2(1 - \beta\theta)}$$

which in turn translates to a Bernstein-like inequality for $\phi(X)$ by taking $\theta := t/(\gamma + \beta t)$ in Proposition 8:

**Theorem A.** *(52)[Thm. 3.1] Suppose there exist constants $\gamma, \beta \ge 0$, $s > 0$ such that*

$$V_\phi(X) \le s^{-1}(\gamma I + \beta\phi(X)), \qquad V_K(X) \le s(\gamma I + \beta\phi(X)) \quad a.s.$$

*Then for all $t \ge 0$*

$$\mathbb{P}\left[\lambda_{\max}(\phi(X)) \ge t\right] \le \exp\left(\frac{-t^2}{2(\gamma + \beta t)}\right)$$

$$\mathbb{P}\left[\lambda_{\min}(\phi(X)) \ge t\right] \le \exp\left(\frac{-t^2}{2\gamma}\right).$$

The previous arguments require the random matrix $\phi(X)$ to be Hermitian. The more general case can easily be dealt with thanks to a Hermitian dilation trick, namely by considering the random matrix $\begin{pmatrix} 0 & \phi(X) \\ \phi(X)^\dagger & 0 \end{pmatrix}$.

### E.3 Exchangeable pairs for independent multinomial variables

We now show how the method of exchangeable pairs can be applied to prove concentration for functionals of multinomial variables, which is the setting that appears in the case of transitions observed independently. Let $P \in \mathbb{R}^{n \times m}$ be a stochastic matrix, $Z = (Z_i)_{i \in [n]}$ a deterministic sequence of integers, $N := \sum_{i=1}^n Z_i$ and $Y = (Y_{i\cdot})_{i \in [n]}$ a random matrix of independent multinomial variables with $Y_{i\cdot} \sim \mathrm{Multinom}(Z_i, P(i, \cdot))$ for each $i$. We write $\widehat{P}(i, j) = Y_{ij}/Z_i$ for the empirical estimator of the matrix $P$. All norms $\|\cdot\|$ considered in this section are spectral norms with respect to underlying probability measures $\mu, \nu$ on $[m], [n]$.

The first step is to devise a nice exchangeable pair. A very natural one is as follows: let $I \in [n]$, $J, K \in [m]$ be three random indices such that $J, K$ are independent conditional on $I$ and with law given by

$$\mathbb{P}\left[I = i \mid Y\right] = \frac{Z_i}{N}$$

$$\mathbb{P}\left[J = j \mid I = i, Y\right] = \frac{Y_{ij}}{Z_i} = \widehat{P}(i, j), \quad \mathbb{P}\left[K = k \mid I = i, Y\right] = P(i, k). \tag{51}$$

Then let $\tilde{Y} := Y + \mathbb{1}_{IK} - \mathbb{1}_{IJ}$. To see why $(Y, \tilde{Y})$ forms an exchangeable pair, interpret $Y_{ij}$ as follows: consider $N$ balls of $n$ different colors are distributed in $m$ urns independently, such that for each $i$,

there are $Z_i$ balls of color $i$, which fall in urn $j$ with probability $P_{ij}$. Then the number of balls of color $i$ in urn $j$ has the law of $Y_{ij}$, and $\tilde{Y}_{ij}$ is realized by choosing one ball uniformly at random and putting it in a new urn. It is thus immediate that $(Y, \tilde{Y})$ forms an exchangeable pair.

Given a function $\phi : \mathbb{R}^{n \times m} \to \mathbb{R}^{n' \times m'}$, let $\Delta_{ij}\phi(Y) := \phi(Y + \mathbb{1}_{ij}) - \phi(Y)$. Note that if $\phi$ is an affine function, $\Delta_{ij}\phi(Y)$ does not in fact depend in $Y$, so we may write only $\Delta_{ij}\phi$.

**Proposition 9.** *Let $\phi : \mathbb{R}^{n \times m} \to \mathbb{R}^{n' \times m'}$ be an affine function with matrix values.*

(i) *For all $t \geq 0$,*

$$\mathbb{P}\left[\|\phi(Y) - \mathbb{E}\left[\phi(Y)\right]\| \geq t\right] \leq (n' + m') \exp\left(\frac{-t^2}{2N\left(\mathbb{E}_{I,J}\|\Delta_{IJ}\phi\|^2 + \mathbb{E}_{I,K}\|\Delta_{IK}\phi\|^2\right)}\right).$$

*the expectations $\mathbb{E}_{I,J}, \mathbb{E}_{I,K}$ being with respect to $I, J, K$ as defined in (51).*

(ii) *Furthermore, if almost surely $\phi(Y)$ is self-adjoint and $\Delta_{ij}\phi(Y) \geq 0$ for all $i, j$, then for all $t \geq 0$*

$$\mathbb{P}\left[\|\phi(Y) - \mathbb{E}\left[\phi(Y)\right]\| \geq t\right] \leq n' \exp\left(\frac{-t^2}{2\max_{i,j}\|\Delta_{ij}\phi\|\left(t + 2\|\mathbb{E}\left[\phi(Y)\right]\|\right)}\right).$$

*Proof.* In order to apply Theorem A, we suppose first $\phi$ to have self-adjoint values and will extend the first inequality to more general functions by a dilation trick. Note furthermore that we can suppose without loss of generality that the constant term of the function iz zero. Combined with the affine assumption, this implies that $\phi$ is linear and can be expressed as

$$\phi(Y) = \sum_{i,j} Y(i,j)\Delta_{ij}\phi$$

with $\Delta_{ij}\phi = \Delta_{ij}\phi(Y)$ being independent of $Y$. Averaging over $Y$ shows

$$\mathbb{E}\left[\phi(Y)\right] = \sum_{i,j} Z_i P(i,j)\Delta_{ij}\phi.$$

Then from the distributions of $I, J, K$ (51) we deduce

$$\mathbb{E}\left[\Delta_{IJ}\phi \mid Y\right] = \frac{1}{N}\phi(Y), \qquad \mathbb{E}\left[\Delta_{IK}\phi \mid Y\right] = \frac{1}{N}\mathbb{E}\left[\phi(Y)\right], \tag{52}$$

From these, we claim that $K(Y, \tilde{Y}) := N(\phi(Y) - \phi(\tilde{Y}))$ satisfies $\mathbb{E}\left[K(Y, \tilde{Y}) \mid Y\right] = \phi(Y) - \mathbb{E}\left[\phi(Y)\right]$. Indeed the definition of $\tilde{Y}$ implies

$$\begin{aligned}
\phi(Y) - \phi(\tilde{Y}) &= \Delta_{IJ}\phi(Y - \mathbb{1}_{IJ}) - \Delta_{IK}\phi(Y - \mathbb{1}_{IJ}) \\
&= \Delta_{IJ}\phi - \Delta_{IK}\phi
\end{aligned} \tag{53}$$

so averaging over $I, J, K$ and using (52) yields the claim.

In view of applying Theorem A, we are only left with upper bounding with $V_\phi(Y) = \frac{1}{2}\mathbb{E}\left[(\phi(Y) - \phi(\tilde{Y})^2) \mid Y\right]$ and $V_K(Y) = \frac{1}{2}\mathbb{E}\left[K(Y, \tilde{Y})^2 \mid Y\right]$, but by what precedes $V_K(Y) = N^2 V_\phi(Y)$. Using (53) and the p.s.d-convexity of the matrix square (ie.. $((1 - t)A + tB)^2 \leq (1 - t)A^2 + tB^2$ for all self-adjoint matrices $A, B$ and $t \in [0, 1]$)

$$\begin{aligned}
V_\phi(Y) &= \frac{1}{2}\mathbb{E}\left[\|\Delta_{IJ}\phi - \Delta_{IK}\phi\|^2 \mid Y\right] \\
&\leq \mathbb{E}\left[\|\Delta_{IJ}\phi\|^2 + \|\Delta_{IJ}\phi\|^2 \mid Y\right] \\
&= \mathbb{E}_{I,J}\|\Delta_{IJ}\phi\|^2 + \mathbb{E}_{I,K}\|\Delta_{IJ}\phi\|^2
\end{aligned}$$

Applying Theorem A with $\gamma = N\left(\mathbb{E}_{I,J}\|\Delta_{IJ}\phi\|^2 + \mathbb{E}_{I,K}\|\Delta_{IJ}\phi\|^2\right)$, $\beta = 0$ and $s = N$ gives thus the first inequality.

If we assume furthermore all the $\Delta_{ij}$ are psd, the positivity implies we can bound

$$(\Delta_{ij}\phi)^2 \leq \|\Delta_{ij}\phi\|\,\Delta_{ij}\phi \leq \max_{k,l}\|\Delta_{kl}\phi\|\,\Delta_{ij}\phi$$

for all $i,j$. Consequently,

$$\begin{aligned}
V_\phi(Y) &\leq \max_{k,l}\|\Delta_{kl}\phi\|\,\mathbb{E}\left[\Delta_{IJ}\phi + \Delta_{IK}\phi \mid Y\right] \\
&= N^{-1}\max_{k,l}\|\Delta_{kl}\phi\|\,(\phi(Y) + \mathbb{E}\left[\phi\right]) \\
&= N^{-1}\max_{k,l}\|\Delta_{kl}\phi\|\,(\phi(Y) - \mathbb{E}\left[\phi\right] + 2\mathbb{E}\left[\phi\right])
\end{aligned}$$

by (52). Thus applying Theorem A again with $s = N$ but this time $\gamma = 2\max_{k,l}\|\Delta_{kl}\phi\|\,\mathbb{E}\left[\phi\right]$ and $\beta = \max_{k,l}\|\Delta_{kl}\phi\|$ yields the second inequality.

Finally the first inequality extends to the non self-adjoint case by considering the self-adjoint dilation $\psi(Y) := \begin{pmatrix} 0 & \phi(Y) \\ \phi(Y)^\dagger & 0 \end{pmatrix}$, simply noticing that $\|\psi(Y) - \mathbb{E}\left[\psi(Y)\right]\| = \|\phi(Y) - \mathbb{E}\left[\phi(Y)\right]\|$ and $\|\Delta_{ij}\psi(Y)\|^2 = \|\Delta_{ij}\phi(Y)\|^2$. $\qquad\square$

### E.4 Concentration of the empirical estimator

We now apply the previous results in order to prove Theorem 8. Note that as a function of $Y$,

$$\Delta_{ij}\widehat{P} = \frac{1}{Z_i}\mathbb{1}_i\mathbb{1}_j^\top \tag{54}$$

if $P(i,j) > 0$ and $0$ otherwise.

The proof will require controlling the adjoint $\widehat{P}^\dagger$, which leads us to first prove concentration of the functional $\nu\widehat{P}(j) - \nu P(j)$, for each $j \in [n]$.

**Lemma 11.** *For all $j \in [m]$, $t \geq 0$,*

$$\mathbb{P}\left[\left|\nu\widehat{P}(j) - \nu P(j)\right| \geq t\right] \leq 2\exp\left(\frac{-t^2\min_{i:i\sim j}\frac{Z_i}{\nu(i)}}{2(t + 2\nu P(j))}\right) \tag{55}$$

*where $i \sim j$ denotes the fact that $P(i,j) > 0$.*

*As a consequence*

$$\mathbb{P}\left[\left|\|\widehat{P}^\dagger\|_{\infty,\infty} - \|P^\dagger\|_{\infty,\infty}\right| \geq t\right] \leq 2m\exp\left(\frac{-t^2\min_{i:i\sim j}\frac{Z_i\mu(j)}{\nu(i)}}{2(t + 2\|P^\dagger\|_{\infty,\infty})}\right) \tag{56}$$

*Proof.* Fix $j \in [m]$ and let $\phi(Y) := \nu\widehat{P}(j)$. This is a scalar function, linear with respect to $Y$, with

$$0 \leq \Delta_{ik}\phi(Y) = \frac{\nu(i)}{Z_i}\mathbb{1}_{k=j} \leq \max_{l:l\sim j}\frac{\nu(l)}{Z_l}.$$

Thus we can apply the second inequality of Proposition 9 to obtain the first inequality.

The second inequality is a consequence of the first: note that by (11) $\|\widehat{P}^\dagger\|_{\infty,\infty} = \max_{j\in[m]}\frac{\nu\widehat{P}(j)}{\mu(j)}$ and so by union bound

$$\begin{aligned}
\mathbb{P}\left[\left|\|\widehat{P}^\dagger\|_{\infty,\infty} - \|P^\dagger\|_{\infty,\infty}\right| \geq t\right] &= \mathbb{P}\left[\max_{j\in[m]}\left|\nu\widehat{P}(j) - \nu P(j)\right| \geq t\mu(j)\right] \\
&\leq 2m\max_{j\in[m]}\exp\left(\frac{-t^2\min_{i\sim j}\frac{Z_i\mu(j)^2}{\nu(i)}}{2(t\mu(j) + 2\nu P(j))}\right) \\
&\leq 2m\exp\left(\frac{-t^2\min_{i\sim j}\frac{Z_i\mu(j)}{\nu(i)}}{2(t + 2\|P^\dagger\|_{\infty,\infty})}\right).
\end{aligned}$$

$\qquad\square$

Moving to the concentration of $\widehat{P}$, the inequality of Point (i) in Proposition 9 does not yield an optimal result due to the additional factor $N$. To resort to the second inequality, we need to consider a p.s.d. matrix. The idea is that for any square self-adjoint stochastic matrix $Q$, $I - Q$ is p.s.d.. Thus $Q = I - (I - Q)$ can always be expressed as a difference of two p.s.d matrices, which motivates us to also express our self-adjoint random matrix as the difference of two psd matrices.

**Lemma 12.** *Let $\mu$ be a measure on $[n]$, $Q \in \mathbb{R}^{n \times n}$ such that $Q^{\dagger} = Q$ with respect to $\mu$, and $f \in \mathbb{R}^n$. If $Q\mathbb{1} = 0$ and $Q(i, j) \leq 0$ for all $i \neq j$, then*

$$\langle Qf, f \rangle = -\frac{1}{2} \sum_{i,j \in [n]} \mu(i)Q(i,j)\left(f(i) - f(j)\right)^2.$$

*In particular $Q \geq 0$.*

*Proof.* Let $f \in \mathbb{R}^n$. Then

$$\langle Qf, f \rangle_\mu = \sum_{i,j} \mu(i)Q(i,j)f(i)f(j)$$

$$= \sum_{i,j} \mu(i)Q(i,j)f(i)\left(f(j) - f(i)\right)$$

$$= \frac{1}{2} \sum_{i,j} \mu(i)Q(i,j)f(i)\left(f(j) - f(i)\right) + f(j)\left(f(i) - f(j)\right)$$

$$= -\frac{1}{2} \sum_{i,j} \mu(i)Q(i,j)\left(f(i) - f(j)\right)^2 \geq 0.$$

The second equality uses $Q\mathbb{1} = 0$, the third uses $Q^{\dagger} = Q$ and the inequality arises from the hypothesis that $Q(i, j) \leq 0$ whenever $i \neq j$. $\qquad\square$

*Proof of Theorem 8.* We use the standard dilation trick to reduce to the self-adjoint case, i.e. we prove concentration of the $(n + m) \times (n + m)$ matrix $\left(\begin{smallmatrix} 0 & \widehat{P} \\ \widehat{P}^{\dagger} & 0 \end{smallmatrix}\right)$. The concentration is in spectral norm with respect to the probability measure $\frac{1}{2}(\nu_{|[n]} + \mu_{|[m]})$, which gives the same adjoint operators.

Let $D_1, D_2$ be the two random diagonal matrices defined by $D_1(i) = \sum_{j \in [m]} \widehat{P}(i, j)$ and $D_2(j) = \sum_{i \in [n]} \nu(i)\widehat{P}(i,j)/\mu(j) = \sum_{i \in [n]} \widehat{P}^{\dagger}(j,i)$. Note that $D_1$ evaluates to the identity matrix, however it is not equal to the identity as a formal function of the random variable $Y$. One can then express

$$\begin{pmatrix} 0 & \widehat{P} \\ \widehat{P}^{\dagger} & 0 \end{pmatrix} = \begin{pmatrix} D_1 & 0 \\ 0 & D_2 \end{pmatrix} - \begin{pmatrix} D_1 & -\widehat{P} \\ -\widehat{P}^{\dagger} & D_2 \end{pmatrix}.$$

and thus

$$\left\|\widehat{P} - P\right\| \leq \lambda_{\max}\left(\left(\begin{smallmatrix} D_1 & 0 \\ 0 & D_2 \end{smallmatrix}\right) - \left(\begin{smallmatrix} \mathbb{E}[D_1] & 0 \\ 0 & \mathbb{E}[D_2] \end{smallmatrix}\right)\right) - \lambda_{\min}\left(\left(\begin{smallmatrix} D_1 & -\widehat{P} \\ -\widehat{P}^{\dagger} & D_2 \end{smallmatrix}\right) - \left(\begin{smallmatrix} \mathbb{E}[D_1] & -P \\ -P^{\dagger} & \mathbb{E}[D_2] \end{smallmatrix}\right)\right). \tag{57}$$

Notice that the norm of the diagonal matrix $D_2$ is $\|D_2\| = \max_{j \in [m]} \nu\widehat{P}(j)/\mu(j) = \left\|\widehat{P}^{\dagger}\right\|_{\infty,\infty}$, so from Lemma 11 we have

$$\mathbb{P}\left[\lambda_{\max}\left(\left(\begin{smallmatrix} D_1 & 0 \\ 0 & D_2 \end{smallmatrix}\right) - \left(\begin{smallmatrix} \mathbb{E}[D_1] & 0 \\ 0 & \mathbb{E}[D_2] \end{smallmatrix}\right)\right) \geq t\right] \leq 2m \exp\left(\frac{-t^2 \min_{i \sim j} \frac{Z_i \mu(j)}{\nu(i)}}{2(t + 2\left\|P^{\dagger}\right\|_{\infty,\infty})}\right)$$

$$\leq 2m \exp\left(\frac{-t^2}{2\kappa(t + 2\left\|P^{\dagger}\right\|_{\infty,\infty})}\right) \tag{58}$$

where we write $\kappa := \max_{i \sim j} \frac{\nu(i) + \mu(j)}{Z_i \mu(j)}$. Thus it remains to establish the concentration of the matrix $\phi := \left(\begin{smallmatrix} D_1 & -\widehat{P} \\ -\widehat{P}^{\dagger} & D_2 \end{smallmatrix}\right)$, for which we will apply Proposition 9. By Lemma 12, for all $f = (f_1\, f_2)^{\top} \in \mathbb{R}^{n+m}$,

$$\langle \Delta_{ij}\phi(Y)f, f \rangle = \frac{1}{2} \sum_{x \in [n], y \in [m]} \nu(x)\Delta_{ij}\widehat{P}(x,y)\left(f_1(x) - f_2(y)\right)^2$$

$$= \frac{\nu(i)}{2Z_i}\left(f_1(i) - f_2(j)\right)^2$$

$$\leq \frac{\nu(i)}{2Z_i}\left(\frac{1}{\nu(i)} + \frac{1}{\mu(j)}\right)\left(\nu(i)f_1(i)^2 + \mu(j)f_2(j)^2\right)$$

applying Cauchy-Schwarz inequality. Assuming furthermore $\|f\| = 1$ we can bound $\frac{1}{2}\left(\nu(i)f_1(i)^2 + \mu(j)f_2(j)^2\right) \leq 1$, so we deduce

$$\|\Delta_{ij}\phi(Y)\| = \sup_{\|f\|=1} \langle \Delta_{ij}\phi(Y)f, f \rangle$$

$$\leq \frac{\nu(i) + \mu(j)}{Z_i\mu(j)}$$

$$\leq \kappa.$$

The above computation also shows that $\Delta_{ij}\phi(Y)$ is p.s.d., so Point (ii) of Proposition 9 applies to yield that for all $t \geq 0$

$$\mathbb{P}\left[\,\lambda_{\min}(\phi(Y) - \mathbb{E}\left[\phi\right]) \leq -t \mid Z\right] \leq (n+m)\exp\left(\frac{-t^2}{2\kappa(t + 2\,\|\mathbb{E}\left[\phi\right]\|)}\right).$$

By the Riesz-Thorin interpolation theorem and duality (9), $\|\mathbb{E}\left[\phi\right]\| = \|P\|_{2,2} \leq \|P\|_{1,1}^{1/2} \|P\|_{\infty,\infty}^{1/2} = \left\|P^\dagger\right\|_{\infty,\infty}^{1/2}$, and using (57) and (58) we get finally

$$\mathbb{P}\left[\,\|\widehat{P} - P\| \geq t \mid Z\right] \leq \mathbb{P}\left[\,\left\|\widehat{P}^\dagger\right\|_{\infty,\infty} - \left\|P^\dagger\right\|_{\infty,\infty} \geq t/2 \,\Big|\, Z\right]$$

$$+ \mathbb{P}\left[\,\lambda_{\min}(\phi(Y) - \mathbb{E}\left[\phi\right]) \leq -t/2 \mid Z\right]$$

$$\leq 2m\exp\left(\frac{-t^2}{8\kappa(t + 2\,\|P^\dagger\|_{\infty,\infty})}\right) + (n+m)\exp\left(\frac{-t^2}{8\kappa(t + 2\,\|P^\dagger\|_{\infty,\infty}^{1/2})}\right).$$

which yields the result, observing $\left\|P^\dagger\right\|_{\infty,\infty} \geq 1$. $\qquad\square$

## F  Extension to shifted successor measures and leave-one-out concentration

In this appendix we leverage the concentration for the empirical estimator $\widehat{P} \in \mathbb{R}^{\mathcal{X} \times \mathcal{S}}$ established in Theorem 8 to deduce concentration for more complex functionals of $\widehat{P}$. First we exploit linearity and contraction properties of the map $P \mapsto P_\pi$ to obtain concentration in spectral norm for the policy-evaluated matrix $\widehat{P}_\pi$. Then we use simple identities to deduce concentration for the shifted successor measures $\widehat{M}_{\pi,k}$. Finally we establish the technical concentration inequalities of Lemma 7.

### F.1  Contraction properties of the map $P \mapsto P_\pi$

Given a policy $\pi$, consider the following linear operator on vectors:
$$
\begin{array}{rcl}
K_\pi : \mathbb{R}^{\mathcal{X}} & \to & \mathbb{R}^{\mathcal{S}} \\
f & \mapsto & \sum_{a \in \mathcal{A}} \pi(s,a) f(s,a).
\end{array}
\tag{59}
$$
Note that $K$ can be identified with a $\mathcal{S} \times \mathcal{X}$ matrix, namely $K_\pi(s',s,a) = \pi(s,a)\mathbb{1}_{s'=s}$ so we can see that
$$
P_\pi = PK_\pi.
\tag{60}
$$
Given a probability measure $\mu$ on $\mathcal{S}$, let us write $\mu \rtimes \pi$ the probability measure on $\mathcal{X}$ given by $\mu \rtimes \pi(s,a) := \mu(s)\pi(s,a)$. Note any probability measure on $\mathcal{X}$ has this form, as $\pi(s,\cdot)$ is thus the law of the action conditional of the state.

**Lemma 13.** *For all probability measure $\mu$ on $\mathcal{S}$, policy $\pi$,*

*(i)* $\|K_\pi\|_{\infty,\infty} = 1$,

*(ii)* $\|K_\pi\|_{\ell^2(\mu \rtimes \pi), \ell^2(\mu)} \le 1$,

*(iii) for all* $(s,a),(s',a') \in \mathcal{X}$, $P^\dagger(s',s,a) = (P_\pi)^\dagger(s',a',s,a)$.

*Here $P^\dagger$ is the adjoint of $P$ as an operator $\ell^2(\mu) \to \ell^2(\mu \rtimes \pi)$, while $P_\pi^\dagger$ is the adjoint of $P_\pi$ which is an operator $\ell^2(\mu \rtimes \pi) \to \ell^2(\mu \rtimes \pi)$.*

*Proof.* Let $\mu$ be a probability measure on $\mathcal{S}$ and $\pi$ a policy. Point (i) comes from the fact that $K$ is a stochastic matrix. Then by Jensens's inequality for all $f \in \mathbb{R}^{\mathcal{X}}$

$$
\begin{aligned}
\|K_\pi f\|_{\ell^2(\mu)}^2 &= \sum_{s \in \mathcal{S}} \mu(s) \left( \sum_{a \in \mathcal{A}} \pi(s,a) f(s,a) \right)^2 \\
&\le \sum_{(s,a) \in \mathcal{X}} \mu(s)\pi(s,a) f(s,a)^2 = \|f\|_{\ell^2(\mu \rtimes \pi)}^2
\end{aligned}
$$

which implies that $\|K_\pi\|_{\ell^2(\mu \rtimes \pi), \ell^2(\mu)} \le 1$. Finally, by definition

$$
\begin{aligned}
P^\dagger(s',s,a) &= \frac{\mu(s)\pi(s,a)P(s,a,s')}{\mu(s')} \\
&= \frac{\mu(s)\pi(s,a)P(s,a,s')\pi(s',a')}{\mu(s')\pi(s',a')} \\
&= \frac{\mu \rtimes \pi(s,a)P_\pi(s,a,s',a')}{\mu \rtimes \pi(s',a')} = P_\pi^\dagger(s',a',s,a)
\end{aligned}
$$

and thus $\|P^\dagger\|_{\infty,\infty} = \|P_\pi^\dagger\|_{\infty,\infty}$. $\qquad\square$

### F.2  Extension to shifted successor measure: proof of Theorem 7

The concentration of shifted successor measures will be the consequence of a deterministic mean-value like bound, itself a consequence submultiplicativity and the following well-known identities: the telescopic sum formula

$$
\prod_{i=1}^{k} a_i - \prod_{i=1}^{k} b_i = \sum_{j=1}^{k} \left( \prod_{i=1}^{j-1} a_i \right) (a_j - b_j) \left( \prod_{i=j+1}^{k} b_k \right)
\tag{61}
$$

and the resolvent identity:

$$a^{-1} - b^{-1} = a^{-1}(b-a)b^{-1} = b^{-1}(b-a)a^{-1}. \tag{62}$$

**Lemma 14.** *Let $\mu$ a probability measure on $[n]$ and consider here $\|\cdot\| = \|\cdot\|_{\ell^2(\mu), \ell^2(\mu)}$. Let $A, B \in \mathbb{R}^{n \times n}$ with $1 \le \|B\| \le \|A\|$, $k \ge 0, \gamma \in [0, \|A\|^{-1})$ and write $\phi_{k,\gamma}(A) := A^k(I - \gamma A)^{-1}$. Suppose $\|A - B\| \le \min\left(\frac{\|B\|}{k}, \frac{1-\gamma\|B\|}{2}\right)$. Then*

$$\|\phi_{k,\gamma}(A) - \phi_{k,\gamma}(B)\| \le \frac{8\|B\|^k \max(k, (1-\gamma\|B\|)^{-1})}{1-\gamma\|B\|} \|A - B\|.$$

*Proof.* First decomposing,

$$A^k(I-\gamma A)^{-1} - B^k(I-\gamma B)^{-1} = (A^k - B^k)(I-\gamma B)^{-1} + B^k\left[(I-\gamma A)^{-1} - (I-\gamma B)^{-1}\right]$$
$$+ (A^k - B^k)\left[(I-\gamma A)^{-1} - (I-\gamma B)^{-1}\right]$$

submultiplicativity of the spectral norm allows to bound

$$\|\phi_{k,\gamma}(A) - \phi_{k,\gamma}(B)\| \le \|A^k - B^k\| \|(I-\gamma B)^{-1}\| + \|B^k\| \|(I-\gamma A)^{-1} - (I-\gamma B)^{-1}\|$$
$$+ \|A^k - B^k\| \|(I-\gamma A)^{-1} - (I-\gamma B)^{-1}\| \tag{63}$$

so it suffices essentially to consider the case of powers and successor measure separately. By the telescopic sum formula (61)

$$\|A^k - B^k\| \le \sum_{i=1}^k \|A\|^{i-1} \|A - B\| \|B\|^{k-i}$$
$$= \|A - B\| \|B\|^{k-1} \frac{\left(\frac{\|A\|}{\|B\|}\right)^k - 1}{\frac{\|A\|}{\|B\|} - 1}.$$

Next we use that $\|B\| \le \|A\|$ with mean value inequality and the inequality $1 + x \le e^x$ to bound

$$\frac{\left(\frac{\|A\|}{\|B\|}\right)^k - 1}{\frac{\|A\|}{\|B\|} - 1} \le k\left(\frac{\|A\|}{\|B\|}\right)^{k-1}$$
$$= k\left(1 + \frac{\|A - B\|}{\|B\|}\right)^{k-1}$$
$$\le k e^{(k-1)\frac{\|A-B\|}{\|B\|}}.$$

Supposing now $\|A - B\| \le \|B\|/k$ the exponential term is bounded by 3.

On the other hand the resolvent identity (62) implies

$$\|(I-\gamma A)^{-1} - (I-\gamma B)^{-1}\| \le \gamma \|(I-\gamma A)^{-1}\| \|A - B\| \|(I-\gamma B)^{-1}\|$$
$$\le \frac{\gamma\|A - B\|}{(1-\gamma\|B\|)(1-\gamma(\|B\| + \|A-B\|).)}$$

Using the assumption that $\|A - B\| \le \frac{1-\gamma\|B\|}{2}$ the right hand side is bounded by $\frac{2\gamma\|A-B\|}{(1-\gamma\|B\|)^2}$. Plugging the previous bounds in (63) we deduce

$$\|\phi_{k,\gamma}(A) - \phi_{k,\gamma}(B)\| \le \frac{3k\|B\|^{k-1}\|A-B\|}{1-\gamma\|B\|} + \frac{2\|B\|^k\|A-B\|}{(1-\gamma\|B\|)^2} + \frac{6k\|B\|^{k-1}\|A-B\|^2}{(1-\gamma\|B\|)^2}$$
$$\le \frac{\|B\|^k\|A-B\|}{1-\gamma\|B\|}\left(3d + \frac{2}{1-\gamma\|B\|} + \frac{6d\|A-B\|}{1-\gamma\|B\|}\right)$$

which gives the result after using again $\|A - B\| \le \frac{1-\gamma\|B\|}{2}$. $\qquad\square$

*Proof of Theorem 7.* Let $\nu$ be a probability measure on $\mathcal{X}$, which can always be written as $\nu =: \mu \rtimes \pi$ for some probability measure $\mu$ on $\mathcal{S}$ and a policy $\pi$. Apply Theorem 8 to $P \in \mathbb{R}^{\mathcal{X} \times \mathcal{S}}$ to obtain

$$\mathbb{P}\Big[ \big\| \widehat{P} - P \big\|_{\ell^2(\mu), \ell^2(\nu)} \geq t \Big] \leq 4n \exp\left( \frac{-t^2 \min_{(s,a) \sim s'} \frac{Z_{s,a}\mu(s')}{\nu(s,a)+\mu(s')}}{8(t + 2\,\|P^\dagger\|_{\infty,\infty})} \right).$$

Then Point (iii) of Lemma 13 shows $\big\| P^\dagger \big\|_{\infty,\infty} = \big\| P_\pi^\dagger \big\| = 1$ if $\nu$ is supposed invariant. Then from (60) and Point (ii) of the lemma

$$\begin{aligned}
\big\| \widehat{P}_\pi - P_\pi \big\|_{\ell^2(\nu),\ell^2(\nu)} &= \big\| (\widehat{P} - P)K_\pi \big\|_{\ell^2(\nu),\ell^2(\nu)} \\
&\leq \big\| \widehat{P} - P \big\|_{\ell^2(\mu),\ell^2(\nu)} \|K_\pi\|_{\ell^2(\nu),\ell^2(\mu)} \\
&\leq \big\| \widehat{P} - P \big\|_{\ell^2(\mu),\ell^2(\nu)}
\end{aligned}$$

thus the concentration of $\widehat{P}$ immediately transfers to $\widehat{P}_\pi$. Finally we deduce the concentration of $\widehat{M}_{\pi,k}$ from the deterministic bound of Lemma 14. Supposing $\nu$ invariant also implies $\|P\| = 1$. Thus for $t \leq 1$ if $\big\| \widehat{P} - P \big\| < t/C_{k,\gamma} \leq 1/C_{k,\gamma}$ the conditions of Lemma 14 are satisfied, which thus implies $\big\| \widehat{M}_{\pi,k} - M_{\pi,k} \big\| \leq C_{k,\gamma} \big\| \widehat{P} - P \big\| < t$. Therefore the events $\{ \big\| \widehat{P} - P \big\| < t/C_{k,\gamma} \}$ and $\{ \big\| \widehat{M}_{\pi,k} - M_{\pi,k} \big\| \geq t \}$ are disjoint. $\qquad\square$

### F.3 Leave-one-out concentration

We now establish the technical concentration inequalities of Lemma 7. The proof strategy is similar to that of Theorem 7: we first establish concentration for linear functional of $\widehat{P}$ in the following proposition, to combine them with the contraction properties of Lemma 13 and the identities (61) and (62).

**Proposition 10.** *Consider the same setting as in Theorem 8. Let $A \in \mathbb{R}^{m \times p}$ and let $\rho$ be a probability measure on $[p]$. For all $l \in [n]$, and $t \geq 0$*

*(i)* $\mathbb{P}\Big[ \big\| \widehat{P}(l,\cdot)A - \widehat{P}^{(l)}(l,\cdot)A \big\|_{\ell^2(\rho)} \geq t \ \Big| \ (Y_{i\cdot})_{i \neq l} \Big] \leq (p+1)\exp\left( \frac{-t^2 Z_l}{2\|A\|_{\ell^2(\rho),\ell^\infty}^2} \right),$

*(ii)* $\mathbb{P}\Big[ \big\| \widehat{P}A - PA \big\|_{\ell^2(\rho),\ell^\infty} \geq t \Big] \leq n(p+1)\exp\left( \frac{-t^2 Z_{\min}}{2\|A\|_{\ell^2(\rho),\ell^\infty}^2} \right),$

*Proof.* For (i), fix $l \in [n]$ and $\phi(Y) := \widehat{P}(l,\cdot)A \in \mathbb{R}^{1 \times p}$. Since we reason conditional on $(Y_{i\cdot})_{i \neq l}$, $\phi$ is in fact here a function of the multinomial variable $(Y_{lj})_{j \in [n]}$ only, so Proposition 9 applies with $I = l$ a.s., and $N$ replaced with $Z_l$ here. We can then bound

$$\|\Delta_{lJ}\phi\|_{\ell^2(\rho)}^2 = \frac{\|A(J,\cdot)\|_{\ell^2(\rho)}^2}{Z_l^2} \leq \frac{\|A\|_{\ell^2(\rho),\ell^\infty}^2}{Z_l^2}, \qquad \|\Delta_{lK}\phi\|_{\ell^2(\rho)}^2 \leq \frac{\|A\|_{\ell^2(\rho),\ell^\infty}^2}{Z_l^2},$$

so applying the point (i) of Proposition 9 gives (i). For (ii), noting that $\big\| \widehat{P}^d A - P^d A \big\|_{\ell^2(\rho),\ell^\infty} :=$ $\max_{l \in [n]} \big\| (\widehat{P}(l,\cdot) - P(l,\cdot)A) \big\|_{\ell^2(\rho)}$, it suffices to prove concentration of the latter row matrix for fixed $l$, which can be done as above, and use a union bound argument. The only difference lies in that we do not reason conditional on $(Y_{i\cdot})_{i \neq l}$ anymore, so now we bound

$$\begin{aligned}
\mathbb{E}_{I,J}\Big[ \|\Delta_{IJ}\phi\|_{\ell^2(\rho)}^2 \Big] &= \frac{1}{N} \sum_{i,j \in [n]} Z_i \widehat{P}(i,j)\, \|\Delta_{ij}\phi\|_{\ell^2(\rho)}^2 \\
&= \frac{1}{N} \sum_{i,j \in [n]} \frac{\widehat{P}(i,j)}{Z_i} \mathbb{1}_{i=l}\, \|A(j,\cdot)\|_{\ell^2(\rho)}^2 \\
&\leq \frac{\|A\|_{\ell^2(\rho),\ell^\infty}^2}{N Z_l} \leq \frac{\|A\|_{\ell^2(\rho),\ell^\infty}^2}{N Z_{\min}}
\end{aligned}$$

and similarly for $\mathbb{E}_{I,K}\|\Delta_{IK}\phi\|_{\ell^2(\rho)}^2$. Applying point (i) of Proposition 9 gives (ii). $\qquad\square$

*Proof of Lemma 7.* We first by claim that the result of Proposition 10 also applies with $P_\pi, \widehat{P}_\pi$ in place of $P$ and $\widehat{P}$. The latter proved two inequalities of the form $\mathbb{P}\left[\left\|(\widehat{P}-P)A\right\| \geq t\right] \leq f_t(\|A\|_{2,\infty})$ where $f_t$ is a non-decreasing function. From (60) we can bound

$$
\begin{aligned}
\mathbb{P}\left[\left\|(\widehat{P}_\pi - P_\pi)A\right\| \geq t\right] &= \mathbb{P}\left[\left\|(\widehat{P}-P)K_\pi A\right\| \geq t\right] \\
&\leq f_t(\|K_\pi A\|_{2,\infty}) \\
&\leq f_t(\|K_\pi\|_{\infty,\infty} \|A\|_{2,\infty}) \\
&= f_t(\|A\|_{2,\infty})
\end{aligned}
$$

using the fact that $f_t$ is non-decreasing and Point (i) of Lemma 13. This proves the claim. We will thus apply Proposition 10 as if it applied directly to $P_\pi$. For simplicity we omit the subscript for the rest of the proof, writing $P$ in place of $P_\pi$.

The proof of 24 is similar to that of Theorem 8. For other points, we also start by decomposing

$$
\widehat{M}_{\pi,k}A - \widehat{M}^{(l)}_{\pi,k}A = \left[\widehat{P}^k - (\widehat{P}^{(l)})^k\right](I - \gamma\widehat{P}^{(l)})^{-1}A + (\widehat{P}^{(l)})^k\left[(I - \gamma\widehat{P})^{-1} - (I - \gamma\widehat{P}^{(l)})^{-1}\right]A
$$
$$
+ \left[\widehat{P}^k - (\widehat{P}^{(l)})^k\right]\left[(I - \gamma\widehat{P})^{-1} - (I - \gamma\widehat{P}^{(l)})^{-1}\right]A. \quad (64)
$$

Let $B := (I - \gamma\widehat{P}^{(l)})^{-1}A$. By the telescopic sum formula (61) the first term can be bounded as

$$
\begin{aligned}
\left\|\left[\widehat{P}^k(l,\cdot) - (\widehat{P}^{(l)})^k(l,\cdot)\right]B\right\|_{\ell^2(\rho)} &\leq \sum_{i=1}^{k}\left\|\widehat{P}^{i-1}(l,\cdot)(\widehat{P} - \widehat{P}^{(l)})(\widehat{P}^{(l)})^{k-i}B\right\|_{\ell^2(\rho)} \\
&= \sum_{i=1}^{k}\left|\widehat{P}^{i-1}(l,l)\right|\left\|(\widehat{P} - \widehat{P}^{(l)})(l,\cdot)(\widehat{P}^{(l)})^{k-i}B\right\|_{\ell^2(\rho)}
\end{aligned}
$$

as $(\widehat{P} - \widehat{P}^{(l)})(j,\cdot) = 0$ if $j \neq l$. Now observe the matrix $(\widehat{P}^{(l)})^{k-i}B$ is independent of $Y_l$. so by point (i) of Proposition 10 and union bound the probability conditional on $(Y_{i\cdot})_{i\neq l}$ that one norm factor in the above sum is larger than $t$ is at most $k(p+1)\max_{i\in[k]}\exp\left(\frac{-t^2 Z_l}{2\|(\widehat{P}^{(l)})^{k-i}B\|^2_{\ell^2(\rho),\ell^\infty}}\right)$. However for all $i \in [k]$

$$
\left\|(\widehat{P}^{(l)})^{k-i}B\right\|^2_{\ell^2(\rho),\ell^\infty} \leq \left\|(\widehat{P}^{(l)})^{k-i}(I - \gamma\widehat{P}^{(l)})^{-1}\right\|_{\infty,\infty}\|A\|^2_{\ell^2(\rho),\ell^\infty} = \frac{\|A\|^2_{\ell^2(\rho),\ell^\infty}}{1-\gamma}
$$

as $(1-\gamma)(\widehat{P}^{(l)})^{k-i}(I - \gamma\widehat{P}^{(l)})^{-1}$ is a stochastic matrix. Bounding also $\left|\widehat{P}^{i-1}(l,l)\right| \leq 1$ we get eventually that

$$
\mathbb{P}\left[\left\|\left[\widehat{P}^k(l,\cdot) - (\widehat{P}^{(l)})^k(l,\cdot)\right]B\right\|_{\ell^2(\rho)} \geq t \;\middle|\; (Y_{i\cdot})_{i\neq l}\right] \leq k(p+1)\exp\left(\frac{-t^2(1-\gamma)^2 Z_l}{2k^2\|A\|^2_{\ell^2(\rho),\ell^\infty}}\right).
$$

For the second term of (64), the resolvent identity (62) gives

$$
\begin{aligned}
\left\|(\widehat{P}^{(l)})^k\left[(I - \gamma\widehat{P})^{-1} - (I - \gamma\widehat{P}^{(l)})^{-1}\right](l,\cdot)A\right\| &= \gamma\left|(\widehat{P}^{(l)})^k(I - \gamma\widehat{P})^{-1}(l,l)\right|\left\|(\widehat{P} - \widehat{P}^{(l)})(l,\cdot)B\right\| \\
&\leq \frac{1}{1-\gamma}\left\|(\widehat{P} - \widehat{P}^{(l)})(l,\cdot)B\right\|.
\end{aligned}
$$

Thus with the same arguments as above point (i) of Proposition 10 shows

$$
\mathbb{P}\left[\left\|(\widehat{P}^{(l)})^k\left[(I - \gamma\widehat{P})^{-1} - (I - \gamma\widehat{P}^{(l)})^{-1}\right](l,\cdot)A\right\| \geq t \;\middle|\; (Y_{i\cdot})_{i\neq l}\right] \leq (p+1)\exp\left(\frac{-t^2(1-\gamma)^4 Z_l}{2\|A\|^2_{2,\infty}}\right).
$$

Finally the third term of (64) can be bounded as

$$
\begin{aligned}
\left\|\left[\widehat{P}^k - (\widehat{P}^{(l)})^k\right](l,\cdot)\left[(I - \gamma\widehat{P})^{-1} - (I - \gamma\widehat{P}^{(l)})^{-1}\right]A\right\| &\\
= \left\|\left[(\widehat{P}^k - (\widehat{P}^{(l)})^k)(I - \gamma\widehat{P})^{-1}\right](l,l)\right|\left\|(\widehat{P} - \widehat{P}^{(l)})(l,\cdot)B\right\| &\\
\leq \frac{2}{1-\gamma}\left\|(\widehat{P} - \widehat{P}^{(l)})(l,\cdot)B\right\| &
\end{aligned}
$$

and is thus controlled as the second term. Combining all three bounds yields 21.

The proof of (22) follows similar arguments, using point( ii) of Proposition 10 instead.

Finally for inequality (23) the proof is almost the same except there is no multiplication by $\mathbb{1}_l^\top$ on the left, so the factors $\left|\widehat{P}^{i-1}(l,l)\right|$, $\left|(\widehat{P}^{(l)})^k(I-\gamma\widehat{P})^{-1}(l,l)\right|$ and $\left\|\left[(\widehat{P}^k-(\widehat{P}^{(l)})^k)(I-\gamma\widehat{P})^{-1}\right](l,l)\right\|$ need to be replaced with $\left\|\widehat{P}^{i-1}(\cdot,l)\right\|_{\ell^2(\nu)}$, $\left\|(\widehat{P}^{(l)})^k(I-\gamma\widehat{P})^{-1}(\cdot,l)\right\|_{\ell^2(\nu)}$ and $\left\|\left[(\widehat{P}^k-(\widehat{P}^{(l)})^k)(I-\gamma\widehat{P})^{-1}\right](\cdot,l)\right\|_{\ell^2(\nu)}$ respectively. We bound these terms as

$$\left\|\widehat{P}^{i-1}(\cdot,l)\right\|_{\ell^2(\nu)} = \left\|\widehat{P}^{i-1}\mathbb{1}_l\right\|_{\ell^2(\nu)} \leq \left\|\widehat{P}^{i-1}\right\|_{\ell^2(\nu),\ell^2(\nu)}\left\|\mathbb{1}_l\right\|_{\ell^2(\nu)} = \left\|\widehat{P}^{i-1}\right\|_{\ell^2(\nu),\ell^2(\nu)}\sqrt{\nu(l)}$$

$$\left\|(\widehat{P}^{(l)})^k(I-\gamma\widehat{P})^{-1}(\cdot,l)\right\|_{\ell^2(\nu)} \leq \left\|(\widehat{P}^{(l)})^k(I-\gamma\widehat{P})^{-1}\right\|_{\ell^2(\nu),\ell^2(\nu)}\sqrt{\nu(l)}$$

and

$$\left\|\left[(\widehat{P}^k-(\widehat{P}^{(l)})^k)(I-\gamma\widehat{P})^{-1}\right](\cdot,l)\right\|_{\ell^2(\nu)} \leq \left(\left\|\widehat{P}^k(I-\gamma\widehat{P})^{-1}\right\|_{\ell^2(\nu),\ell^2(\nu)} \right. $$
$$\left. + \left\|(\widehat{P}^{(l)})^k(I-\gamma\widehat{P})^{-1}\right\|_{\ell^2(\nu),\ell^2(\nu)}\right)\sqrt{\nu(l)}$$

Suppose now that the terms of the right-hand side concentrate: then for some constant $C > 0$, using that $\nu$ is invariant we would get $\left\|\widehat{P}^i\right\|_{\ell^2(\nu),\ell^2(\nu)} \leq C\left\|P^i\right\|_{\ell^2(\nu),\ell^2(\nu)} \leq C$ for all $i \in [k]$, $\left\|(\widehat{P}^{(l)})^k(I-\gamma\widehat{P})^{-1}\right\|_{\ell^2(\nu),\ell^2(\nu)} \leq C\left\|M_{\pi,i}\right\|_{\ell^2(\nu)} \leq C/(1-\gamma)$ and $\left\|\widehat{P}^k(I-\gamma\widehat{P})^{-1}\right\|_{\ell^2(\nu)} \leq C/(1-\gamma)$. Then on this event reiterating the above argument would eventually give the bounds

$$\left\|\left[\widehat{P}^k-(\widehat{P}^{(l)})^k\right]B\right\|_{\ell^2(\nu),\ell^2(\rho)} \leq C\sqrt{\nu(l)}kt,$$

$$\left\|(\widehat{P}^{(l)})^k\left[(I-\gamma\widehat{P})^{-1}-(I-\gamma\widehat{P}^{(l)})^{-1}\right]A\right\|_{\ell^2(\nu),\ell^2(\rho)} \leq \frac{C\sqrt{\nu(l)}t}{(1-\gamma)^2}$$

$$\left\|\left[\widehat{P}^k-(\widehat{P}^{(l)})^k\right]\left[(I-\gamma\widehat{P})^{-1}-(I-\gamma\widehat{P}^{(l)})^{-1}\right]A\right\|_{\ell^2(\nu),\ell^2(\rho)} \leq \frac{2C\sqrt{\nu(l)}t}{(1-\gamma)^2}$$

with probability at least $1 - (k+2)(p+1)\exp\left(\frac{-t^2(1-\gamma)^2 Z_l}{2\max(k,(1-\gamma)^{-1})^2\|A\|_{\ell^2(\rho),\ell^\infty}^2}\right)$, conditional on $(Y_{i\cdot})_{i\neq l}$ and thus also unconditional. We then deduce

$$\mathbb{P}\left[\left\|\widehat{M}_{\pi,k}A-\widehat{M}_{\pi,k}^{(l)}A\right\| \geq t\right]_{\ell^2(\nu),\ell^2(\rho)} \leq (k+2)(p+1)\exp\left(\frac{-t^2 Z_l}{2C_{k,\gamma}^2\nu(l)\|A\|_{\ell^2(\rho),\ell^\infty}^2}\right)$$
$$+ \mathbb{P}\left[\exists i \in [k] : \left\|\widehat{P}^{i-1}\right\|_{\ell^2(\nu),\ell^2(\nu)} > C\right]$$
$$+ \mathbb{P}\left[\left\|(\widehat{P}^{(l)})^k(I-\gamma\widehat{P})^{-1}\right\|_{\ell^2(\nu),\ell^2(\nu)} > C/(1-\gamma)\right]$$
$$+ \mathbb{P}\left[\left\|\widehat{P}^k(I-\gamma\widehat{P})^{-1}\right\|_{\ell^2(\nu),\ell^2(\nu)} > C/(1-\gamma)\right].$$

The three remaining terms can be controlled by Theorem 7 and (24), which gives the second term in (23). We omit the details. $\qquad\square$

# G   Local mixing phenomena

In this appendix we prove the results of Section 5. We start by proving Proposition 1 in §G.1. In §G.2, we explain how bounding the spectral recoverabilty reduces to bounding the $2 - \infty$ norm, at least for *normal* chains. Then in §G.3, we give a detailed background on functional inequalities for Markov chains and explain how our results differ from the classical analysis of mixing times. Finally, in §G.4, we extend these inequalities and prove Theorems 2, 3, 4 and Proposition 2.

## G.1   Singular value bound: proof of Proposition 1

Proposition 1 will be a straightforward application of the following, more general result. Here we consider the norms, singular values, etc. to be defined w.r.t. any probability measure.

**Proposition 11.** *Let $A \in \mathbb{R}^{n \times m}$. For all $\gamma \in (0, \|A\|_{2,2}^{-1}), k \geq 0, i \in [n]$,*

$$\frac{\sigma_i(A^k)}{1 + \gamma \|A\|_{2,2}} \leq \sigma_i\left(A^k(I - \gamma A)^{-1}\right) \leq \frac{\sigma_i(A^k)}{1 - \gamma \|A\|_{2,2}} \tag{65}$$

*Consequently*

$$\left\|\left(\sum_{t \geq k} \gamma^t A^t\right)\right\|_{2,\infty} \geq \frac{\|A^k\|_F}{1 + \gamma \|A\|_{2,2}}. \tag{66}$$

*Proof.* Using the classical inequality for singular values $\sigma_n(A)\sigma_i(B) \leq \sigma_i(AB) \leq \sigma_1(A)\sigma_i(B)$ (see e.g. (27)) valid for all matrices $A, B$ and $i$, we get

$$\sigma_n((I - \gamma A)^{-1})\sigma_i(A^k) \leq \sigma_i\left(A^k(I - \gamma A)^{-1}\right) \leq \sigma_1((I - \gamma A)^{-1})\sigma_i(A^k).$$

Now simply notice $\sigma_1((I - \gamma A)^{-1}) = \|(I - \gamma A)^{-1}\|_{2,2} \leq (1 - \gamma \|A\|_{2,2})^{-1}$ and $\sigma_1(I - \gamma A) = \|I - \gamma A\|_{2,2} \leq 1 + \gamma \|A\|_{2,2}$, hence

$$\sigma_n\left((I - \gamma A)^{-1}\right) = \frac{1}{\sigma_1(I - \gamma A)} \geq \frac{1}{1 + \gamma \|A\|_{2,2}}.$$

Summing over $i$ and using (13) yields

$$\left\|\left(\sum_{t \geq k} \gamma^t A^t\right)\right\|_{2,\infty} \geq \left\|\left(\sum_{t \geq k} \gamma^t A^t\right)\right\|_F \geq \frac{\|A^k\|_F}{1 + \gamma \|A\|}.$$

$\square$

*Proof of Proposition 1.* Apply the previous Proposition with $A = P_\pi$ and note that if the underlying probability measure is invariant then $\|P_\pi\|_{2,2} = 1$. $\square$

## G.2   Spectral recoverability for chains with normal transition matrices

From Definition 3 and (12), for any matrix $A$ with SVD $A = U\Sigma V^\dagger$, we can express $\xi(A) = \||A|^{1/2}\|_{2,\infty}^2$ where the absolute square root is defined by $|A|^{1/2} := U\Sigma^{1/2}V^\dagger$. When $A = P^{2k}$ is an even power of $P$, it is thus tempting to try relating $\xi(P^{2k})$ with $\|P^k\|_{2,\infty}^2$. However we do not know how to achieve this, as the singular vectors of $P^k$ and $P^{2k}$ may be very different. A case where this is possible is when we assume the chain to be reversible or more generally normal (12), in the sense that $PP^\dagger = P^\dagger P$. By the spectral theorem, such matrices are diagonalizable in orthonormal basis, making the singular vectors coincide with eigenvectors.

**Lemma 15.** *Suppose $PP^\dagger = P^\dagger P$. Then for all $k \geq 0$,*

$$\xi(P^{2k}) \leq \|P^k\|_{2,\infty}^2.$$

*Similarly $\xi(M_{2k}) \leq (1 - \gamma\sigma_1(P))^{-1} \|M_k\|_{2,\infty}^2$ where we recall $M_k := P^k(I - \gamma P)^{-1}$*

*Proof.* Let $P := \sum_i \sigma_i \psi_i \phi_i^\dagger$ be the SVD of $P$. By normality and the spectral theorem, the singular vectors coincide with eigenvectors, so the SVD of $P^k$ is $P^k = \sum_i \sigma_i^k \psi_i \phi_i^\dagger$ for all $k \geq 0$. Consequently

$$\xi(P^{2k}) = \max_x \sum_i \sigma_i(P^{2k})\psi_i(x)^2 = \max_x \sum_i \sigma_i(P^k)^2 \psi_i(x)^2 = \left\| P^k \right\|_{2,\infty}^2 .$$

For the shifted successor measure, the singular values of $M_k$ are $\sigma_i^k(1 - \gamma\sigma_i)^{-1}$ hence

$$\begin{aligned}
\xi(M_{2k}) &= \max_x \sum_i \sigma_i(P^k)^2 (1 - \sigma_i(P))^{-1} \psi_i(x)^2 \\
&\leq (1 - \gamma\sigma_1(P))^{-1} \max_x \sum_i \sigma_i(P^k)^2 \psi_i(x)^2 \\
&= (1 - \gamma\sigma_1(P))^{-1} \left\| P^k \right\|_{2,\infty}^2 .
\end{aligned}$$

$\square$

We leave as an open problem the question of how to extend this result to non-normal chains, but consider it as a heuristic proof that having $\xi(P^k)$ bounded should in general be essentially the same as having $\left\| P^k \right\|_{2,\infty}^2$ bounded, up to multiplying $k$ by 2.

### G.3   Functional inequalities for Markov chains

From now on, we consider $P \in \mathbb{R}^{n \times n}$ to be the transition matrix of an irreducible Markov chain with invariant measure $\nu$. Using the framework of A.1, the underlying measure will here be $\nu$ until further notice.

Identifying $\nu$ with a row vector, the rank one matrix $\mathbb{1}\nu$ is the matrix of the chain at stationarity, and it is readily seen from (11) that $\left\| P^t - \mathbb{1}\nu \right\|_{2,\infty} = \left\| P^t \right\|_{2,\infty} - 1$. It makes sense to define the $\ell^2$-mixing time as $t_2(\varepsilon) := \inf\{t \geq 0 : \left\| P^t - \mathbb{1}\nu \right\|_{2,\infty} \leq \varepsilon\}$, which may be infinite. We also write $\mathbb{E}_\nu[f] := \sum_x \nu(x)f(x)$ and $\mathrm{Var}_\nu(f) = \mathbb{E}_\nu[f^2] - \mathbb{E}_\nu[f]^2$.

Recall the definition of the Dirichlet form

$$\mathcal{E}_{PP^\dagger}(f, g) = \langle (I - P)f, g \rangle_\nu . \tag{67}$$

*Remark* 1.  We consider the Dirichlet form of the multiplicative reversibilization $PP^\dagger$, which appears naturally when working with discrete-time Markov chains (22). The arguments that follow also extend, and in fact are simpler, for continuous-time Markov chains, for which we can directly work with $P$. We refer to (49) for a comprehensive reference. It is also possible to reduce to considerations on $P$ only with laziness, i.e. if the chain has a uniformly lower bounded probability to stay put. If $P(x, x) \geq \alpha$ for all $x \in [n]$, (49, Equation (1.12)) shows that $\mathcal{E}_{PP^\dagger} \geq 2\alpha\mathcal{E}_P(f, f)$.

The argument behind the use of functional inequalities is as follows: by duality (9), $\left\| P^t \right\|_{2,\infty} = \left\| (P^t)^\dagger \right\|_{1,2} = \sup_{\|f\|_1 = 1} \left\| (P^t)^\dagger f \right\|_2$. Therefore it suffices to bound $\left\| (P^t)^\dagger f \right\|_2$ for all $f \in \mathbb{R}^n$. Now for fixed $f$, it is easy to compute

$$\left\| (P^t)^\dagger f \right\|_2^2 - \left\| (P^{t-1})^\dagger f \right\|_2^2 = -\mathcal{E}_{PP^\dagger}((P^{t-1})^\dagger f, (P^{t-1})^\dagger f). \tag{68}$$

(This is really a discrete counterpart of differentiating $\left\| P^t f \right\|_2$). The goal of using functional inequalities is thus to obtain a lower bound $\mathcal{E}_{P^\dagger P}(g, g) \geq F(\|g\|_2^2)$ valid for all $g$ such that $\|g\|_1 = 1$, that can be "integrated" to get estimates on $\left\| P^t f \right\|_2$ and eventually on $\left\| P^t \right\|_{2,\infty}$. The most classical inequalities are Poincaré (22), log-Sobolev (16) and Nash inequalities (17), to which we can also add the spectral profile technique, which stems from Faber-Krahn inequalities (24). We focus in this paper on Poincaré, which are the simplest to establish, and Nash inequalities, which served as our main inspiration and can prove complementary to Poincaré inequalities.

**Poincaré inequality:** the classical Poincaré inequality takes the form

$$\forall f \in \mathbb{R}^n : \quad \lambda \operatorname{Var}_\nu(f) \le \mathcal{E}_{PP^\dagger}(f, f), \tag{69}$$

for some constant $\lambda \ge 0$. Plugged in (68) and applying the above argument, it implies the decay rate $\left\| P^t - \mathbb{1}\nu \right\|_{2,\infty} \le (1-\lambda)^t \nu_{\min}^{-1}$ (see Corollary 1.14 of (49)). This gives in particular a bound on the mixing time:

$$t_2(\varepsilon) \le \lambda^{-1} \log(\nu_{\min}^{-1} \varepsilon^{-1}). \tag{70}$$

For our purpose of applying Theorem 1, we do not require that strong mixing estimates: we could be content with $\left\| P^t \right\|_{2,\infty} = O(1)$, which could occur on time scales much smaller than the mixing time. The Nash inequalities of (17) were introduced precisely to get such decay rates, when the Poincaré inequality alone is not sharp. Nash inequalities are however notoriously difficult to establish.

**Nash inequalities:** in view of (54), we distinguish two types of Nash inequalities, which we call type I and type II

- Type I reads

$$\operatorname{Var}_\nu(f)^{1+2/d} \le C \mathcal{E}_{PP^\dagger}(f, f) \|f\|_1^{4/d} \tag{71}$$

  for some constants $C, d > 0$. Plugged in (68) and applying Lemma 3.1 of (17) yields the bound

$$\left\| P^k - \mathbb{1}\nu \right\|_{2,\infty}^2 \le \left( \frac{C(1+\lceil d \rceil)}{k+1} \right)^{d/2}$$

  which in turn gives the mixing time bound $t_2(\varepsilon) \le \frac{C(1+\lceil d \rceil)}{\varepsilon^{2/d}}$.

  Using Jensen's inequality, we also see that (71) implies a Poincaré inequality $\operatorname{Var}_\nu(f) \le C \mathcal{E}_P(f, f)$. Thus Nash inequality can be combined or used in place of the Poincaré inequality to get rid of the $\log(\nu_{\min}^{-1})$ factor in (70). This is generally sharp for "low-dimensional chains" like random walk on grids, where the constant $d$ that appears in the Nash inequality coincides with the dimension parameter.

- Type II has the form

$$\|f\|_2^{2(2+2/d)} \le C \left( \mathcal{E}_{PP^\dagger}(f, f) + \frac{1}{T} \|f\|_2^2 \right) \|f\|_1^{4/d} \tag{72}$$

  for some constant $C, d, T > 0$. Theorem 3.1 and Remark 3.1 of (17) show that this implies the decay

$$\forall k \in [0, T]: \quad \left\| P^t \right\|_{2,\infty}^2 \le \left( \frac{C(1+1/T)(1+\lceil d \rceil)}{k+1} \right)^{d/2}.$$

  Unlike the type I inequality, (72) implies no Poincaré inequality and no mixing time estimate. Note also that by moving the expectation term of $\operatorname{Var}_\nu(f)$ to the right hand side, a type I inequality (71) implies a type II inequality with a slightly worse constant $C$ and $T = 1/C$.

## G.4 Type II Poincaré inequalities and applications

### G.4.1 Proofs of Theorems 2 and 3

As seen above, Nash inequalities, when they can be established at all, provide only a polynomial decay of the $2 - \infty$ norm. To obtain an exponential decay, we consider extending Poincaré inequalities instead. The clear analogy between (71) and (69) motivated us to develop analogous "type II" versions of the Poincaré inequality, that incorporate an additive $\ell^1$ term. This is exactly the result of Theorem 2, which we now prove.

*Proof of Theorem 2.* We use the argument sketched in the previous section. Let $f \in \mathbb{R}^n$ be such that $\|f\|_1 = 1$ and set $u_t := \left\| (P^t)^\dagger f \right\|_2^2$. Note that $\left\| (P^t)^\dagger f \right\|_1 \le \left\| (P^t)^\dagger \right\|_{1,1} \|f\|_1$, however by duality (9) $\left\| P^\dagger \right\|_{1,1} = \|P\|_{\infty,\infty} = 1$. Thus $\left\| (P^t)^\dagger f \right\|_1 \le 1$ for all $t \ge 0$. Consequently, the type II inequality (5) plugged in (68) yields

$$u_t - u_{t-1} \le -\lambda u_{t-1} + \lambda C$$

which in turn gives $u_t = \left\| (P^t)^\dagger f \right\|_2 \le (1-\lambda)^t (u_0 - C) + C$ by an easy induction. Then remark that $u_0 = \|f\|_2^2 \le \nu_{\min}^{-1} \|f\|_1^2 = \nu_{\min}^{-1}$. Since this is valid for all $f$ such that $\|f\|_1 = 1$ we deduce $\left\| (P^t)^\dagger \right\|_{1,2} = \left\| P^t \right\|_{2,\infty} \le (1-\lambda)^t (\nu_{\min}^{-1} - C) + C$. $\qquad\square$

*Remark* 2. The same arguments could be applied by exchanging $P$ and $P^\dagger$ to give a similar bound for $\left\| (P^k)^\dagger \right\|_{2,\infty}$, as is required for Theorem 1. There is one difference however in that we need the invariance of $\nu$ to have $\left\| P^\dagger \right\|_{\infty,\infty} = 1$.

*Proof of Theorem 3.* Let $f \in \mathbb{R}^n$ and write $f_r := U_r U_r^\dagger f$ for its projection onto the $r$ first singular vectors. Note that $\mathcal{E}_{PP^\dagger}(f - f_r, f_r) = 0$ and hence

$$\mathcal{E}_{PP^\dagger}(f,f) = \mathcal{E}_{PP^\dagger}(f_r, f_r) + \mathcal{E}_{PP^\dagger}(f - f_r, f - f_r).$$

If the underlying measure is invariant, $PP^\dagger$ is a stochastic matrix so Lemma 12 implies that $I - PP^\dagger \ge 0$ and thus $\mathcal{E}_{PP^\dagger}(f,f) \ge \mathcal{E}_{PP^\dagger}(f - f_r, f - f_r)$. Thus the Courant-Fischer theorem (27, Theorem 3.1.2) gives

$$\lambda_{r+1} \|f - f_r\|_2^2 \le \mathcal{E}_{PP^\dagger}(f - f_r, f - f_r) \le \mathcal{E}_{PP^\dagger}(f,f)$$

where we write $\lambda_{r+1} = 1 - \sigma_{r+1}^2$. On the other hand use Hölder's inequality to bound

$$\|f\|_2^2 = \langle f - f_r, f \rangle + \langle f_r, f \rangle \le \|f - f_r\|_2 \|f\|_2 + \|f_r\|_\infty \|f\|_1.$$

Observe then that

$$\|f_r\|_\infty = \left\| U_r U_r^\dagger f \right\|_\infty \le \|U_r\|_{2,\infty} \|f\|_2,$$

so after simplifying by $\|f\|_2$, we deduce

$$\lambda_{r+1}^{1/2} \|f\|_2 \le \mathcal{E}_{PP^\dagger}(f,f)^{1/2} + \lambda_{r+1}^{1/2} \|U_r\|_{2,\infty} \|f\|_1.$$

Using $(a+b)^2 \le 2(a^2 + b^2)$, we finally get

$$\frac{\lambda_{r+1}}{2} \|f\|_2^2 \le \mathcal{E}_{PP^\dagger}(f,f) + \lambda_{r+1} \|U_r\|_{2,\infty}^2 \|f\|_1^2.$$

$\qquad\square$

### G.4.2 Combining inequalities of induced chains

In (17), Diaconis and Saloff-Coste showed how to establish type II Nash inequalities from local Poincaré inequalities. This suggested that type II inequalities are related to a local notion of mixing, which we establish formally in Proposition 2. Given the definition of induced chains (Definition 4) it is immediate that for all $f \in \mathbb{R}^n$

$$\begin{aligned}
\mathcal{E}_{\nu,P}(f,f) &\ge \mathcal{E}_{\nu,P_S}(f,f) + \mathcal{E}_{\nu,P_{S^c}}(f,f) \\
&= \nu(S)\mathcal{E}_{\nu_S,P_S}(f_{|S}, f_{|S}) + \nu(S^c)\mathcal{E}_{\nu_{S^c},P_{S^c}}(f_{|S^c}, f_{|S^c}).
\end{aligned} \tag{73}$$

On the other hand it is also straightforward that $\|f\|_{\ell^p(\nu)}^p = \nu(S) \left\| f_{|S} \right\|_{\ell^p(\nu_S)}^p + \nu(S^c) \left\| f_{|S^c} \right\|_{\ell^p(\nu_{S^c})}^p$ for all $p \in [1,\infty)$. Our decomposition result is based on these two simple facts.

*Proof of Proposition 2.* The result is a consequence of the following inequalities:

$$\begin{aligned}
\|f\|_{\ell^2(\nu)}^2 &= \nu(S) \left\| f_{|S} \right\|_{\ell^2(\nu_S)}^2 + \nu(S^c) \left\| f_{|S^c} \right\|_{\ell^2(\nu_{S^c})}^2 \\
&\le \nu(S) \left[ \lambda_S^{-1} \mathcal{E}_{\nu_S,P_S}(f_{|S}, f_{|S}) + C_S \left\| f_{|S} \right\|_{\ell^1(\nu_S)}^2 \right] \\
&\quad + \nu(S^c) \left[ \lambda_{S^c}^{-1} \mathcal{E}_{\nu_{S^c},P_{S^c}}(f_{|S^c}, f_{|S^c}) + C_{S^c} \left\| f_{|S^c} \right\|_{\ell^1(\nu_{S^c})}^2 \right] \\
&\le \min(\lambda_S, \lambda_{S^c})^{-1} \mathcal{E}_{\nu,P}(f,f) + \max\left( \frac{C_S}{\nu(S)}, \frac{C_{S^c}}{\nu(S^c)} \right) \left( \left\| f_{|S} \right\|_{\ell^1(\nu)}^2 + \left\| f_{|S^c} \right\|_{\ell^1(\nu)}^2 \right) \\
&\le \min(\lambda_S, \lambda_{S^c})^{-1} \mathcal{E}_{\nu,P}(f,f) + \max\left( \frac{C_S}{\nu(S)}, \frac{C_{S^c}}{\nu(S^c)} \right) \|f\|_{\ell^1(\nu)}^2.
\end{aligned}$$

The equality uses that $S, S^c$ are disjoint, the first inequality comes from applying the Poincaré inequalities, the second uses (73) and $\nu(S) \|f_{|S}\|_{\ell^1(\nu_S)} = \|f_{|S}\|_{\ell^1(\nu)}$, the last inequality is a consequence of $a^2 + b^2 \leq (a+b)^2$ for $a, b \geq 0$. $\qquad\square$

Proposition 2 requires functional inequalities for induced chains, with respect to the induced measures. It is wrong in general that the induced measures are invariant for the induced chains, but it is true for reversible chains (34). For completeness, we prove it in the following Lemma, to justify the consideration of induced chains with induced measures. We recall a chain is reversible if it satisfies the detailed balance equation, which translates matricially into $P^\dagger = P$.

**Lemma 16.** *Suppose $P$ is a reversible Markov chain on $[n]$ with invariant measure $\nu$. Then for all subset $S \subset [n]$ the restriction $\nu_{|S}$ to $S$ is invariant for the induced chain $P_S$.*

*Proof.* $P$ is reversible if and only if it satisfies the detailed balanced equation $\nu(x)P(x,y) = \nu(y)P(y,x)$ for all $x, y \in [n]$. Taking the induced chain on $S$ does not affect the transition probabilities between $x \neq y$ in $S$, so the equation still holds for the induced chain and the measure induced by $\nu$. $\qquad\square$

### G.4.3 The 4-room examples: proof of Theorem 4

We now proceed to prove the bounds for the 4-room environment of Theorem 4.

*Proof of Theorem 4.* As a random walk on a graph $P$ is reversible with invariant measure being given by $\nu(x) = \deg(x)/\sum_y \deg(y)$, where $\deg(x)$ denotes the degree of $x$. Thus we need to consider the Dirichlet form of $PP^\dagger = P^2$. The latter is also reversible hence by Lemma 16, so are all induced chains $(P^2)_{|G_i}$ with the induced measures as invariant measures. Now for each $i \in [4]$, $(P^2)_{|G_i}$ satisfies a type II Poincaré inequality: namely for all $f \in \mathbb{R}^{G_i}$

$$\lambda_i \|f\|^2_{\ell^2(\nu_{V_i})} \leq \mathcal{E}_{(P^2)_{|G_i}}(f, f) + \lambda_i \|f\|^2_{\ell^1(\nu_{V_i})}$$

with $\lambda_i = 1 - \sigma_2((P^2)_{|G_i})$ the spectral gap of the p.s.d. matrix $(P^2)_{|G_i}$. This a consequence of the Courant-Fischer theorem as for Theorem 3. It is thus a simple application of Proposition 2 that the whole chain satisfies

$$(1 - \lambda)\|f\|^2_{\ell^2(\nu)} \leq \mathcal{E}_{P^2}(f, f) + \frac{1 - \lambda}{\min_i \nu(V_i)} \|f\|^2_{\ell^1(\nu)}.$$

with $\lambda := \min_i \lambda_i$, which by Theorem 2 implies the decay rate

$$\|P^t\|^2_{2,\infty} \leq (1 - \lambda)^t \nu^{-1}_{\min} + \frac{1}{\min_i \nu(V_i)}.$$

This proves the first part of the theorem.

For the second part, suppose that $\min_i \nu(G_i) \geq c$. Then for $t \geq \lambda^{-1} \log(\nu^{-1}_{\min}\varepsilon^{-1})$ we obtain $\|P^t\|^2_{2,\infty} \leq \varepsilon + c^{-1}$. Since the chain is reversible we can use Lemma 15 to bound the spectral recoverability as well and get $\xi(P^{2t}) \leq \varepsilon + c^{-1}$. Then Lemma 1 shows that $\|P^{2t} - [P^{2t}]_r\|_{2,\infty} \leq \varepsilon$ for the smallest $r$ such that $\sigma_{r+1}(P^{2t}) \leq \varepsilon^2/(c^{-1} + \varepsilon)$. We claim that:

$$\sigma_5(P^{2s}) \leq (1 - \lambda)^s, \quad \text{for all } s \geq 0. \tag{74}$$

Provided the claim holds, this implies that $\sigma_5(P^{4t}) \leq e^{-2\lambda t} \leq \nu^2_{\min}\varepsilon^2$ by the choice if $t$.

Let us prove the claim (74. Note that by reversibility $\sigma_5(P^{2t}) = \sigma_5(P^2)^t$ so it suffices to prove that $1 - \sigma_5(P^2) \geq \lambda$. From the Courant-Fischer theorem:

$$1 - \sigma_5(P^2) = \sup_{\text{codim } W = 4} \inf_{\substack{f \in W \\ f \neq 0}} \frac{\mathcal{E}_{P^2}(f, f)}{\|f\|^2_2}.$$

Let $W$ be the subspace orthogonal to the subspace $\mathrm{Span}(\mathbb{1}_{G_i}, i \in [4])$ spanned by the indicator of each subgraph. It has codimension $4$ so we can lower bound

$$1 - \sigma_5(P^2) \geq \inf_{\substack{f \in W \\ f \neq 0}} \frac{\mathcal{E}_{P^2}(f, f)}{\|f\|_2^2}$$

for this particular subspace. Now decompose $f = \sum_{i=1}^4 f_{|G_i}$. As in (73) we can lower bound

$$\mathcal{E}_{P^2}(f, f) = \sum_{i=1}^4 \nu(G_i) \mathcal{E}_{\nu_{G_i}, (P^2)_{|G_i}}(f_{|G_i}, f_{|G_i}).$$

Now observe that for each $i$, since $\langle f, \mathbb{1}_{G_i} \rangle = \langle f_{|G_i}, \mathbb{1}_{G_i} \rangle = 0$ if $f \in W$, we can lower bound

$$\mathcal{E}_{\nu_{G_i}, (P^2)_{|G_i}}(f_{|G_i}, f_{|G_i}) \geq \lambda_i \|f_{|G_i}\|_{\ell^2(\nu_{G_i})}^2.$$

Therefore

$$\mathcal{E}_P(f, f) \geq \sum_{i=1}^4 \lambda_i \nu(G_i) \|f_{|G_i}\|_{\ell^2(\nu_{G_i})}^2$$
$$\geq \lambda \|f\|_{\ell^2(\nu)}^2$$

which proves the claim. $\qquad\square$

*Remark* 3. We note that Theorem 4 is quite general and applies to arbitrary decompositions of the state space. However, our framework is particularly effective in scenarios where there is a significant gap between the global mixing time of the Markov chain and the local mixing time within each "room." In favorable cases—such as when each room is an expander graph—this difference can be substantial. In contrast, if each room is a 2D grid with $n^2$ states and the policy corresponds to a random walk, the local mixing time scales as $\mathcal{O}(n^2)$, while the global mixing time scales as $\mathcal{O}(n^2 \log n)$. This setup closely resembles the so-called "$n$-dog" graph studied in Example 3.3.5 of (54), where two $n \times n$ grids are connected at a single corner. In this case, the difference between local and global mixing is relatively mild. Nonetheless, in our numerical experiments, which include scenarios resembling this more challenging setting. we already observe significant gains from shifting the successor measure.

# H Further Numerical Experiments

To complement the theoretical insights and main experimental findings, we provide additional numerical results that investigate the behavior of shifted successor measures across a wider range of settings. These experiments aim to probe the robustness and generality of our approach in different environments, under different data collection policies, and with both model-based and model-free estimators. All experiments were run on a single CPU and are reproducible within a day. As mentioned in the main text, all code is available at `https://github.com/stestoKTH/shift-SM`.

## H.1 The 4-room environment

We now revisit the 4-room environment theoretically analyzed in Theorem 4, where the state space is partitioned into four well-connected regions (rooms) linked by narrow passageways. As discussed in the main text, this structure induces metastable behavior: the chain mixes rapidly within each room, while transitions between rooms are relatively infrequent. In this section, we additionally make the Markov chain aperiodic by allowing the agent to remain in its current state with probability $0.1$.

Figure 6 illustrates several empirical findings. On the left, we show the 15x15 discretization of the 4-room domain that we use in this section. In the center panel, we plot the singular values of the shifted successor measures $M_{\pi,k}$ for various values of the shift $k$. As theoretically predicted, increasing $k$ leads to a sharper spectral decay, indicating stronger low-rank structure. Notably, for higher shifts - when all states within a room are reachable - the effective rank is close to 4, matching the number of rooms.

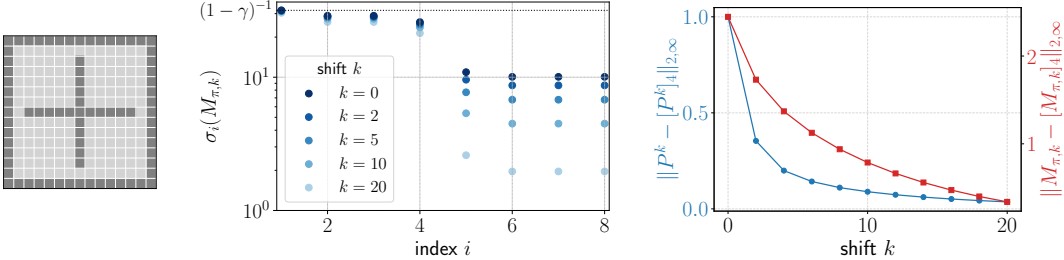

Figure 6: Left: 4-room environment with a 15x15 discrete space; Center: singular values of shifted successor measures $M_{\pi,k}$ for various shifts $k$ (uniform policy $\pi$, discount factor $\gamma = 0.97$); Right: entrywise norm differences between $P^k$ and its rank-4 approximation (blue circles), and between $M_{\pi,k}$ and its rank-4 approximation (red squares). As in Figure 2, we use the standard $\|\cdot\|_{2,\infty}$ norm, which coincides with the norm in Section 3.2 up to a $\sqrt{n}$ multiplicative factor under the uniform measure $\nu$.

On the right, we plot two metrics as a function of $k$: the entry-wise norms $\|P^k - [P^k]_4\|_{2,\infty}$ and $\|M_{\pi,k} - [M_{\pi,k}]_4\|_{2,\infty}$. Both metrics decay rapidly with $k$, consistent with the bounds in Theorem 4. The behavior confirms that moderate values of $k$ (e.g., $k = 4 - 10$) are sufficient to approximate $P^k$ with a rank$-4$ matrix. While such a representation may suffice for navigating between rooms, accurately reaching specific target states within a room may require a higher-rank approximation. Nevertheless, shifting the successor measure consistently improves the learnability of low-rank representations.

These results validate our theoretical predictions in a structured setting and demonstrate how temporal shifting can uncover the environment's block structure. We next turn to more complex and less regular domains.

## H.2 Additional Navigation Tasks

We now extend the results from Section 6 to additional environments of increasing complexity. Specifically, we evaluate the impact of shifting and low-rank approximation of successor measures in two additional mazes: the U-maze and the Large-maze. All three mazes are discretized versions of the Maze2D environments from (23).

Here we repeat the setup from Section 6 and provide additional details. Unless stated otherwise (as in Section H.3), all data is collected using a uniformly random policy. This simplifies the estimation process: under a uniform data distribution, the invariant measure $\nu$ is uniform, and thus the measure-dependent norms introduced in Section 3.2 reduce to their standard variants. In particular, the $\|\cdot\|_{2,\infty}$ norm and the singular value decomposition (SVD) used for low-rank approximation become standard.

Once the successor measures $M_{\pi,k}$ are estimated, we evaluate policies that act greedily with respect to them. More specifically, given a current state $s$ and a goal $g$, the policy selects actions according to: $\mathrm{argmax}_{a\in\mathcal{A}}\sum_{b\in\mathcal{A}}M_{\pi,k}(s,a,g,b)$, as described in Section 6. In the low-rank setting, the same greedy procedure is applied to the rank$-r$ approximation $[M_{\pi,k}]_r$. To quantify goal-reaching performance and evaluate the obtained policy, we report two metrics: accuracy, the probability of reaching the exact goal (from a random initial state), and relaxed accuracy, the probability of reaching any state within two steps of the goal.

Figures 7 and 8 mirror the structure of Figure 4 in the main paper. In each case, we compare unshifted and shifted successor measures across several metrics: spectrum decay (panel b), goal-reaching accuracy using ground-truth successor measures (panels c–d), performance of TD-learned measures (panels e–f), and sample efficiency as a function of dataset size (panels g–h).

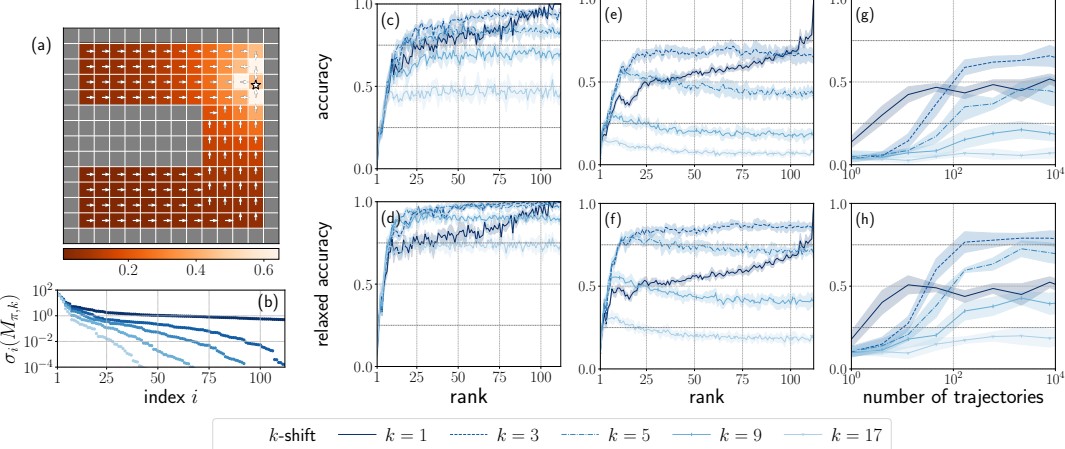

Figure 7: Successor measure analysis in the U-maze environment with $\gamma = 0.98$ and uniformly random policy $\pi$. TD estimates use 10k trajectories of length $H = 100$; rank is fixed to 40 in (g–h). Results are averaged over 5 seeds and 100 random goals and initial positions.

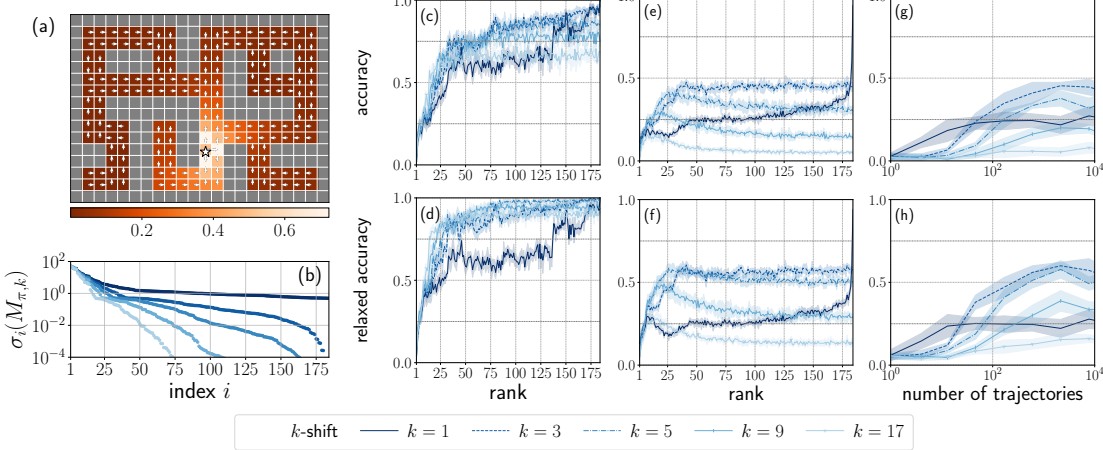

Figure 8: Same setup as in Figure 7, now for the Large-maze environment. Rank is fixed to 60 in (g–h). All results are averaged over 5 seeds and 100 random goals and initial positions.

In both environments, we observe a consistent pattern: shifting enhances spectral decay (Figure 7b and 8b), making the structure more amenable to low-rank approximation. When true successor

measures are available (panels c–d), moderate shift values yield better planning performance at low ranks, consistent with our observations in the Medium-maze environment. However, beyond a certain point, excessive shifting discards too much information, leading to worse performance. This effect is more pronounced when successor measures are learned (panels e–f), likely due to compounding estimation error over long horizons.

Finally, we evaluate how accuracy varies with the number of trajectories (panels g–h). As in the main experiments, moderate shifts ($k = 3$ or $k = 5$) often strike the best balance between representational power and sample efficiency. The trade-off seen in Figure 4 g–h, where small shifts underexploit structure and large shifts overburden estimation, persists across these environments.

Overall, these experiments reinforce our findings from Section 6 and demonstrate the robustness of temporal shifting across domains. Even in larger and more complex mazes, appropriately calibrated shifting enables more compact representations, improves planning accuracy, and enhances sample efficiency.

### H.3   Non-uniform Data Collecting Policy

In contrast to the previous experiments that used a uniformly random data-collection policy, we now evaluate a mixed policy composed of 80% uniformly random actions and 20% averaged goal-conditioned behavior. Specifically, the latter operates by sampling a goal uniformly at random and following the optimal policy to reach it, repeating this process for all goals (corresponding to $\pi_\mathcal{D}(a|s) = \int_\mathcal{S} \pi_g(a|s) d\rho_\mathcal{D}(g)$ from Section 6 with uniform $\rho_\mathcal{D}$). As shown in the leftmost panel of Figure 9, this results in a non-uniform invariant measure $\nu$, with states near the geometric center of the maze being visited more frequently.

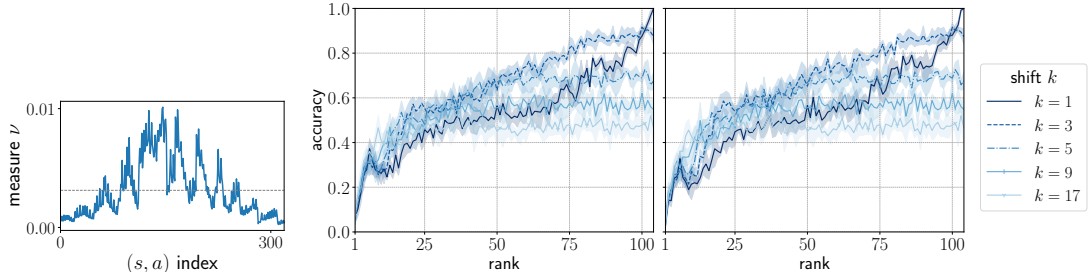

Figure 9: Left: invariant measure $\nu$ with respect to $M_\pi$, dashed line represents uniform distribution. Center/right: accuracy vs. rank for standard SVD and $\nu$-SVD, same setting as in Figure 4c), with only the data-collection policy modified.

To account for this skewed distribution, we use the $\nu$-weighted SVD (as described in Section 3.2) when computing low-rank approximations of $M_{\pi,k}$. Figure 10 shows that the reconstructions obtained with $\nu$-SVD differ significantly from those of the standard SVD, especially at low ranks. However, despite this discrepancy, goal-reaching performance remains nearly unchanged, as seen in the center and right panels of Figure 9.

All experiments were performed in the Medium-maze using the same setting as in Section 6. Interestingly, the results suggest that the uniformly random policy actually yields slightly better performance at low ranks (compare with Figure 4c), suggesting that more uniform exploration may facilitate learning better goal-reaching representations.

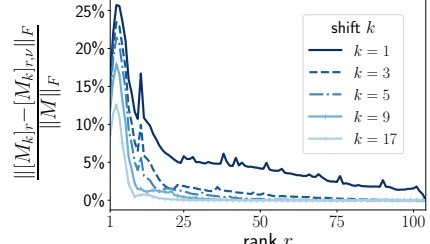

Figure 10: Relative Frobenius difference between rank$-r$ approximations of $M_{\pi,k}$ using standard SVD vs. $\nu$-SVD.

### H.4   Model-Based Estimation of Shifted Successor Measures

We now compare temporal-difference (TD) learning with a simple model-based (MB) approach for estimating shifted successor measures. In the model-based case, we first estimate the transition matrix

$P_\pi$ from data, and then compute the shifted successor measure $M_{\pi,k} = \sum_{t=0}^{\infty} \gamma^t P_\pi^{t+k}$ using a truncated power series expansion. Figure 11 (left) reproduces the TD-based results from Figure 4 (g) in the Medium-maze, while Figure 11 (right) shows the corresponding performance of the model-based estimator.

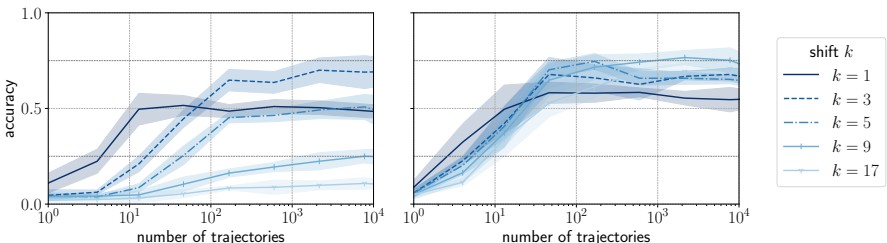

Figure 11: Goal-reaching accuracy in the Medium-maze using TD (left) and model-based (right) estimation. In both cases, we use trajectories of length 100, collected with a uniformly random policy, $\gamma = 0.95$ and fixed rank $r = 40$.

We observe that the model-based approach maintains higher goal-reaching accuracy even for larger shift values $k$. This is expected: unlike TD, which relies on sparse, temporally aligned supervision (i.e., observing specific $(s_t, a_t, s_{t+k+1})$ transitions), the model-based method can leverage all available transitions to estimate $P_\pi$, making it less sensitive to the horizon length. In particular, long-range transitions needed for higher shifts are harder to estimate via TD when data is limited, whereas they are implicitly captured in $P_\pi$ and recovered through matrix powers in the MB estimator.

While model-based estimation proves more robust in this tabular setting, it does not easily scale to environments with large or continuous state spaces. Storing and computing with full transition matrices becomes infeasible, making function approximation challenging. In such cases, TD learning might be more practical and scalable despite its limitations with longer shifts.

## H.5 Extension to the Non-Tabular Setting

A natural question is whether the benefits of shifted successor measures observed in discrete maze environments carry over to more complex settings with stochastic dynamics and continuous state-action spaces. We believe that learning shifted successor measures may yield similar benefits in such environments - particularly in cases where learning the standard, non-shifted successor measure proves challenging.

While we do not provide formal guarantees under function approximation, we believe similar effects are likely to emerge in practice. This intuition aligns with prior work on hierarchical reinforcement learning (ex. (50; 51)), where high-level policies capture the coarse structure of the task and steer the agent toward the vicinity of its goal. It would be interesting to explore whether shifted successor measures could similarly encode such high-level behaviors.

One particularly promising direction is contrastive learning. For example, (21) samples positive examples from a geometrically distributed time offset governed by the discount factor $\gamma$. To incorporate a shift $k$, one could instead sample the offset from $\mathrm{Geom}(1 - \gamma) + k$, effectively biasing learning toward more temporally distant predictions.

By contrast, extending these ideas to Forward-Backward (FB) algorithm of (63) is less straightforward. A key strength of FB is its ability to learn from one-step transitions $(s_t, a_t, s_{t+1})$ independently of the data collection policy. How to integrate a meaningful notion of temporal shift into such a framework remains an open and intriguing challenge.

We see these directions as promising opportunities to extend the benefits of temporal shifting beyond tabular settings, and hope that the theoretical insights in this work will help guide future progress in more realistic domains.

