# OpenReview forum: "Shift Before You Learn: Enabling Low-Rank Representations in Reinforcement Learning"
_NeurIPS.cc/2025/Conference — NeurIPS 2025 spotlight_

### Official Review · Reviewer_6ieF · 2025-07-01

**Clarity:** 3
**Significance:** 2
**Originality:** 3
**Rating:** 4
**Confidence:** 3

**Summary:**

Successor representation (SR) is a widely used approach for reward-free representation learning in reinforcement learning. This paper focuses on computing low-rank approximations of SR. However, if the underlying MDP is not low-rank, its successor representation is unlikely to be low-rank either, which introduces approximation error. To address this, the authors propose a new variant called the $k$-shifted SR, that captures the dynamics of policy $\pi$ starting from time step  $k$.

**Questions:**

- If the MDP we aim to represent is not inherently low-rank, it’s unclear why we would prefer a low-rank representation in the first place. Why not simply use the standard full-rank successor representation (SR)? The justification for opting for a low-rank SR remains unclear. Is the goal to reduce memory usage, computational cost, or sample complexity? Or does it empirically lead to faster learning? These motivations need to be more explicitly articulated.
- The paper shows results on the regret when estimating the shifted representation $M_{\pi,k}$. However, it is unclear how this quantity is related to the regret of a policy that is computed via the shifted representation. How is $M_{\pi,k}$ related to $M_{\pi}$? One could accurately estimate  $M_{\pi,k}$ but still get a bad policy. This could happen, for example, if the policy improvement method (e.g., GPI or policy iteration) amplifies the approximation error in an undesirable way. In my view, unless it is demonstrated that using  $M_{\pi}$​ for policy computation leads to low regret in control, the importance of estimating it accurately remains unclear. This is likely my primary concern regarding the contribution. If this issue were resolved, the significance of the work would become much more apparent.

**Ethical Concerns:**

["NO or VERY MINOR ethics concerns only"]

**Final Justification:**

Considering the initial criticism and the authors' responses to it, I have decided to raise the final grade and consider this paper to be suitable for acceptance.

**Limitations:**

Some of the weakness listed below needs further comments on the text to address this potential limitations.

**Paper Formatting Concerns:**

No concerns.

**Quality:**

2

**Strengths And Weaknesses:**

Strengths:
- The paper is overall well written and easy to follow.
- The proposed shifted SR is a natural and clever response to the limitations of low-rank SR in general MDPs.

Weaknesses:
- The relevance of the proposed approach is unclear. For instance, it remains uncertain whether obtaining the shifted-SR representation is actually useful for deriving good policies. The authors provide no theoretical guarantees about the quality of policies obtained after estimating the shifted-SR. Consequently, the practical utility of these representations is unclear.
Given this lack of clarity, the paper essentially proposes the problem of estimating shifted-SRs and a method to do so, but fails to establish why this is meaningful in the first place.

---

> ### Author Rebuttal · Authors · 2025-07-31
>
> Thank you for reviewing our paper. Before answering your questions, we would like to briefly highlight our contributions.
>
> Our main contribution is to identify conditions under which low-rank structure naturally emerges in zero-shot reinforcement learning. Unlike prior work that assumes universal low-dimensional representations that generalize across tasks, e.g. Forward-Backward model (Touati and Ollivier, 2021; Touati et al., 2023) or goal-conditioned RL frameworks (Andrychowicz et al., 2017; Eysenbach et al., 2022b), we provide a theoretical basis for when and why such structure arises.
>
> Specifically, we show that learning successor measures directly is statistically challenging, and demonstrate that a simple temporal shift of successor measures naturally induces low-dimensional representations.
>
> Furthermore, we present preliminary numerical results in section 6 and additional experiments in Appendix H that illustrate the benefits of learning shifted successor measures. We believe that our work lays the foundation for practical algorithms for policy learning via shifted successor measures.
>
>
> **Motivation for using low-rank successor representations.**
>
> Thanks for this question: the objective of our paper is really to answer it! Without any structure, learning cannot be statistically efficient. That is why (and we explain this in the intro) many successful existing approaches in RL assume the existence of a low-rank structure, usually by assuming low-dimensional representation vectors $\phi(s,a)$ and $\psi(g)$. In this paper, we explain how such a structure appears naturally and universally (section 5), and we exemplify how the statistical efficiency of RL estimation procedure is improved by leveraging this structure (section 4). Note however that our results in Theorem 1 are always valid (not only for low-rank) and yet show the impact of having low-rank structures.
>
> **Connection between shifted-SR and optimal policy.**
>
> We thank the reviewer for raising this important question. There are three crucial points in addressing this issue:
>
> (a) **On the infeasibility of estimating $M_\pi$.** If the reward function is known, the matrix product $M_{\pi} R$ directly gives the Q-function. Therefore, estimating $M_{\pi}$ allows policy evaluation for all rewards, which is of prime interest for transfer learning, multi-goal RL, etc. However, as discussed earlier, we argue that accurately estimating $M_{\pi}$ is generally infeasible due to the absence of a low-rank structure, and propose shifted successor measures as a way to circumvent this issue. For a moderate shift, $M_{\pi,k}$ remains reasonably close to $M_{\pi}$, while being more easily recoverable using low-rank approximation.
>
> (b) **Shifted-SR recover (nearly)-optimal policy in Goal Conditioned RL.** In sparse reward settings, the most challenging in practice, policy updates based on shifted objectives yield policies that coincide with the optimal policies except when very close to the goal. Intuitively, for a fixed shift parameter $k$, if the goal is more than $k$ steps away from the current state, the Q-values computed using $M_{\pi,k}R$ are identical to those computed from $M_{\pi}R$. This ensures that the resulting policy is optimal until the agent enters the $k$-neighborhood of the goal, where exact optimality cannot be guaranteed. Nonetheless, as shown empirically in section 6 (and Appendix H), shifted-SR still recovers nearly optimal policies in practice.
>
> (c) **On the importance of entrywise guarantees for good policy recovery.** Our work focuses on policy evaluation, and in this context, the entrywise guarantees we provide i.e. bounds $\Vert M - \widehat{M}\Vert_{2\to\infty}$ are crucial. Such guarantees imply that greedy policy $\pi_{\widehat{M}}$ derived from $\widehat{M}$ will perform well. This contrasts with weaker guarantees in global norms such as the Frobenius norm, which do not provide the same level of assurance for individual states. Thus, the strong entrywise bounds we establish are central to ensuring that accurate estimation of $M$ leads to high-quality policies.

---

> ### Author Response · Authors · 2025-08-06
>
> Dear reviewer,
>
> We hope our rebuttal has addressed your questions. We would greatly appreciate it if you could let us know whether there are any points you would like to discuss further or would like us to clarify. Thank you very much for your time and consideration.
>
> Best regards,
>
> The authors.

---

> > ### Comment · Reviewer_6ieF · 2025-08-06
> >
> > I would like to thank the authors for your informative answers, and thank you also to the area chair for sharing his thoughts on the matter. To be honest, one of my main concerns regarding the contribution of this article has been well addressed by response c), and supported by the area chair. I think this convinces me to raise my rating slightly. However, I still think that making this connection more explicit in the paper, or presenting some results that explicitly point to the quality of the policy obtained with $M_{\pi,k}$ would be of great value.

---

### Official Review · Reviewer_tYCN · 2025-07-02

**Clarity:** 3
**Significance:** 2
**Originality:** 3
**Rating:** 5
**Confidence:** 3

**Summary:**

The authors present a theoretical work focused on challenges associated with building successor measures in goal-conditioned RL. In particular, the authors analyze a common assumption that the successor measure can be effectively modeled via low-rank representation and find that, whereas vanilla approaches are not guaranteed to admit a low-rank structure, such low-rank structure naturally emerges in shifted successor measures (where "shift" refers to skipping the first k time-steps of the trajectory before building the successor measure). Furthermore, the authors provide sample complexity bounds and errors for learning the shifted successor measure, which are shown to be related to the spectral recoverability of the successor matrix. Finally, the authors present a small experimental section consisting of a discrete, goal-conditioned maze. These experiments show that, in line with the provided theory, low-rank approximation works better with shifted successor measures.

**Questions:**

1.How restrictive are the assumptions made in corollary 1? In what types of setups do these hold?
2.How do these theoretical considerations apply to continuous state/action environments?
3.What is the significance of relaxed accuracy as compared to accuracy? Why do the policies with higher shifts perform relatively better in terms of relaxed accuracy than in terms of accuracy?
4.Could the authors expand the intuition on why shifting would lead to a low-rank structure? Does this observation hold in more stochastic problems?

**Ethical Concerns:**

["NO or VERY MINOR ethics concerns only"]

**Limitations:**

Authors could slightly improve the following aspects of their work:
1.The limitations of the theoretical results - adding a paragraph summarizing the impact of the theoretical result on the practical workflows of reward-free RL, as well as the limitations associated with the assumptions
2.The limitations of the experimental section - adding a paragraph discussing which assumptions and other elements of the theoretical analysis correspond to the experimental setup

**Quality:**

3

**Strengths And Weaknesses:**

Strengths:
1.Despite being quite mathematically dense, the paper is clearly written (except for the experimental section).
2.The manuscript provides an analysis of a common assumption used in reward-free RL with results that can inform future algorithm design.
3.To the best of my knowledge, the strategy for obtaining the theoretical results is novel in the context of RL (although I did not check the correctness of the proofs).

Weaknesses:
1.The experimental section is relatively small and dense. It would be helpful to describe the experimental settings and the results in more detail. For example, a more direct discussion of the learning algorithm (e.g. hyperparameters), implications of the experiments with respect to the presented theory, and limitations of the experimental setup would be particularly helpful for the reader. Having said this, I understand that the theoretical analysis is the central contribution of this paper.

---

> ### Author Rebuttal · Authors · 2025-07-31
>
> Thank you for reviewing our paper. Before answering your questions, we would like to briefly highlight our contributions.
>
> Our main contribution is to identify conditions under which low-rank structure naturally emerges in zero-shot reinforcement learning. Unlike prior work that assumes universal low-dimensional representations that generalize across tasks, e.g. Forward-Backward model (Touati and Ollivier, 2021; Touati et al., 2023) or goal-conditioned RL frameworks (Andrychowicz et al., 2017; Eysenbach et al., 2022b), we provide a theoretical basis for when and why such structure arises.
>
> Specifically, we show that learning successor measures directly is statistically challenging, and demonstrate that a simple temporal shift of successor measures naturally induces low-dimensional representations.
>
> Furthermore, we present preliminary numerical results in section 6 and additional experiments in Appendix H that illustrate the benefits of learning shifted successor measures. We believe that our work lays the foundation for practical algorithms for policy learning via shifted successor measures.
>
> **On the scope and complexity of experimental validation.**
>
> As noted in the main text, Appendix H provides additional results on goal-conditioned tasks, which are discretized versions of standard (continuous) benchmarks for navigation tasks. There, we also discussed the impact of different data-collection policies and learning algorithms. Moreover, Appendix H.5 discusses extensions to non-tabular environments.
>
> We agree that evaluating on high-dimensional environments is a valuable future direction. However, the goal of our experiments is to validate the theoretical insights under controlled conditions, where factors can be isolated and analyzed rigorously. We believe the current experiments, together with our theoretical contributions, make the paper comprehensive and self-contained.
>
> **Restrictiveness of assumptions in Corollary 1.**
>
> The validity of the assumptions depend on the intrinsic dynamics but also on the choice of the policy. Overall, Corollary 1 requires the existence of an underlying structure (e.g. to have $\sigma_1, \ldots \sigma_r$ of order $1$, and $\sigma_{r+1}$ small), a sufficient shift (to have $\xi(M_{\pi,k})$ bounded), and a reasonably exploring policy (to have an underlying measure essentially invariant and uniform). We discuss these more in detail in Appendix C.1. The kind of setup where these apply well is for instance maze environments, like the four-room environment we give as an example in Theorem 4.
>
> **Extending the theory to continuous environments.**
>
> The notion of successor measure naturally extends to continuous state/action spaces using more general linear operators and measures instead of matrices (as is done in the forward-backward papers). We could extend Theorem 1 to this setting by discretization and leveraging smoothness properties of the dynamics (thereby requiring another assumption), in the spirit of the papers [1,2,3]. We restrict to the discrete setting first for simplicity, secondly because continuous settings are to be discretized anyway.
>
> [1] Devavrat Shah, Dogyoon Song, Zhi Xu, and Yuzhe Yang. Sample efficient reinforcement learning via low-rank matrix estimation. NeurIPS 2020.
>
> [2] Stefan Stojanovic, Yassir Jedra, and Alexandre Proutiere. Model-free low-rank reinforcement learning via leveraged entry-wise matrix estimation. NeurIPS 2024.
>
> [3] Jiaming Xu. Rates of convergence of spectral methods for graphon estimation, ICML 2018
>
> **On the significance of relaxed accuracy.**
>
> In many scenarios, reaching the exact goal is unnecessary - being close to the goal is sufficient. Relaxed accuracy captures this notion. As shown in Figure 4 in section 6, shifted successor measures not only improve exact goal-reaching with low-rank approximation (c), but also perform better at reaching the neighborhood of the goal (d). This demonstrates the robustness of shifted successor measures - even when they do not yield the optimal policy, they still outperform the non-shifted baseline in getting close to the goal.
>
> Policies with higher shifts perform relatively better because they are better aligned with relaxed objectives: if we consider a $k$-shift and thus neglect to collect the reward in the first $k$ steps, this matters at most within a radius $k$ around the reward, where a sub-optimal trajectory may not be penalized. By looking at relaxed accuracy, we do not account for this sub-optimality.
>
> **Intuition behind why shifting induces low-rank structure.**
>
> Shifting corresponds to multiplying by $P_{\pi}^k$, hence the shifted SM is the discounted sum of matrices $P_{\pi}^{k+l}$ for $l\ge 0$ which are in general more approximately low rank for large $k$: on the one hand, $I$ or $P_{\pi}$ present in the non-shifted SM have no reason to be low-rank, while $P_{\pi}^{k}$ converges to a rank $1$ matrix as $k \rightarrow \infty$ in the ergodic case (one eigenvalue is equal to $1$ and all others are strictly below $1$ in absolute value). The shift is a way to interpolate between these two extreme cases. This is the mixing intuition we make more rigorous in section 5.
>
> Furthermore, this intuition works even better in more stochastic problems, since stochastic dynamics are more likely to be ergodic. Ergodicity may not be required however, provided we restrict to an appropriate sub-dynamics: see our discussion in Appendix C.1.
>
> **On the limitations of theoretical results.**
>
> Thanks for this remark. Regarding limitations, we discuss briefly the different terms of our upper bound after Theorem 1, while a more complete discussion is also given in Appendix C. We will highlight that more and summarize in the camera-ready version of the paper.
>
> **On the limitations of experimental results.**
>
> We thank the reviewer for this comment and refer them to Appendix H for a detailed description of the experimental setup. In the camera-ready version, we will add a paragraph in section 6 explicitly linking the experimental assumptions and design to the theoretical analysis.

---

> ### Author Response · Authors · 2025-08-06
>
> Dear reviewer,
>
> We hope our rebuttal has addressed your questions. We would greatly appreciate it if you could let us know whether there are any points you would like to discuss further or would like us to clarify. Thank you very much for your time and consideration.
>
> Best regards,
>
> The authors.

---

### Official Review · Reviewer_xQzx · 2025-07-02

**Clarity:** 3
**Significance:** 3
**Originality:** 3
**Rating:** 4
**Confidence:** 2

**Summary:**

This paper proposes ​shift successor measures​ to better separate two types of transitions: those involving short-range dynamics (which remain inherently high-rank) from those governed by long-range interactions. The experiments demonstrate that shifted successor measures admit more accurate low-rank SVD approximations than non-shifted counterparts, and that enhanced representation yields consistent performance gains in goal-conditioned reinforcement learning tasks.

**Questions:**

1. What's the computational efficiency of deriving the shifted successor measure estimator?

**Ethical Concerns:**

["NO or VERY MINOR ethics concerns only"]

**Final Justification:**

As indicated in my prior comments, while my expertise is less aligned with this particular paper's specific scope, the contributions described lead me to uphold my original decision.

**Limitations:**

Yes, the author presents the discussion on limitations in the appendix.

**Paper Formatting Concerns:**

There are no major formatting issues in the paper.

**Quality:**

3

**Strengths And Weaknesses:**

**Pros:**

While I'm not familiar in low-rank RL theory, the paper demonstrate its contributions to this research area. It clearly introduces the problem setting in the introduction, followed by explanations of the methodology grounded in strong intuition that build toward a rigorous theoretical analysis with detailed proofs. Critically, the empirical validation demonstrates that shifted successor measures achieve notably better low-rank approximations than standard counterparts, confirming the method's effectiveness.

**Cons:**
1. Experiments are confined to a ​single discrete environment​. Broader validation on a continuous environment would be better to support claims of general applicability.

2. Adding comparisons against baselines would better demonstrate the advantages of the proposed method.

---

> ### Author Rebuttal · Authors · 2025-07-31
>
> Thank you for reviewing our paper. Before answering your questions, we would like to briefly highlight our contributions.
>
> Our main contribution is to identify conditions under which low-rank structure naturally emerges in zero-shot reinforcement learning. Unlike prior work that assumes universal low-dimensional representations that generalize across tasks, e.g. Forward-Backward model (Touati and Ollivier, 2021; Touati et al., 2023) or goal-conditioned RL frameworks (Andrychowicz et al., 2017; Eysenbach et al., 2022b), we provide a theoretical basis for when and why such structure arises.
>
> Specifically, we show that learning successor measures directly is statistically challenging, and demonstrate that a simple temporal shift of successor measures naturally induces low-dimensional representations.
>
> Furthermore, we present preliminary numerical results in section 6 and additional experiments in Appendix H that illustrate the benefits of learning shifted successor measures. We believe that our work lays the foundation for practical algorithms for policy learning via shifted successor measures.
>
> **On the scope and complexity of experimental validation.**
>
> As noted in the main text, Appendix H provides additional results on goal-conditioned tasks, which are discretized versions of standard (continuous) benchmarks for navigation tasks. There, we also discussed the impact of different data-collection policies and learning algorithms. Moreover, Appendix H.5 discusses extensions to non-tabular environments.
>
> We agree that evaluating on high-dimensional environments is a valuable future direction. However, the goal of our experiments is to validate the theoretical insights under controlled conditions, where factors can be isolated and analyzed rigorously. We believe the current experiments, together with our theoretical contributions, make the paper comprehensive and self-contained.
>
>
> **Comparison with the baselines.**
>
> This paper introduces a novel insight not previously reported - assuming low-dimensional successor representations apriori is unnecessary, as low-dimensional structure naturally emerges when shifting successor measures. We believe this theoretical contribution is significant on its own and can lead to further advances.
>
> Our algorithm is not directly comparable to standard baselines because it relies on SVD-based estimators, which are tailored to the tabular setting and do not trivially extend to high-dimensional environments. As discussed in Appendix H.5, extending the approach is possible but would likely come at the cost of the theoretical guarantees established in this work.
>
> Finally, our experiments are designed to illustrate and validate the theoretical results in a controlled setting rather than to benchmark against existing methods, ensuring a clear link between theory and empirical results.
>
> **On the computational efficiency of shifted estimator.**
>
> In the tabular setting, the computational complexity of our algorithm is dominated by the singular value decomposition step. For more general settings discussed in Appendix H.5, the efficiency depends on the underlying algorithm for learning the successor measure. In some cases, such as when using contrastive learning approaches, incorporating the shift introduce minimal overhead and does not increase computational complexity.

---

> > ### Comment · Reviewer_xQzx · 2025-08-05
> >
> > I thank the author’s response, I’ll maintain the current score.

---

### Official Review · Reviewer_FgQx · 2025-07-08

**Clarity:** 3
**Significance:** 3
**Originality:** 4
**Rating:** 4
**Confidence:** 3

**Summary:**

The paper studies the low-rank representations in RL through a shifted successor measure, supported by strong theoretical contributions. It claims that the standard successor measure is not inherently low-rank, and a shifted successor measure, which captures dynamics subsequent to omitting earlier transitions, can demonstrate the low-rank property. This paper provides a novel perspective on the analysis of long-term dynamics. However, the proposed method may lose practicality since the assumptions in Section 5 are somewhat strong, which may fail in practice (e.g., in complex or dynamic environments).

**Questions:**

- How to choose k in practical implementation?
- The proposed method may lose practicality since the assumptions in Section 5 are somewhat strong, which may fail in practice (e.g., in complex or dynamic environments). In addition, the experiments validate the results in a small, discrete envs. Demonstrating the method's effectiveness on a more challenging, high-dimensional benchmark would be better to show its practical values in real-world tasks.

**Ethical Concerns:**

["NO or VERY MINOR ethics concerns only"]

**Final Justification:**

Thanks for the responses of the authors, as they addressed many of my concerns.

**Limitations:**

Yes

**Quality:**

3

**Strengths And Weaknesses:**

# Strengths:
- Novel Perspective: The authors redefine the low-rank assumptions in RL by introducing the shifted successor measure, providing a novel perspective on the analysis of long-term dynamics.
- Theoretical Proof: The authors provide rigorous proofs with spectral recoverability and Type II Poincaré inequalities to enhance the understanding and effectiveness of low-rank approximations in RL.

# Limitations:
- Selection of the Shift Parameter: The theoretical results explain why a  shifted successor helps, but without a guidance to choose k, which is critical in practical implementation.
- Strong Assumptions: the proposed method may lose practicality since the assumptions in Section 5 are somewhat strong, which may fail in practice (e.g., in complex or dynamic environments). In addition, the experiments validate the results in a small, discrete envs. Demonstrating the method's effectiveness on a more challenging, high-dimensional benchmark would be better to show its practical values in real-world tasks.

---

> ### Author Rebuttal · Authors · 2025-07-31
>
> Thank you for reviewing our paper. Before answering your questions, we would like to briefly highlight our contributions.
>
> Our main contribution is to identify conditions under which low-rank structure naturally emerges in zero-shot reinforcement learning. Unlike prior work that assumes universal low-dimensional representations that generalize
>     across tasks, e.g. Forward-Backward model (Touati and Ollivier, 2021; Touati et al., 2023) or goal-conditioned RL frameworks (Andrychowicz et al., 2017; Eysenbach et al., 2022b), we provide a theoretical basis for when and why such structure arises.
>
> Specifically, we show that learning successor measures directly is statistically challenging, and demonstrate that a simple temporal shift of successor measures naturally induces low-dimensional representations.
>
> Furthermore, we present preliminary numerical results in section 6 and additional experiments in Appendix H that illustrate the benefits of learning shifted successor measures. We believe that our work lays the foundation for practical algorithms for policy learning via shifted successor measures.
>
>
> **Choosing the shift parameter $k$ in practice.**
>
> As discussed in section 5, the optimal shift value $k$ can be derived in certain settings. For general environments with complex dynamics, $k$ must be treated as a tunable hyperparameter. Its value is related to other hyperparameters, such as the discount factor. This is consistent with prior work - for instance, HIQL (Park et al., 2024) treats the number of subgoal steps $k$ as a hyperparameter, tuning it per environment. We argue that selecting shift parameter $k$ in our setting is no more difficult than setting the subgoal step parameter in HIQL.
>
> **Practicality of the assumptions in section 5.**
>
> As far as we are aware, we are first to analyze what happens when shifting successor measures, and we believe this idea is novel and interesting for further practical analysis. Our main result of section 4 makes no assumption and applies to arbitrarily complex, discrete dynamics. It could be extended to continuous settings, with the right regularity assumptions, as we answered Reviewer tYCN. The ideas of section 5 are also universal: mixing phenomena exist beyond discrete Markov chains, which is mainly a way to avoid more tedious formalism in continuous setting. Furthermore, these ideas could apply even to Markov processes which do not mix in the classical sense. We discuss some extensions in Appendix C.1. Thus we believe our analysis holds in good generality and provides insight that might be interesting for their own sake. What we do in section 5.2 (decomposable Markov chains) is to just provide an example where the analysis can be directly applied. There, we indeed make some assumptions.
>
> **On the scope and complexity of experimental validation.**
>
> As noted in the main text, Appendix H provides additional results on goal-conditioned tasks, which are discretized versions of standard (continuous) benchmarks for navigation tasks. There, we also discussed the impact of different data-collection policies and learning algorithms. Moreover, Appendix H.5 discusses extensions to non-tabular environments.
>
> We agree that evaluating on high-dimensional environments is a valuable future direction. However, the goal of our experiments is to validate the theoretical insights under controlled conditions, where factors can be isolated and analyzed rigorously. We believe the current experiments, together with our theoretical contributions, make the paper comprehensive and self-contained.

---

### Note · Authors · 2025-08-12

We would like to thank the reviewers and the Area Chair for their valuable feedback, and insightful questions, that, we hope, helped us to clarify our contributions. We are grateful to Reviewer 6ieF for raising the paper rating. Once again, thank you or your time and consideration.

Best regards,

The authors

---

### Decision · Program_Chairs · 2025-09-17

**Decision:**

Accept (spotlight)

**Comment:**

A lot of RL problems become easier if we can estimate the successor measure for a given environment and behavior policy: a matrix whose (s, a, s’, a’) element is the expected discounted frequency of visits to (s’, a’) if we start at (s, a). Many approaches to estimating this matrix implicitly assume that it is low rank. This paper points out (1) that it requires restrictive assumptions to ensure low rank, (2) that a related matrix, the shifted successor measure, can be low rank under different (possibly weaker) assumptions, and (3) that the shifted successor measure can be almost equally useful for many RL problems. The k-shifted successor measure has elements equal to the expected discounted frequency of visits to (s’, a’) if we start at (s, a) *and ignore the first k time steps*.

The bulk of the paper provides definitions and analysis to prove low rank and reconstruction guarantees: when is the shifted successor measure matrix approximately low rank, and when does this lead to accurate recovery from few samples. Some of the quantities that the bounds depend on:
* the ratio between the visitation distribution of the behavior policy and the weights used to do the SVD
* the distance from uniformity of this visitation distribution (this seems somewhat restrictive, and isn’t discussed much in the main paper)
* the existence of an appropriate rank r: the r-th singular value should be large enough and there should be enough of a gap to the (r+1)-st singular value
* the incoherence (at rank r) of the successor measure matrix — if it is too coherent, we have to get lucky enough to visit a few important states if we want to collect enough data to have low error.

The paper provides a bit of intuition about sufficient conditions for the rank/gap condition to hold. One condition is based on a new variant of Poincaré inequalities; it yields sharp results but requires information that may not be available in practice. Another condition is based on local mixing: e.g., when the states are divided into a few rooms, and the process mixes rapidly within a room and slowly between rooms.

The paper doesn’t spend much time on the question of when good recovery of the successor measure is enough to ensure that RL succeeds. It references prior (classic) results that relate performance of greedy policy improvement to recovery error in (\infty, \infty) norm, as well as prior positive experimental results that do RL from successor measure estimates based on low-rank features. Further information came up in the reviewer-author discussion, and would be helpful to add to the paper. It also seems like RL complexity measures like Bellman rank may be useful in connecting recovery to performance (or if they are not, it would be helpful to have an explanation of why).

Finally, the paper gives numerical experiments: it estimates the successor measure from exploration data in some benchmark environments, then uses it to do greedy policy improvement. In keeping with the theory, all environments are tabular. And in keeping with the theory, nonzero shifts do outperform the previous standard approach of no shift. The experiments also illustrate that moderate values of the shift are the ones that perform best. As pointed out in the discussion, this is helpful since in practice we won’t know enough to estimate the optimal shift, so we need a good range for hyperparameter search. Also pointed out in the discussion is that the experiments section is dense, and a fuller description of the methods would be helpful.

I noticed while compiling this meta-review that there don’t seem to be citations to the literature on predictive state representations / observable operator models; these would be helpful to add, since that line of work is exactly about RL using the assumption of low-rank transitions.

Overall the paper does a good job of setting up an interesting set of questions and goes a long way toward answering them. As an initial attempt, some assumptions are strong (e.g., restriction to tabular case, requirement of full exploration, goal of recovery in a very strong norm), but this is appropriate: the paper is the first to aim at a novel spot in between pure theory and relevance to practice, and mostly hits this spot. The writing is clear (though dense in places), the math brings in appropriate techniques, the bounds seem reasonable, and the paper tries to explain where each term in the bounds comes from.